# SGD with shuffling: optimal rates without component convexity and large epoch requirements

**Kwangjun Ahn\***
Department of EECS
MIT
kjahn@mit.edu

**Chulhee Yun\***
Department of EECS
MIT
chulheey@mit.edu

**Suvrit Sra**
Department of EECS
MIT
suvrit@mit.edu

## Abstract

We study without-replacement SGD for solving finite-sum optimization problems. Specifically, depending on how the indices of the finite-sum are shuffled, we consider the RANDOMSHUFFLE (shuffle at the beginning of each epoch) and SINGLESHUFFLE (shuffle only once) algorithms. First, we establish minimax optimal convergence rates of these algorithms up to poly-log factors. Notably, our analysis is general enough to cover gradient dominated *nonconvex* costs, and does not rely on the convexity of individual component functions unlike existing optimal convergence results. Secondly, assuming convexity of the individual components, we further sharpen the tight convergence results for RANDOMSHUFFLE by removing the drawbacks common to all prior arts: large number of epochs required for the results to hold, and extra poly-log factor gaps to the lower bound.

## 1 Introduction

Stochastic gradient descent (SGD) [8, 16] is a widely used optimization method for solving finite-sum optimization problems that arise in many domains such as machine learning:

$$\underset{\boldsymbol{x}\in\mathbb{R}^d}{\text{minimize}} \quad F(\boldsymbol{x}) := \frac{1}{n}\sum_{i=1}^{n} f_i(\boldsymbol{x}) . \tag{1.1}$$

Given an initial iterate $\boldsymbol{x}_0$, at iteration $t \geq 1$, SGD samples a component index $i(t)$ and updates the current iterate using the (sub)gradient $\boldsymbol{g}_{i(t)}$ of $f_{i(t)}$ at $\boldsymbol{x}_{t-1}$:

$$\boldsymbol{x}_t = \boldsymbol{x}_{t-1} - \eta_t \boldsymbol{g}_{i(t)} \text{ for some step size } \eta_t > 0.$$

There are two versions of SGD, depending on how we sample the index $i(t)$: *with-replacement* and *without-replacement*. With-replacement SGD samples $i(t)$ uniformly and independently from the set of indices $\{1, \ldots, n\}$; hereafter, we use SGD to denote with-replacement SGD. For without-replacement SGD, there are two popular versions in practice. Calling one pass over the entire set of $n$ components an *epoch*, one version randomly shuffles the indices at each epoch (which we call RANDOMSHUFFLE), and the other version shuffles the indices only once and reuses that order for all the epochs (which we call SINGLESHUFFLE).

In modern machine learning applications, RANDOMSHUFFLE and SINGLESHUFFLE are much more widely used than SGD, due to their simple implementations and better empirical performance [2, 3]. However, most theoretical analyses have been devoted to SGD for its easy-to-analyze setting: each stochastic gradient is an i.i.d. unbiased estimate of the full gradient. Whereas for RANDOMSHUFFLE and SINGLESHUFFLE, not only is each stochastic gradient a *biased* estimate of the full gradient,

Table 1: A summary of existing convergence rates and our results for RANDOMSHUFFLE. All the convergence rates are with respect to the suboptimality of objective function value. Note that since the function classes become more restrictive as we go down the table, the noted lower bounds are also valid for upper rows, and the upper bounds are also valid for lower rows. In the "Assumptions" column, inequalities such as $K \gtrsim \kappa^\alpha$ mark the requirements $K \geq C\kappa^\alpha \log(nK)$ for the bounds to hold, and (A1) denotes the assumption that all the iterates remain in a bounded set (see Assumption 1). Also, (LB) stands for "lower bound." For SINGLESHUFFLE, please see Table A in Section F.

| Convergence rates for RANDOMSHUFFLE | | | | |
|---|---|---|---|---|
| Settings | | References | Convergence rates | Assumptions |
| (1) $F$ satisfies PŁ condition | $f_i$ smooth | Haochen and Sra [7][†] | $O\left(\frac{\log^3(nK)}{(nK)^2} + \frac{\log^4(nK)}{K^3}\right)$ | $K \gtrsim \kappa^2$ & (A1) |
| | | Nguyen et al. [13] | $O\left(\frac{1}{K^2}\right)$ | $K \geq 1$ |
| | | **Ours** (Thm 1) | $O\left(\frac{\log^3(nK)}{nK^2}\right)$ | $K \gtrsim \kappa$ |
| | | Rajput et al. [14] | $\Omega\left(\frac{1}{nK^2}\right)$ (LB) | const. step size |
| (2) $F$ strongly convex | $f_i$ smooth | **Ours** (Thm 1) | $O\left(\frac{\log^3(nK)}{nK^2}\right)$ | $K \gtrsim \kappa$ |
| | $f_i$ smooth convex | Nagaraj et al. [11] | $O\left(\frac{\log^2(nK)}{nK^2}\right)$ | $K \gtrsim \kappa^2$ & (A1) |
| | | Mishchenko et al. [10][‡] | $O\left(e^{-\frac{K}{4\kappa}} + \frac{\log^2(nK)}{nK^2}\right)$ | $K \geq 1$ |
| | | **Ours** (Thm 3) | $O\left(\frac{1}{nK^2}\right)$ | $K \geq 1$ & (A1) |
| | | Rajput et al. [14] | $\Omega\left(\frac{1}{nK^2}\right)$ (LB) | const. step size |
| (3) $F$ strongly convex quadratic | $f_i$ smooth quadratic | Gürbüzbalaban et al. [5] | $O\left(\frac{1}{(nK)^2}\right) + o\left(\frac{1}{K^2}\right)$ | asymptotic |
| | | **Ours** (Thm 2) | $O\left(\frac{\log^2(nK)}{(nK)^2} + \frac{\log^4(nK)}{nK^3}\right)$ | $K \gtrsim \kappa$ |
| | $f_i$ smooth quadratic convex | Haochen and Sra [7] | $O\left(\frac{\log^3(nK)}{(nK)^2} + \frac{\log^4(nK)}{K^3}\right)$ | $K \gtrsim \kappa$ & (A1) |
| | | Rajput et al. [14][*] | $O\left(\frac{\log^2(nK)}{(nK)^2} + \frac{\log^3(nK)}{nK^3}\right)$ | $K \gtrsim \kappa^2$ & (A1) |
| | | **Ours** (Thm 4)[*] | $O\left(\frac{1}{(nK)^2} + \frac{1}{nK^3}\right)$ | $K \geq 1$ & (A1) |
| | | Safran and Shamir [17] | $\Omega\left(\frac{1}{(nK)^2} + \frac{1}{nK^3}\right)$ (LB) | const. step size |

[†] They additionally assume that the hessian is Lipschtiz continuous. Note that the lower bound construction [14] does not require the hessian to be Lipschitz continuous; hence, their result does not contradict the lower bound when $K \gtrsim n$.
[‡] They also present a better convergence rate during the initial epochs ($K \lesssim \kappa$) assuming *strongly convex* $f_i$'s.
[*] These results do not require that $f_i$'s are quadratic.

but each sample $i(t)$ is also *dependent* on the previous samples within the epoch. This dependence poses significant challenges toward analyzing shuffling based SGD. Recently, several works have (in part) overcome such challenges and initiated theoretical studies of RANDOMSHUFFLE and SINGLESHUFFLE [5–7, 10, 11, 13, 14, 17, 19].

## 1.1 What is known so far?

We provide in Table 1 a comprehensive summary of the known upper and lower bounds for convergence of RANDOMSHUFFLE. For a similar summary of SINGLESHUFFLE, please refer to Table A in Section F. There are three classes of differentiable functions $F := \frac{1}{n}\sum_{i=1}^n f_i$ considered in the table, in decreasing order of generality: (1) $F$ satisfies the Polyak-Łojasiewicz (PŁ) condition and $f_i$'s are smooth; (2) $F$ is strongly convex and $f_i$'s are smooth; (3) $F$ is strongly convex quadratic and $f_i$'s are quadratic. Many existing results additionally assume that the component functions $f_i$'s are convex and/or all the iterates are bounded.

In this paper, $n$ denotes the number of component functions, $K$ denotes the number of epochs, and $\kappa$ denotes the condition number of the problem: i.e., $\kappa := L/\mu$ where $L$ is the smoothness constant

and $\mu$ is the strong convexity or PŁ constant. All the convergence rates are with respect to the suboptimality of objective function value (i.e., $F(x) - F(x^*)$ for a suitable iterate $x$). In our notation the well-known optimal convergence rate of SGD is $O\left(1/nK\right)$, which we will refer to as the baseline.

**Initial progress.** One of the first works to report progress is due to by Gürbüzbalaban, Ozdaglar, and Parrilo [5, 6] for strongly convex $F$ and smooth quadratic $f_i$'s. They prove an asymptotic convergence rate of $O\left(1/K^2\right)$ for $K$ epochs when $n$ is treated as a constant, both for RANDOMSHUFFLE and SINGLESHUFFLE. This is indeed an asymptotic improvement over the convergence rate of $O\left(1/nK\right)$ achieved by SGD. The scope of the inspiring result in [5], however, does not match the scope of modern machine learning applications for its asymptotic nature and its treatment of $n$ as a constant[1]. Indeed, in modern machine learning, $n$ cannot be regarded as a constant as it is equal to the number of data items in a training set, and the *non-asymptotic* convergence rate is of greater significance as the algorithm is only run a few epochs in practice.

**First non-asymptotic results.** Subsequent recent efforts seek to characterize non-asymptotic convergence rates in terms of both $n$ and $K$. For the same setting as [5], Haochen and Sra [7] develop the first *non-asymptotic* convergence rate of $O\left(\log^3(nK)/(nK)^2 + \log^4(nK)/K^3\right)$ for RANDOMSHUFFLE under the condition $K \gtrsim \kappa \log(nK)$. They extend this result also to smooth functions satisfying the PŁ condition and show the same convergence rate, albeit with an additional Lipschtiz Hessian assumption and a more stringent requirement $K \gtrsim \kappa^2 \log(nK)$. These rates, however, improve upon the baseline rate $O\left(1/nK\right)$ only after $\omega(\sqrt{n})$ epochs.

**Tight upper and lower bounds.** The non-asymptotic results in [7] are strengthened in follow-up works. Nagaraj, Jain, and Netrapalli [11] consider a setting where $F$ is strongly convex and the $f_i$'s are no longer assumed to be quadratic, just convex and smooth. They introduce coupling arguments to prove a non-asymptotic convergence rate of $O\left(\log^2(nK)/nK^2\right)$ for RANDOMSHUFFLE, under the epoch requirement $K \gtrsim \kappa^2 \log(nK)$. Note that this rate is better than the baseline $O\left(1/nK\right)$ as soon as the condition $K \gtrsim \kappa^2 \log(nK)$ is fulfilled.

The result in [11] has motivated researchers to revisit the quadratic finite-sum case and obtain a convergence rate that has a better dependency on $n$ than that of [7]. The first set of results in this direction are given by Safran and Shamir [17], who develop a lower bound of $\Omega\left(1/(nK)^2 + 1/nK^3\right)$ for RANDOMSHUFFLE under the assumption of constant step size. They also prove a lower bound of $\Omega\left(1/nK^2\right)$ for SINGLESHUFFLE[2] under constant step size, and establish matching upper bounds for the *univariate* case up to poly-logarithmic factors, evidencing that their lower bounds are likely to be tight. For RANDOMSHUFFLE, the question of tightness is settled by Rajput, Gupta, and Papailiopoulos [14] who establish the non-asymptotic convergence rate of $O\left(\log^2(nK)/(nK)^2 + \log^3(nK)/nK^3\right)$ under the condition $K \gtrsim \kappa^2 \log(nK)$ by building on the coupling arguments in [11]. Assuming constant step size, they also prove the lower bound $\Omega\left(1/nK^2\right)$ for strongly convex $F$, showing the tightness (up to poly-logarithmic factors) of the result in [11].

Moreover, there is a concurrent work by Mishchenko et al. [10], which establishes a refined analysis and improves upon the epoch requirement of [11] (see Table 1). They also show better convergence rates during the initial epochs ($K \lesssim \kappa$) by additionally assuming that individual components are *strongly* convex. Notably, their analysis for strongly convex individual components applies to SINGLESHUFFLE as well, and guarantees the same rate. We note that [10] also provides analyses of general convex and nonconvex costs.

**Other related works.** Nguyen et al. [13] provide a unified analysis for both RANDOMSHUFFLE and SINGLESHUFFLE, and prove $O\left(1/K^2\right)$ convergence rates for $F$ satisfying the PŁ condition or strong convexity. Although these results do not have epoch requirements, they do *not* beat the baseline rate $O\left(1/nK\right)$ of SGD unless $K \gtrsim n$. Lastly, Shamir [19] considers the case where $F$ is a generalized linear function and shows that without-replacement SGD is not worse than SGD. His proof techniques use tools from transductive learning theory, and as a result, his results only cover the first epoch.

## 1.2 Limitations of the prior arts

Despite such noticeable progress, there are two primary limitations shared by many existing results, including the minimax optimal upper bounds [11, 14]:

- The convergence results assume that the component functions $f_i$'s are convex, which is not necessarily the case in practical applications.[3] For example, the tight rates on strongly convex $F$ [11] and quadratic $F$ [14] are obtained using coupling arguments showing that each iterate of RANDOMSHUFFLE makes progress on par with SGD, which crucially exploits the convexity of individual $f_i$'s. This dependence limits one from extending their results to nonconvex functions.

- The upper bounds require that the number of epochs be of the form $K \gtrsim \kappa^{\alpha} \log(nK)$ for some constant $\alpha \geq 1$, and have extra poly-logarithmic factors of $nK$. In many cases, these limitations stem from choosing a constant step size $\eta$ in the analysis. More specifically, it turns out that one needs to choose a step size of order $\eta \asymp \kappa \log(nK)/nK$, while in order to ensure a sufficient progress during each epoch, one also needs to choose $\eta \lesssim (\kappa^{\alpha-1} n)^{-1}$. From these conditions, the requirement $K \gtrsim \kappa^{\alpha} \log(nK)$ and poly-log factors arise (see Section 6.1 for details).

## 1.3 Summary of our contributions

We overcome the limitations pointed out above. Our theorems can be put into three groups: (i) using techniques that do *not* require individual convexity, we extend the tight convergence bounds of RANDOMSHUFFLE to the more general class of nonconvex PŁ functions, importantly, while also improving the epoch requirements of existing results (Theorems 1 and 2); (ii) by adopting varying step sizes, we prove convergence bounds of RANDOMSHUFFLE that are *free* of epoch requirements and poly-log factors, this time with convexity (Theorems 3 and 4); and (iii) we also prove a *tight* convergence bound of SINGLESHUFFLE for strongly convex functions without individual convexity (Theorem F.1)—see Tables 1 and A for a quick comparison with other results. Since the majority of our results are on RANDOMSHUFFLE, we defer our SINGLESHUFFLE result to Section F of the appendix, to better streamline the flow of the paper.

- In Theorem 1, we prove that if $F$ satisfies the PŁ condition and has a nonempty and compact solution set, and $f_i$'s are smooth, then RANDOMSHUFFLE converges at the rate $O\left(\log^3(nK)/nK^2\right)$. This bound holds as soon as $K \gtrsim \kappa \log(nK)$, and they match the lower bounds up to poly-log factors. Remarkably, Theorem 1 improves upon the epoch requirement $K \gtrsim \kappa^2 \log(nK)$ of an existing bound [11] for RANDOMSHUFFLE on smooth strongly convex functions, *without* needing convexity of $f_i$'s.

- In Theorem 2, we prove a tight upper bound on RANDOMSHUFFLE for strongly convex quadratic functions that improves the existing epoch requirement $K \gtrsim \kappa^2 \log(nK)$ of [14] to $K \gtrsim \kappa \max\{1, \sqrt{\frac{\kappa}{n}}\} \log(nK)$, *without* assuming the $f_i$'s to be convex. We develop a fine-grained analysis on the expectation over random permutations to overcome issues posed by noncommutativity; for instance, we prove contraction bounds for small step sizes which circumvent the need for a conjectured (now false, see [9]) matrix AM-GM inequality of [15].

- Under the additional assumption that the $f_i$'s are convex, we establish the same convergence rates of RANDOMSHUFFLE for smooth strongly convex functions (Theorem 3) and strongly convex quadratic functions (Theorem 4) *without epoch requirements*; i.e., for all $K \geq 1$. The key to obtaining this improvement is to depart from constant step sizes analyzed in most prior works and consider varying step sizes. To analyze such varying step sizes, we develop a variant of Chung's lemma (Lemma D.5) that can handle our case where the convergence rate depends on two parameters $n$ and $K$; this lemma may be of independent interest. Notably, our approach also removes the extra poly-logarithmic factors in the convergence rates.

- Finally, we provide a tight convergence analysis for SINGLESHUFFLE, again without requiring the convexity of individual component functions. Theorem F.1 shows that for smooth strongly convex functions, SINGLESHUFFLE converges at the rate $O\left(\log^3(nK)/nK^2\right)$ as soon as $K \gtrsim \kappa^2 \log(nK)$. We remark that this rate matches (up to poly-log factors) the existing lower bound $\Omega\left(1/nK^2\right)$ [17] proven for a *subclass*, namely, strongly convex quadratic functions.

## 2 Problem setup and notation

We first summarize the notation used in this paper. For a positive integer $a$, we define $[a] := \{1, 2, \ldots, a\}$; and for integers $a, b$ satisfying $a \leq b$, we let $[a : b] := \{a, a+1, \ldots, b-1, b\}$. For a vector $\boldsymbol{v}$, $\|\boldsymbol{v}\|$ denotes its Euclidean norm. Given a function $h(\boldsymbol{x})$, we use $\mathcal{X}_h^* \subseteq \mathbb{R}^d$ to denote its solution set (the set of its global minima). We omit the subscript $h$ when it is clear from the context.

For solving the finite-sum optimization (1.1) with more than one component ($n \geq 2$), we consider RANDOMSHUFFLE and SINGLESHUFFLE over $K$ epochs, i.e., $K$ passes over the $n$ component functions. The distinction between these two methods lies in the way we shuffle the components at each epoch. For RANDOMSHUFFLE, at the beginning of the $k$-th epoch, we pick a random permutation $\sigma_k : [n] \to [n]$ and access component functions in the order $f_{\sigma_k(1)}, f_{\sigma_k(2)}, \ldots, f_{\sigma_k(n)}$. We initialize $\boldsymbol{x}_0^1 := \boldsymbol{x}_0$, and call $\boldsymbol{x}_i^k$ the $i$-th iterate of $k$-th epoch. Then, we update the iterate using the stochastic gradient $\nabla f_{\sigma_k(i)}$ as follows:

$$\boldsymbol{x}_i^k \leftarrow \boldsymbol{x}_{i-1}^k - \eta_i^k \nabla f_{\sigma_k(i)}(\boldsymbol{x}_{i-1}^k), \tag{2.1}$$

where $\eta_i^k$ is the step size for the $i$-th iteration of the $k$-th epoch. We start the next epoch by setting $\boldsymbol{x}_0^{k+1} := \boldsymbol{x}_n^k$. For SINGLESHUFFLE, we randomly pick a permutation $\sigma : [n] \to [n]$ at the first epoch, and use the same permutation over all epochs, i.e., $\sigma_k = \sigma$ for $k \in [K]$.

Next, we introduce a standard assumption for analyzing incremental methods, extensively used in the prior works [6, 7, 11, 14, 20] (see e.g. [6, Assumption 3.8]):

**Assumption 1** (Bounded iterates assumption). *We say the bounded iterates assumption holds if all the iterates $\{\boldsymbol{x}_1^k, \boldsymbol{x}_2^k, \ldots, \boldsymbol{x}_n^k\}_{k \geq 1}$ are uniformly bounded, i.e., stay within some compact set. We only use this assumption for Theorems 3 and 4.*

We remark that the bounded iterates assumption is not stringent as one can enforce this assumption by explicit projections [11] or by adopting adaptive stepsizes [20] if projection is undesirable.

**Function classes studied in this paper.** Let $h : \mathbb{R}^d \to \mathbb{R}$ be a differentiable function. We say $h$ is *L-smooth* if $h(\boldsymbol{y}) \leq h(\boldsymbol{x}) + \langle \nabla h(\boldsymbol{x}), \boldsymbol{y} - \boldsymbol{x} \rangle + \frac{L}{2} \|\boldsymbol{y} - \boldsymbol{x}\|^2$ for all $\boldsymbol{x}, \boldsymbol{y} \in \mathbb{R}^d$. We use $C_L^1(\mathbb{R}^d)$ to denote the class of differentiable and $L$-smooth functions on $\mathbb{R}^d$. A function $h$ is *$\mu$-strongly convex* if $h(\boldsymbol{y}) \geq h(\boldsymbol{x}) + \langle \nabla h(\boldsymbol{x}), \boldsymbol{y} - \boldsymbol{x} \rangle + \frac{\mu}{2} \|\boldsymbol{y} - \boldsymbol{x}\|^2$ for all $\boldsymbol{x}, \boldsymbol{y} \in \mathbb{R}^d$. Lastly, a function $h$ satisfies the *$\mu$-Polyak-Łojasiewicz condition* (also known as *gradient dominance*) if $\frac{1}{2} \|\nabla h(\boldsymbol{x})\|^2 \geq \mu(h(\boldsymbol{x}) - h^*)$ for any $\boldsymbol{x} \in \mathbb{R}^d$, where $h^* = \min_{\boldsymbol{x}} h(\boldsymbol{x})$; we say $h$ is a $\mu$-PŁ function.

## 3 Tight convergence analysis of RANDOMSHUFFLE for PŁ functions

To prove fast convergence rates of RANDOMSHUFFLE, we need to characterize the aggregate progress made over one epoch as a whole. A general property of without-replacement SGD is the following observation due to Nedić and Bertsekas [12, Chapter 2]: for an epoch $k$, assuming that the iterates $\{\boldsymbol{x}_i^k\}_{i=1}^n$ stay close to $\boldsymbol{x}_0^k$, the aggregate update direction will closely approximate the *full* gradient at $\boldsymbol{x}_0^k$, i.e.,

$$\sum_{i=1}^n \nabla f_{\sigma_k(i)}(\boldsymbol{x}_{i-1}^k) \approx \sum_{i=1}^n \nabla f_{\sigma_k(i)}(\boldsymbol{x}_0^k) = \sum_{i=1}^n \nabla f_i(\boldsymbol{x}_0^k) = n \nabla F(\boldsymbol{x}_0^k). \tag{3.1}$$

Making this heuristic approximation (3.1) rigorous, we prove the following theorem:

**Theorem 1** (PŁ class). *Assume that $F$ is $\mu$-PŁ and its solution set $\mathcal{X}^*$ is nonempty and compact. Also, assume each $f_i \in C_L^1(\mathbb{R}^d)$. Consider RANDOMSHUFFLE for the number of epochs $K$ satisfying $K \geq 10\kappa \log(n^{1/2}K)$, step size $\eta_i^k = \eta := \frac{2 \log(n^{1/2}K)}{\mu n K}$, and initialization $\boldsymbol{x}_0$. Then, with probability at least $1 - \delta$, the following bound holds with $G := \sup_{\boldsymbol{x}: F(\boldsymbol{x}) \leq F(\boldsymbol{x}_0)} \max_{i \in [n]} \|\nabla f_i(\boldsymbol{x})\|$ and some constant $c = O(\kappa^3)$[4]:*

$$\min_{k \in [K+1]} F(\boldsymbol{x}_0^k) - F^* \leq \frac{F(\boldsymbol{x}_0) - F^*}{nK^2} + \frac{c \cdot G^2 \cdot \log^2(nK) \log \frac{nK}{\delta}}{nK^2}.$$

**Proof:** See Section 6.1 for a proof sketch and Section A for the full proof. □

**Optimality of convergence rate.** It is important to note that Theorem 1 matches the lower bound $\Omega\left(1/nK^2\right)$ [14] for strongly convex costs, up to poly-logarithmic factors. What is somewhat surprising is that our upper bound holds for a *broader* class of (nonconvex) PŁ functions compared to this lower bounds. Since Theorem 1 applies to subclasses of PŁ functions, it also gives the minimax optimal rates (up to log factors) for smooth strongly convex functions (see Table 1). Notably, Theorem 1 improves the epoch requirement $K \gtrsim \kappa^2 \log(nK)$ of the prior work [11] to $K \gtrsim \kappa \log(n^{1/2}K)$, *without* requiring the convexity of $f_i$'s. Note that in [11], convexity is crucial in the coupling argument to achieve the tight convergence rate.

**Remark 1** (Best iterate v.s. last iterate). Note that Theorem 1 is a guarantee for the best iterate, not the last iterate. However, the best iterate is needed only in pathological cases where some early iterate $\boldsymbol{x}_0^k$ is already too close to the optimum, so that the "noise" dominates the updates and makes the last iterate have worse optimality gap than $\boldsymbol{x}_0^k$. By assuming bounded iterates (Assumption 1) in place of the compactness of $\mathcal{X}^*$, the convergence rate in Theorem 1 holds for the *last* iterate (see Section A.1 for details). Conversely, for an existing last-iterate bound (e.g., [14]), one can prove a corresponding best-iterate bound without Assumption 1 if $\mathcal{X}^*$ is compact.

**Remark 2** (Similar bound for SINGLESHUFFLE). In Section F, we show a similar convergence bound $O\left(\log^3(nK)/nK^2\right)$ for SINGLESHUFFLE on smooth strongly convex functions (Theorem F.1). Interestingly, Theorems 1 and F.1 together demonstrate that the optimal dependences on $n$ and $K$ are identical for RANDOMSHUFFLE and SINGLESHUFFLE on smooth strongly convex costs; in other words, for this function class, there is no additional provable gain from reshuffling in terms of the dependence on $n$ and $K$.

From the high-probability bound in Theorem 1, we can derive a corresponding expectation bound.

**Corollary 1** (PŁ class). *Under the same setting as Theorem 1, the following bound holds:*

$$\mathbb{E}\left[\min_{k\in[K+1]} F(\boldsymbol{x}_0^k)\right] - F^* \leq \frac{3(F(\boldsymbol{x}_0) - F^*)}{2nK^2} + \frac{c \cdot G^2 \cdot \log^3(nK)}{nK^2} .$$

So far, we have developed the optimal convergence rate of RANDOMSHUFFLE for PŁ costs, which turn out to be also optimal for smooth strongly convex costs. However, there is one case which Theorem 1 does not match the lower bound, namely RANDOMSHUFFLE for quadratic costs: the lower bound of $\Omega(1/(nK)^2 + 1/nK^3)$ is proved in [17]. Although Rajput et al. [14] actually obtain this rate, they assume the convexity of each component. In light of Theorem 1, it is therefore natural to ask if we can close the gap without assuming convexity of each component.

## 4 Tight bound on RANDOMSHUFFLE for quadratic functions

We prove below a tight bound for RANDOMSHUFFLE on quadratics without the convexity of $f_i$'s. For simplicity, we assume (without loss) that the global optimum of $F$ is achieved at the origin.

**Theorem 2** (Quadratic costs). *Assume that $F(\boldsymbol{x}) := \frac{1}{n}\sum_{i=1}^n f_i(\boldsymbol{x}) = \frac{1}{2}\boldsymbol{x}^T \boldsymbol{A}\boldsymbol{x}$ and $F$ is $\mu$-strongly convex. Let $f_i(\boldsymbol{x}) := \frac{1}{2}\boldsymbol{x}^T \boldsymbol{A}_i\boldsymbol{x} + \boldsymbol{b}_i^T \boldsymbol{x}$ and $f_i \in C_L^1(\mathbb{R}^d)$. Consider RANDOMSHUFFLE for the number of epochs $K$ satisfying $K \geq \frac{32}{3}\kappa \max\{1, \sqrt{\frac{\kappa}{n}}\}\log(nK)$, step size $\eta_i^k = \eta := \frac{2\log(nK)}{\mu nK}$, and initialization $\boldsymbol{x}_0$. Then for $G := \max_{i\in[n]}\|\boldsymbol{b}_i\|$ and some constant $c = O(\kappa^4)$,*

$$\mathbb{E}\left[F(\boldsymbol{x}_0^{K+1})\right] - F^* \leq \frac{2L\|\boldsymbol{x}_0 - \boldsymbol{x}^*\|^2}{n^2K^2} + \frac{c \cdot G^2 \cdot \log^2(nK)}{n^2K^2} + \frac{c \cdot G^2 \cdot \log^4(nK)}{nK^3} .$$

**Proof:** See Section 6.2 for a proof sketch and Section B for the full proof. ☐

**Improvements.** Theorem 2 improves the prior result [14] in many ways. Most importantly, Theorem 2 does not require $f_i$'s to be convex, an assumption exploited in [14] for their coupling argument. Moreover, Theorem 2 imposes a *milder* epoch requirement: it only assumes $K \geq \frac{32}{3}\kappa \max\{1, \sqrt{\frac{\kappa}{n}}\}\log(nK)$, which is better than $K \geq 128\kappa^2\log(nK)$ of [14]. As long as $\kappa \leq n$, our epoch requirement is $K \gtrsim \kappa\log(nK)$, matching that of the univariate case [17]. Lastly, unlike [14], Theorem 2 does not rely on the bounded iterates assumption.

**Remark 3** (Tail averaging tricks)**.** Following [11], we can also obtain a guarantee for the tail average of the iterates $\boldsymbol{x}_0^{\lceil K/2 \rceil}, \ldots, \boldsymbol{x}_0^K$, which improves the constants appearing in the bound by a factor of $\kappa$. Due to space limitation, the statement and the proof of this improvement are deferred to Section C.

Thus far, we have established the optimal convergence rates of RANDOMSHUFFLE for PŁ costs, strongly convex costs, and quadratic costs *without* assuming the convexity of individual components, an assumption crucial to the analysis of prior arts [11, 14]. Due to our results, it may seem that there is no additional gain from the convexity of the $f_i$'s. *Is it really the case?*

## 5   Eliminating epoch requirements with varying step sizes

In this section, we show that the convexity of $f_i$'s *does* lead to gains. In particular, we show how this convexity helps one remove the epoch requirements as well as extra poly-log terms in previous convergence bounds [7, 11, 14]. The main technical distinction of this sharper result is to depart from constant step sizes and consider varying step sizes. We begin with the strongly convex case:

**Theorem 3** (Strongly convex costs)**.** *Assume that $F$ is $\mu$-strongly convex, and each $f_i$ is convex and $f_i \in C_L^1(\mathbb{R}^d)$. Assume that the bounded iterates assumption (Assumption 1) holds. For any constant $\alpha > 2$, let $k_0 := \alpha \cdot \kappa$, and consider the step sizes $\eta_i^1 = \frac{2\alpha/\mu}{k_0+i}$ for $i \in [n]$, and $\eta_i^k = \frac{2\alpha/\mu}{k_0+nk}$ for $k \in [2 : K]$ and $i \in [n]$. Then, for any $K \geq 1$, the following convergence bound holds for RANDOMSHUFFLE with step sizes $\eta_i^k$, initialization $\boldsymbol{x}_0$, and some $c_1 = O(\kappa^4)$ and $c_2 = O(\kappa^\alpha)$:*

$$\mathbb{E}\left[F(\boldsymbol{x}_0^{K+1})\right] - F^* \leq \frac{c_1 \cdot n}{(k_0 + nK)^2} + \frac{c_2 \cdot \|\boldsymbol{x}_0 - \boldsymbol{x}^*\|^2}{(k_0 + nK)^\alpha} .$$

**Proof:** See Section 6.3 for a proof sketch and Section D.5 for the full proof. □

**Removing epoch requirements.**   The most important feature of Theorem 3 is that its rate holds for all $K \geq 1$. This is in stark contrast with the existing minimax optimal result [11] which requires $K \gtrsim \kappa^2 \log(nK)$. Moreover, the dependency of the leading constant $c_1$ on $\kappa$ is identical with [11][5].

**Removing extra poly-log factors.**   Another notable feature of our bound is that it is not beset with extra poly-log factors appearing in the previous minimax optimal result [11], and thereby it closes the poly-log gap between the lower bound [14].

**Remark 4.**   We compare our result with that of a concurrent work [10]. In particular, for the same setting as Theorem 3, they obtain the convergence rate $O(e^{-K/4\kappa} + \log^2(nK)/nK^2)$ for all $K \geq 1$. Hence, to obtain $\epsilon$-approximation solution, the iteration complexity reads $\Omega(\kappa n \cdot \log(1/\epsilon) + \sqrt{n/\epsilon})$. In contrast, the iteration complexity of our result reads $\Omega(\kappa \cdot \epsilon^{-1/\alpha} + \sqrt{n/\epsilon})$. Hence, we expect that for the practical setting where $n$ is in a much larger scale than $\epsilon^{-1/\alpha}$ (note that $\alpha > 2$), our bound gives a better result. Note also that our result does not come with extra poly-log factors, which requires a nontrivial modification of Chung's lemma as we outlined in Section 6.3.

With a similar technique, we can also prove the tight convergence rates for the case of quadratic $F$:

**Theorem 4** (Quadratic costs)**.** *Under the setting of Theorem 3, we additionally assume that $F$ is quadratic. For any constant $\alpha > 4$, consider the same varying step sizes as in Theorem 3. Then, for any $K \geq 1$, the following convergence bound holds for RANDOMSHUFFLE with step sizes $\eta_i^k$, initialization $\boldsymbol{x}_0$, and some $c_1 = O(\kappa^4)$, $c_2 = O(\kappa^6)$ and $c_3 = O(\kappa^\alpha)$:*

$$\mathbb{E}\left[F(\boldsymbol{x}_0^{K+1})\right] - F^* \leq \frac{c_1}{(k_0 + nK)^2} + \frac{c_2 \cdot n^2}{(k_0 + nK)^3} + \frac{c_3 \cdot \|\boldsymbol{x}_0 - \boldsymbol{x}^*\|^2}{(k_0 + nK)^\alpha} .$$

**Proof:** Similar to Theorem 3. See Section D.6. □

**Remark 5** (Tightness of bounds)**.** We believe that the upper bounds developed in this section are likely tight, though we note that this tightness is not yet guaranteed because the existing lower bounds are all developed under the assumption of constant step sizes. Extension of the lower bounds to varying step sizes would be an interesting future research direction.

# 6 Proof sketches

## 6.1 Proof sketch of Theorem 1

As we mentioned in Section 3, the key to obtaining a faster rate is to capture the per-epoch property (3.1). By decomposing $\nabla f_{\sigma_k(i)}(\boldsymbol{x}_{i-1}^k)$ into "signal" $\nabla f_{\sigma_k(i)}(\boldsymbol{x}_0^k)$ and "noise," we develop the following approximate version of (3.1) by carefully expanding the updates (2.1) over the $k$-th epoch:

$$\boldsymbol{x}_0^{k+1} = \boldsymbol{x}_0^k - \eta n \nabla F(\boldsymbol{x}_0^k) + \eta^2 \boldsymbol{r}_k \,, \tag{3.1$'$}$$

where the error term $\boldsymbol{r}_k$ is defined as the sum $\sum_{i=1}^{n-1} \boldsymbol{M}_i \sum_{j=1}^{i} \nabla f_{\sigma_k(j)}(\boldsymbol{x}_0^k)$ for some matrices $\boldsymbol{M}_i$'s of bounded spectral norm. Note that without the term $\eta^2 \boldsymbol{r}_k$, (3.1$'$) is precisely equal to gradient descent update with the cost function $F$. By smoothness of $F$, we have

$$
\begin{aligned}
F(\boldsymbol{x}_0^{k+1}) - F(\boldsymbol{x}_0^k) &\leq \left\langle \nabla F(\boldsymbol{x}_0^k), \boldsymbol{x}_0^{k+1} - \boldsymbol{x}_0^k \right\rangle + \tfrac{L}{2} \left\| \boldsymbol{x}_0^{k+1} - \boldsymbol{x}_0^k \right\|^2 \\
&\leq (-\eta n + \eta^2 n^2 L) \left\| \nabla F(\boldsymbol{x}_0^k) \right\|^2 + \eta^2 \left\| \nabla F(\boldsymbol{x}_0^k) \right\| \left\| \boldsymbol{r}_k \right\| + L\eta^4 \left\| \boldsymbol{r}_k \right\|^2 . 
\end{aligned} \tag{6.1}
$$

Therefore, it is important to control the norm $\|\boldsymbol{r}_k\|$ of the aggregate error term. We seek sharp bounds on $\|\boldsymbol{r}_k\|$ but cannot invoke a standard concentration inequality as the gradients are sampled *without-replacement*. We overcome this difficulty by applying a vector-valued version of Hoeffding-Serfling[6] inequality [18] to the partial sums $\sum_{j=1}^{i} \nabla f_{\sigma_k(j)}(\boldsymbol{x}_0^k)$. For each of them, we have

$$\left\| \sum_{j=1}^{i} \nabla f_{\sigma_k(j)}(\boldsymbol{x}_0^k) \right\| \leq \left\| \sum_{j=1}^{i} \nabla f_{\sigma_k(j)}(\boldsymbol{x}_0^k) - i \nabla F(\boldsymbol{x}_0^k) \right\| + i \left\| \nabla F(\boldsymbol{x}_0^k) \right\| \lesssim \sqrt{i} + i \left\| \nabla F(\boldsymbol{x}_0^k) \right\| ,$$

with high probability. Applying the union bound and summing over all $i \in [n-1]$, we obtain

$$\| \boldsymbol{r}_k \| \lesssim n^{3/2} + n^2 \left\| \nabla F(\boldsymbol{x}_0^k) \right\| . \tag{6.2}$$

Substituting this bound (6.2) into (6.1), rearranging the terms, and applying the PŁ inequality on $\|\nabla F(\boldsymbol{x}_0^k)\|^2$, we obtain the following per-epoch progress bound, which holds for $\eta \leq \frac{1}{5nL}$:

$$F(\boldsymbol{x}_0^{k+1}) - F^* \lesssim (1 - \eta n \mu)(F(\boldsymbol{x}_0^k) - F^*) + \eta^3 n^2. \tag{6.3}$$

Applying (6.3) over all epochs and substituting $\eta = \frac{2 \log(n^{1/2} K)}{\mu n K}$, the convergence rate follows. Substituting $\eta$ to $\eta \leq \frac{1}{5nL}$ gives the epoch requirement. See Section A for details. $\qquad\square$

## 6.2 Proof sketch of Theorem 2

The proof builds on the techniques from [17] for one-dimensional quadratic functions. In place of the per-epoch analysis in Theorem 1, we recursively apply (2.1) all the way from the initial iterate $\boldsymbol{x}_0^1$ to the last iterate $\boldsymbol{x}_0^{K+1}$ and directly bound $\mathbb{E}[\|\boldsymbol{x}_0^{K+1} - \boldsymbol{x}^*\|^2]$. Indeed, as pointed out by Safran and Shamir [17], the main technical difficulty in extending this approach to higher dimensions comes from the noncommutativity of matrix multiplication which, for example, results in the absence of the matrix AM-GM inequality [9, 15]. Through a fine-grained analysis of the expectation over uniform permutation, we overcome the problems posed by the noncommutativity and develop a tight upper bound. For instance, we prove the following contraction bound (Lemma B.1) as an approximate alternative to the recently disproved matrix AM-GM inequality [9], which holds for small enough $\eta$:

$$\left\| \mathbb{E}\left[ \prod_{t=1}^{n} (1 - \eta \boldsymbol{A}_{\sigma_k(t)}) \prod_{t=n}^{1} (1 - \eta \boldsymbol{A}_{\sigma_k(t)}) \right] \right\| \leq 1 - \eta n \mu . \tag{6.4}$$

Although (6.4) only holds for small $\eta$ and is looser than the AM-GM inequality, it is remarkable that this bound holds for *any* $n \geq 2$, especially given the result [9] that the matrix AM-GM inequality conjecture breaks as soon as $n = 5$. Please refer to Section B for the full proof. $\qquad\square$

### 6.3 Proof sketch of Theorem 3

First, due to the convexity of $f_i$'s, it turns out that not only one can characterize the per-epoch progress bound akin to (6.3), but also the progress made by each iteration (Proposition D.1). This per-iteration progress bound is due to [11], which uses coupling arguments to demonstrate that with convexity, RANDOMSHUFFLE makes progress on par with SGD.

Having such a fine control over the progress made by RANDOMSHUFFLE, one can imagine that the varying step size choice takes *aggressive* steps in the initial epochs which in turn results in a warm start. Despite this simple intuition, it turns out that the rigorous analysis is non-trivial. In fact, there is a technical tool called *non-asymptotic Chung's lemma* [4] for turning individual progress made at each iteration/epoch into a global convergence bound. However, as we illustrate in Section D.3, the non-asymptotic Chung's lemma does not yield the desired convergence rate; the main issue is that for RANDOMSHUFFLE, the convergence bound needs to capture the right order for *two* parameters $n$ and $K$. To overcome such a limitation, we develop a variant of Chung's lemma (Lemma D.5) which gives rise to the bound that captures the right order for both $n$ and $K$. See Section D for the full proof. $\square$

## 7 Conclusion and future work

Motivated by some limitations of the previous efforts, this paper establishes optimal convergence rates of RANDOMSHUFFLE and SINGLESHUFFLE. Notably, our optimal convergence rates are obtained without relying on convex component functions, which are exploited in the prior works [11, 14]. We also show that exploiting the convexity of component functions allows for further improvements for RANDOMSHUFFLE. By adopting time-varying step sizes and applying a variant of Chung's lemma, we develop sharper convergence bounds that do not come with any epoch requirement and extra poly-log factors. We conclude this paper with several interesting open questions:

- (*Extending lower bounds*) As noted in Remark 5, all tight lower bounds known to date hold for constant step sizes and last iterates. It would be interesting to extend these bounds for more general settings, e.g., varying step size and arbitrary linear combination of iterates.

- (*Optimal rates for Lipschitz Hessian class*) Currently, there is a gap between the lower and upper bounds of RANDOMSHUFFLE for the smooth, strongly convex costs with the Lipschitz Hessian assumption: The best known lower bound is $\Omega\left(1/(nK)^2 + 1/nK^3\right)$ [17], while the best known upper bounds are $O\left(1/nK^2\right)$ ([11] and ours) and $O\left(1/(nK)^2 + 1/K^3\right)$ [7]. Closing this gap would be of interest.

- (*Other cost functions*) It is also worthwhile to investigate if without-replacement SGD achieves superior convergence rates over SGD for other classes of convex or nonconvex functions.

- (*Removing epoch requirements*) Our varying step size technique only works under the additional assumption that $f_i$'s are convex (Section 5). It would be interesting to see if such improvements can be made without relying on the convexity assumption, or for more general functions.

- (*Superiority of without-replacement for the first epoch*) Is without-replacement SGD faster than SGD even during the first epoch? It is demonstrated in [7, Section 7.3] that if $f_i$'s are strongly convex and all $f_i$'s share a common minimum point $x^*$, this is indeed true. However, for quadratic $f_i$'s that are not strongly convex, showing this is closely tied to the matrix AM-GM inequality [15], which was recently proven to be false [9].

## Broader Impact

This work is about developing theoretical guarantees for widely used stochastic optimization methods. Therefore, the discussion on its ethical aspects or future societal consequences is not particularly relevant. However, this work definitely brings new insights into the practical methods, which could possibly impact other ML researches.

## Acknowledgments and Disclosure of Funding

All authors were supported from NSF CAREER grant 1846088. CY was supported from Korea Foundation for Advanced Studies. KA was supported from Kwanjeong Educational Foundation.

The authors appreciate Itay Safran and Ohad Shamir for catching an error in the initial claim about SINGLESHUFFLE.

## Footnotes

*The first two authors contributed equally to this work.

[1]Although one can actually deduce the asymptotic convergence rate of $O\left(1/(nK)^2\right) + o\left(1/K^2\right)$ closely following their arguments [5, (58)], the lower order term does not show the exact dependence on $n$.

[2]Note that these lower bounds hold for more general function classes as well.

[3]For instance, in nonconvex optimization problems such as neural network training, the function $F$ would behave like a convex function in a neighborhood around a local minimum, while each component function $f_i$ could be highly nonconvex in the neighborhood.

[4]Throughout this paper, we adopt the convention that $\kappa = \Theta(1/\mu)$, used in prior works [7, 11].

[5]Although the leading constant in [11, Theorem 1] actually reads $O\left(\kappa^3\right)$, it is important to note that the result is for the tail-averaged iterate. For the last iterate, one can see that their leading constant becomes $O\left(\kappa^4\right)$.

[6] The scalar-valued version of Hoeffding-Serfling inequality [1] was also used in [17].

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
