[Supplementary Material]

# Contents

# A   Analysis for PŁ costs (Proofs of Theorem 1 and Corollary 1)

## A.1   Proof outline

In this section, we present the proof of Theorem 1 and Corollary 1. We first show the existence of the following quantity that will be used throughout the proof:

$$G := \sup_{\boldsymbol{x}:\ F(\boldsymbol{x}) \leq F(\boldsymbol{x}_0)} \max_{i \in [n]} \|\nabla f_i(\boldsymbol{x})\| \ .$$

With this quantity, as long as all the iterates stay within the sublevel set $\mathcal{S}_{\boldsymbol{x}_0} := \{x \ : \ F(\boldsymbol{x}) \leq F(\boldsymbol{x}_0)\}$, one can regard each component function $f_i$ as being $G$-Lipschitz. This motivates us to consider the following two cases:

1. In the first case, we assume that all the end-of-epoch iterates $\boldsymbol{x}_0^k$ stay in the sublevel set $\mathcal{S}_{\boldsymbol{x}_0}$.

2. In the second case, we assume that there exists an end-of-epoch iterate $\boldsymbol{x}_0^k \notin \mathcal{S}_{\boldsymbol{x}_0}$.

In both cases, we will show that the best end-of-epoch iterate satisfies

$$\min_{k \in [K+1]} F(\boldsymbol{x}_0^k) - F^* \leq \frac{F(\boldsymbol{x}_0) - F^*}{nK^2} + \mathcal{O}\left( \frac{L^2 G^2}{\mu^3} \frac{\log^2(n^{1/2}K) \log \frac{nK}{\delta}}{nK^2} \right),$$

with high probability.

**Existence of $G$.**   Recall that the function $F : \mathbb{R}^d \to \mathbb{R}$ is $\mu$-PŁ, and the set $\mathcal{X}^*$ of the global optima of $F$ is nonempty and compact. Also, it is a standard fact [4, Theorem 2] that $\mu$-PŁ functions also satisfy the following quadratic growth: Denoting by $\boldsymbol{x}^*$ the closest global optimum to the point $\boldsymbol{x}$ (i.e., the projection of $\boldsymbol{x}$ onto the solution set $\mathcal{X}^*$),

$$F(\boldsymbol{x}) - F^* \geq 2\mu \|\boldsymbol{x} - \boldsymbol{x}^*\|^2 \ .$$

Then, due to the quadratic growth property, it is easy to verify:

$$\mathcal{S}_{\boldsymbol{x}_0} = \{\boldsymbol{x} \in \mathbb{R}^d \mid F(\boldsymbol{x}) \leq F(\boldsymbol{x}_0)\} \subset \left\{\boldsymbol{x} \in \mathbb{R}^d \mid \|\boldsymbol{x} - \boldsymbol{x}^*\|^2 \leq \frac{F(\boldsymbol{x}_0) - F^*}{2\mu}\right\}.$$

Indeed, the inclusion follows since for any $\boldsymbol{x} \in \mathcal{S}_{\boldsymbol{x}_0}$, $F(\boldsymbol{x}_0) - F^* \geq F(\boldsymbol{x}) - F^* \geq 2\mu \|\boldsymbol{x} - \boldsymbol{x}^*\|^2$, which implies $\boldsymbol{x}$ is also in the latter set. Since we assumed that $\mathcal{X}^*$ is compact, $\mathcal{S}_{\boldsymbol{x}_0}$ is also bounded and hence compact. Since $\nabla f_i$ is continuous on a compact set $\mathcal{S}_{\boldsymbol{x}_0}$, there must exist a constant $0 \leq G < \infty$ such that $\|\nabla f_i(\boldsymbol{x})\| \leq G$ for all $i \in [n], \boldsymbol{x} \in \mathcal{S}_{\boldsymbol{x}_0}$.

**What if the bounded iterates assumption holds?**   As noted in Remark 1, if we have the bounded iterates assumption (Assumption 1), one can prove the same bound for the last iterate $\boldsymbol{x}_0^{K+1}$, modulo leading constants. This is because if we have Assumption 1, we have a compact set $\mathcal{S}$ which all the

end-of-epoch iterates $\boldsymbol{x}_0^k$ lie in, which corresponds to the first case of the proof. More specifically, there exists a constant $0 \le G' < \infty$ such that

$$\|\nabla f_i(\boldsymbol{x})\| \le G' \text{ for all } i \in [n], \boldsymbol{x} \in \mathcal{S}.$$

Thus, the proof for the first case stated in Sections A.2–A.4 goes through, modulo $G$ replaced by $G'$. We remark that since we already have a compact set $\mathcal{S}$, we no longer need the additional compactness assumption on $\mathcal{X}^*$.

## A.2 The 1st case: characterizing aggregate update over an epoch

We start by recursively applying the update equations over an epoch. The key idea in doing so is to decompose the gradient $\nabla f_{\sigma_k(i)}(\boldsymbol{x}_{i-1}^k)$ into the "signal" $\nabla f_{\sigma_k(i)}(\boldsymbol{x}_0^k)$ and a noise term:

$$
\begin{aligned}
\nabla f_{\sigma_k(i)}(\boldsymbol{x}_{i-1}^k) &= \nabla f_{\sigma_k(i)}(\boldsymbol{x}_0^k) + \nabla f_{\sigma_k(i)}(\boldsymbol{x}_{i-1}^k) - \nabla f_{\sigma_k(i)}(\boldsymbol{x}_0^k) \\
&= \underbrace{\nabla f_{\sigma_k(i)}(\boldsymbol{x}_0^k)}_{=:\boldsymbol{g}_{\sigma_k(i)}} + \underbrace{\left[\int_0^1 \nabla^2 f_{\sigma_k(i)}(\boldsymbol{x}_0^k + t(\boldsymbol{x}_{i-1}^k - \boldsymbol{x}_0^k))dt\right]}_{=:\boldsymbol{H}_{\sigma_k(i)}}(\boldsymbol{x}_{i-1}^k - \boldsymbol{x}_0^k) \\
&= \boldsymbol{g}_{\sigma_k(i)} + \boldsymbol{H}_{\sigma_k(i)}(\boldsymbol{x}_{i-1}^k - \boldsymbol{x}_0^k),
\end{aligned}
$$

where $\nabla^2 f_i(\boldsymbol{x})$ denotes the Hessian of $f_i$ at $\boldsymbol{x}$, whenever it exists. We remark that the integral $\boldsymbol{H}_{\sigma_k(i)}$ exists, due to the following reason. Since we assumed that each $f_{\sigma_k(i)} \in C_L^1(\mathbb{R}^d)$, its gradient $\nabla f_{\sigma_k(i)}$ is Lipschitz continuous, and hence absolutely continuous. This means that $\nabla f_{\sigma_k(i)}$ is differentiable almost everywhere (i.e., $\nabla^2 f_{\sigma_k(i)}(\boldsymbol{x})$ exists a.e.), and the fundamental theorem of calculus for Lebesgue integral holds; hence the integral exists. Note that $\|\boldsymbol{H}_{\sigma_k(i)}\| \le L$ due to $L$-smoothness of $f_i$'s. We now substitute this decomposition to the update equations. First,

$$\boldsymbol{x}_1^k = \boldsymbol{x}_0^k - \eta \boldsymbol{g}_{\sigma_k(1)}.$$

Substituting this to $\boldsymbol{x}_2^k$ gives

$$
\begin{aligned}
\boldsymbol{x}_2^k &= \boldsymbol{x}_1^k - \eta \nabla f_{\sigma_k(2)}(\boldsymbol{x}_1^k) = \boldsymbol{x}_1^k - \eta \boldsymbol{g}_{\sigma_k(2)} - \eta \boldsymbol{H}_{\sigma_k(2)}(\boldsymbol{x}_1^k - \boldsymbol{x}_0^k) \\
&= \boldsymbol{x}_0^k - \eta \boldsymbol{g}_{\sigma_k(1)} - \eta \boldsymbol{g}_{\sigma_k(2)} + \eta^2 \boldsymbol{H}_{\sigma_k(2)} \boldsymbol{g}_{\sigma_k(1)} = \boldsymbol{x}_0^k - \eta(\boldsymbol{I} - \eta \boldsymbol{H}_{\sigma_k(2)})\boldsymbol{g}_{\sigma_k(1)} - \eta \boldsymbol{g}_{\sigma_k(2)}.
\end{aligned}
$$

Repeating this process until $\boldsymbol{x}_n^k = \boldsymbol{x}_0^{k+1}$, we get

$$
\begin{aligned}
\boldsymbol{x}_0^{k+1} &= \boldsymbol{x}_0^k - \eta \sum_{j=1}^n \left(\prod_{t=n}^{j+1}(\boldsymbol{I} - \eta \boldsymbol{H}_{\sigma_k(t)})\right) \boldsymbol{g}_{\sigma_k(j)} \\
&= \boldsymbol{x}_0^k - \eta n \nabla F(\boldsymbol{x}_0^k) - \eta \left[\sum_{j=1}^n \left(\prod_{t=n}^{j+1}(\boldsymbol{I} - \eta \boldsymbol{H}_{\sigma_k(t)})\right) \boldsymbol{g}_{\sigma_k(j)} - n \nabla F(\boldsymbol{x}_0^k)\right].
\end{aligned}
$$

Due to summation by parts, the following identity holds:

$$\sum_{j=1}^n a_j b_j = a_n \sum_{j=1}^n b_j - \sum_{i=1}^{n-1}(a_{i+1} - a_i) \sum_{j=1}^i b_j.$$

We apply this to the last term, by substituting $a_j = \prod_{t=n}^{j+1}(\boldsymbol{I} - \eta \boldsymbol{H}_{\sigma_k(t)})$ and $b_j = \boldsymbol{g}_{\sigma_k(j)}$:

$$
\begin{aligned}
&\sum_{j=1}^n \left(\prod_{t=n}^{j+1}(\boldsymbol{I} - \eta \boldsymbol{H}_{\sigma_k(t)})\right) \boldsymbol{g}_{\sigma_k(j)} - n \nabla F(\boldsymbol{x}_0^k) \\
&= \sum_{j=1}^n \boldsymbol{g}_{\sigma_k(j)} - \sum_{i=1}^{n-1} \left(\prod_{t=n}^{i+2}(\boldsymbol{I} - \eta \boldsymbol{H}_{\sigma_k(t)}) - \prod_{t=n}^{i+1}(\boldsymbol{I} - \eta \boldsymbol{H}_{\sigma_k(t)})\right) \sum_{j=1}^i \boldsymbol{g}_{\sigma_k(j)} - n \nabla F(\boldsymbol{x}_0^k) \\
&= -\eta \underbrace{\sum_{i=1}^{n-1} \left(\prod_{t=n}^{i+2}(\boldsymbol{I} - \eta \boldsymbol{H}_{\sigma_k(t)})\right) \boldsymbol{H}_{\sigma_k(i+1)} \sum_{j=1}^i \boldsymbol{g}_{\sigma_k(j)}}_{=:\boldsymbol{r}_k}.
\end{aligned}
$$

Therefore, we have $\boldsymbol{x}_0^{k+1} = \boldsymbol{x}_0^k - \eta n \nabla F(\boldsymbol{x}_0^k) + \eta^2 \boldsymbol{r}_k$. By smoothness of $F$, we have

$$
\begin{aligned}
F(\boldsymbol{x}_0^{k+1}) &- F(\boldsymbol{x}_0^k) \\
&\leq \left\langle \nabla F(\boldsymbol{x}_0^k), \boldsymbol{x}_0^{k+1} - \boldsymbol{x}_0^k \right\rangle + \frac{L}{2} \left\| \boldsymbol{x}_0^{k+1} - \boldsymbol{x}_0^k \right\|^2 \\
&\leq -\eta n \left\| \nabla F(\boldsymbol{x}_0^k) \right\|^2 + \eta^2 \left\| \nabla F(\boldsymbol{x}_0^k) \right\| \left\| \boldsymbol{r}_k \right\| + \frac{L\eta^2}{2} \left\| n\nabla F(\boldsymbol{x}_0^k) + \eta \boldsymbol{r}_k \right\|^2 \\
&\leq (-\eta n + \eta^2 n^2 L) \left\| \nabla F(\boldsymbol{x}_0^k) \right\|^2 + \eta^2 \left\| \nabla F(\boldsymbol{x}_0^k) \right\| \left\| \boldsymbol{r}_k \right\| + L\eta^4 \left\| \boldsymbol{r}_k \right\|^2,
\end{aligned}
\tag{A.1}
$$

where the last inequality used $\|\boldsymbol{a} + \boldsymbol{b}\|^2 \leq 2\|\boldsymbol{a}\|^2 + 2\|\boldsymbol{b}\|^2$.

## A.3 The 1st case: bounding noise term using Hoeffding-Serfling inequality

It is left to bound $\|\boldsymbol{r}_k\|$. We have

$$
\begin{aligned}
\|\boldsymbol{r}_k\| &= \left\| \sum_{i=1}^{n-1} \left( \prod_{t=n}^{i+2} (\boldsymbol{I} - \eta \boldsymbol{H}_{\sigma_k(t)}) \right) \boldsymbol{H}_{\sigma_k(i+1)} \sum_{j=1}^{i} \boldsymbol{g}_{\sigma_k(j)} \right\| \\
&\leq \sum_{i=1}^{n-1} \left\| \left( \prod_{t=n}^{i+2} (\boldsymbol{I} - \eta \boldsymbol{H}_{\sigma_k(t)}) \right) \boldsymbol{H}_{\sigma_k(i+1)} \sum_{j=1}^{i} \boldsymbol{g}_{\sigma_k(j)} \right\| \\
&\leq L(1 + \eta L)^n \sum_{i=1}^{n-1} \left\| \sum_{j=1}^{i} \boldsymbol{g}_{\sigma_k(j)} \right\|.
\end{aligned}
\tag{A.2}
$$

Where the last step used $\|\boldsymbol{H}_{\sigma_k(j)}\| \leq L$. Recall from the theorem statement that $K \geq 10\kappa \log(n^{1/2}K)$, and $\eta = \frac{2 \log(n^{1/2}K)}{\mu nK}$. This means that

$$
\eta L = \frac{2\kappa \log(n^{1/2}K)}{nK} \leq \frac{1}{5n},
$$

which implies $(1 + \eta L)^n \leq e^{1/5}$. Now, we use the Hoeffding-Serfling inequality for bounded random vectors, which is taken from [13, Theorem 2]. Note that for any epoch $k$, the permutation $\sigma_k$ is independent of the first iterate $\boldsymbol{x}_0^k$ of the epoch. Therefore, we can apply the following bound for partial sums of $\boldsymbol{g}_{\sigma_k(i)} := \nabla f_{\sigma_k(i)}(\boldsymbol{x}_0^k)$:

**Lemma A.1** ([13, Theorem 2]). *Suppose $n \geq 2$. Let $\boldsymbol{v}_1, \boldsymbol{v}_2, \ldots, \boldsymbol{v}_n \in \mathbb{R}^d$ satisfy $\|\boldsymbol{v}_j\| \leq G$ for all $j$. Let $\bar{\boldsymbol{v}} = \frac{1}{n} \sum_{j=1}^{n} \boldsymbol{v}_j$. Let $\sigma \in \mathcal{S}_n$ be a uniform random permutation of $n$ elements. Then, for $i \leq n$, with probability at least $1 - \delta$, we have*

$$
\left\| \frac{1}{i} \sum_{j=1}^{i} \boldsymbol{v}_{\sigma(j)} - \bar{\boldsymbol{v}} \right\| \leq G \sqrt{\frac{8(1 - \frac{i-1}{n}) \log \frac{2}{\delta}}{i}}.
$$

Recall the mean $\bar{\boldsymbol{v}} = \nabla F(\boldsymbol{x}_0^k)$ for our setting. Using this concentration inequality, with probability at least $1 - \delta$, we have

$$
\left\| \sum_{j=1}^{i} \boldsymbol{g}_{\sigma_k(j)} \right\| \leq \left\| \sum_{j=1}^{i} \boldsymbol{g}_{\sigma_k(j)} - i\nabla F(\boldsymbol{x}_0^k) \right\| + i \left\| \nabla F(\boldsymbol{x}_0^k) \right\| \leq G \sqrt{8i \log \frac{2}{\delta}} + i \left\| \nabla F(\boldsymbol{x}_0^k) \right\|.
$$

We apply the union bound for all $i = 1, \ldots, n-1$ and $k = 1, \ldots, K$. After this, we have with probability at least $1 - \delta$,

$$
\sum_{i=1}^{n-1} \left\| \sum_{j=1}^{i} \boldsymbol{g}_{\sigma_k(j)} \right\| \leq G \sqrt{8 \log \frac{2nK}{\delta}} \sum_{i=1}^{n-1} \sqrt{i} + \left\| \nabla F(\boldsymbol{x}_0^k) \right\| \sum_{i=1}^{n-1} i
$$

$$\leq G\sqrt{8\log\frac{2nK}{\delta}}\int_1^n \sqrt{y}dy + \frac{n^2}{2}\left\|\nabla F(\boldsymbol{x}_0^k)\right\|$$

$$\leq \frac{4\sqrt{2}n^{3/2}G}{3}\sqrt{\log\frac{2nK}{\delta}} + \frac{n^2}{2}\left\|\nabla F(\boldsymbol{x}_0^k)\right\|, \tag{A.3}$$

for each $k \in [K]$. This then leads to

$$\|\boldsymbol{r}_k\| \leq e^{1/5}L\sum_{i=1}^{n-1}\left\|\sum_{j=1}^i \boldsymbol{g}_{\sigma_k(j)}\right\| \leq \frac{4\sqrt{2}e^{1/5}n^{3/2}LG}{3}\sqrt{\log\frac{2nK}{\delta}} + \frac{e^{1/5}n^2 L}{2}\left\|\nabla F(\boldsymbol{x}_0^k)\right\|$$

$$\leq \frac{5n^{3/2}LG}{2}\sqrt{\log\frac{2nK}{\delta}} + \frac{2n^2 L}{3}\left\|\nabla F(\boldsymbol{x}_0^k)\right\|, \tag{A.4}$$

which holds with probability at least $1 - \delta$. By $(a+b)^2 \leq 2a^2 + 2b^2$, we also have

$$\|\boldsymbol{r}_k\|^2 \leq \frac{25n^3 L^2 G^2}{2}\log\frac{2nK}{\delta} + \frac{8n^4 L^2}{9}\left\|\nabla F(\boldsymbol{x}_0^k)\right\|^2. \tag{A.5}$$

### A.4 The 1st case: getting a per-epoch progress bound

Substituting the norm bounds (A.4) and (A.5) to (A.1) and arranging the terms, we get

$$F(\boldsymbol{x}_0^{k+1}) - F(\boldsymbol{x}_0^k) \leq \left(-\eta n + \eta^2 n^2 L + \frac{2\eta^2 n^2 L}{3} + \frac{8\eta^4 n^4 L^3}{9}\right)\left\|\nabla F(\boldsymbol{x}_0^k)\right\|^2$$

$$+ \frac{5\eta^2 n^{3/2}LG}{2}\left\|\nabla F(\boldsymbol{x}_0^k)\right\|\sqrt{\log\frac{2nK}{\delta}} + \frac{25\eta^4 n^3 L^3 G^2}{2}\log\frac{2nK}{\delta}.$$

Using $ab \leq \frac{a^2}{2} + \frac{b^2}{2}$, we can further decompose

$$\frac{5\eta^2 n^{3/2}LG}{2}\left\|\nabla F(\boldsymbol{x}_0^k)\right\|\sqrt{\log\frac{2nK}{\delta}} = \left(\frac{\eta^{1/2}n^{1/2}}{2}\left\|\nabla F(\boldsymbol{x}_0^k)\right\|\right)\left(5\eta^{3/2}nLG\sqrt{\log\frac{2nK}{\delta}}\right)$$

$$\leq \frac{\eta n}{8}\left\|\nabla F(\boldsymbol{x}_0^k)\right\|^2 + \frac{25\eta^3 n^2 L^2 G^2}{2}\log\frac{2nK}{\delta}.$$

Substituting these results back to the above bound and using $1 + \eta nL \leq 6/5$ yields

$$F(\boldsymbol{x}_0^{k+1}) - F(\boldsymbol{x}_0^k) \leq \left(-\frac{7\eta n}{8} + \frac{5\eta^2 n^2 L}{3} + \frac{8\eta^4 n^4 L^3}{9}\right)\left\|\nabla F(\boldsymbol{x}_0^k)\right\|^2 + 15\eta^3 n^2 L^2 G^2\log\frac{2nK}{\delta}.$$

Now, since $\eta nL \leq 1/5$, we have

$$-\frac{7\eta n}{8} + \frac{5\eta^2 n^2 L}{3} + \frac{8\eta^4 n^4 L^3}{9} \leq -\frac{\eta n}{2},$$

which follows since $z \mapsto \frac{3}{8}z - \frac{5}{3}z^2 - \frac{8}{9}z^4$ is nonnegative when $0 \leq z \leq 1/5$. Therefore, we have

$$F(\boldsymbol{x}_0^{k+1}) - F(\boldsymbol{x}_0^k) \leq -\frac{\eta n}{2}\left\|\nabla F(\boldsymbol{x}_0^k)\right\|^2 + 15\eta^3 n^2 L^2 G^2\log\frac{2nK}{\delta}.$$

Now let us apply the $\mu$-PŁ inequality on $\left\|\nabla F(\boldsymbol{x}_0^k)\right\|^2$. This yields

$$F(\boldsymbol{x}_0^{k+1}) - F^* \leq (1 - \eta n\mu)(F(\boldsymbol{x}_0^k) - F^*) + 15\eta^3 n^2 L^2 G^2\log\frac{2nK}{\delta}.$$

Recursively applying this inequality over $k = 1, \ldots, K$ and substituting $\eta = \frac{2\log(n^{1/2}K)}{\mu nK}$ give[1]

$$F(\boldsymbol{x}_0^{K+1}) - F^* \leq (1 - \eta n\mu)^K(F(\boldsymbol{x}_0) - F^*) + 15\eta^3 n^2 L^2 G^2\log\frac{2nK}{\delta}\sum_{k=0}^{K-1}(1 - \eta n\mu)^k$$

$$\leq \frac{F(\boldsymbol{x}_0) - F^*}{nK^2} + \frac{15\eta^2 nL^2 G^2}{\mu} \log \frac{2nK}{\delta}$$

$$= \frac{F(\boldsymbol{x}_0) - F^*}{nK^2} + \mathcal{O}\left( \frac{L^2 G^2}{\mu^3} \frac{\log^2(n^{1/2}K) \log \frac{nK}{\delta}}{nK^2} \right).$$

Note that this bound certainly holds for the best iterate.

## A.5 The 2nd case: escape implies desired best iterate suboptimality

Now consider the case where some end-of-epoch iterates $\boldsymbol{x}_0^k$ escape the $F(\boldsymbol{x}_0)$-sublevel set $\mathcal{S}_{\boldsymbol{x}_0}$. First, note that by definition of sublevel sets, if $F(\boldsymbol{x}_0^k)$ is monotonically decreasing with $k$, then there is no way $\boldsymbol{x}_0^k$ can escape $\mathcal{S}_{\boldsymbol{x}_0}$. Thus, $\boldsymbol{x}_0^k$ escaping $\mathcal{S}_{\boldsymbol{x}_0}$ implies that $F(\boldsymbol{x}_0^k)$ is not monotonically decreasing. Let $k' \in [2 : K+1]$ be the first $k$ such that $\boldsymbol{x}_0^{k'} \notin \mathcal{S}_{\boldsymbol{x}_0}$. This means that for the previous epoch $k' - 1$, we must have

$$-\eta n \mu (F(\boldsymbol{x}_0^{k'-1}) - F^*) + 15\eta^3 n^2 L^2 G^2 \log \frac{2nK}{\delta} > 0 \tag{A.6}$$

because otherwise

$$F(\boldsymbol{x}_0^{k'}) - F^* \leq (1 - \eta n \mu)(F(\boldsymbol{x}_0^{k'-1}) - F^*) + 15\eta^3 n^2 L^2 G^2 \log \frac{2nK}{\delta} \leq F(\boldsymbol{x}_0^{k'-1}) - F^*,$$

which means $\boldsymbol{x}_0^{k'} \in \mathcal{S}_{\boldsymbol{x}_0}$. Then, from (A.6), we get

$$\min_{k \in [K+1]} F(\boldsymbol{x}_0^k) - F^* \leq F(\boldsymbol{x}_0^{k'-1}) - F^*$$

$$< \frac{15\eta^2 nL^2 G^2}{\mu} \log \frac{2nK}{\delta} = \mathcal{O}\left( \frac{L^2 G^2}{\mu^3} \frac{\log^2(n^{1/2}K) \log \frac{nK}{\delta}}{nK^2} \right).$$

## A.6 Proof of Corollary 1

Let $E$ be the event that the bound (A.3) holds for all $k \in [K]$, which happens with probability at least $1 - \delta$. The high probability result (Theorem 1) showed that given this event happens, we have

$$\min_{k \in [K+1]} F(\boldsymbol{x}_0^k) - F^* \leq \frac{F(\boldsymbol{x}_0) - F^*}{nK^2} + \mathcal{O}\left( \frac{L^2 G^2}{\mu^3} \frac{\log^2(n^{1/2}K) \log \frac{nK}{\delta}}{nK^2} \right).$$

We now choose $\delta = 1/n$. Given the event $E^c$, we will get a similar bound, worse by a factor of $n$:

$$\min_{k \in [K+1]} F(\boldsymbol{x}_0^k) - F^* \leq \frac{F(\boldsymbol{x}_0) - F^*}{nK^2} + \mathcal{O}\left( \frac{L^2 G^2}{\mu^3} \frac{\log^2(nK)}{K^2} \right),$$

without using the concentration inequality. Taking expectation gives

$$\mathbb{E}\left[ \min_{k \in [K+1]} F(\boldsymbol{x}_0^k) - F^* \right]$$

$$= \mathbb{E}\left[ \min_{k \in [K+1]} F(\boldsymbol{x}_0^k) - F^* \mid E \right] \mathbb{P}[E] + \mathbb{E}\left[ \min_{k \in [K+1]} F(\boldsymbol{x}_0^k) - F^* \mid E^c \right] \mathbb{P}[E^c]$$

$$\leq \frac{3(F(\boldsymbol{x}_0) - F^*)}{2nK^2} + \mathcal{O}\left( \frac{L^2 G^2}{\mu^3} \frac{\log^3(nK)}{nK^2} \right),$$

as desired. The rest of the proof derives the bound for $E^c$.

**The first case.** The proof goes the same way as in $E$. We first consider the case where all the iterates stay in $\mathcal{S}_{\boldsymbol{x}_0}$, which corresponds to the first case in the proof of Theorem 1. We unroll the updates $\boldsymbol{x}_i^k$ and obtain the bound (A.1). Then, we bound $\|\boldsymbol{r}_k\|$ directly, without the concentration inequality. From (A.2), we have

$$\|\boldsymbol{r}_k\| \leq \eta L (1 + \eta L)^n \sum_{i=1}^{n-1} \left\| \sum_{j=1}^{i} \boldsymbol{g}_{\sigma_k(j)} \right\| \leq \frac{e^{1/5} \eta n^2 LG}{2} \leq \eta n^2 LG.$$

Substituting this bound to (A.1), we get

$$
\begin{aligned}
F(\boldsymbol{x}_0^{k+1}) - F(\boldsymbol{x}_0^k) &\le (-\eta n + \eta^2 n^2 L) \left\| \nabla F(\boldsymbol{x}_0^k) \right\|^2 + \eta \left\| \nabla F(\boldsymbol{x}_0^k) \right\| \left\| \boldsymbol{r}_k \right\| + L\eta^2 \left\| \boldsymbol{r}_k \right\|^2 \\
&\le (-\eta n + \eta^2 n^2 L) \left\| \nabla F(\boldsymbol{x}_0^k) \right\|^2 + \eta^2 n^2 L G \left\| \nabla F(\boldsymbol{x}_0^k) \right\| + \eta^4 n^4 L^3 G^2 \\
&\le \left( -\frac{7\eta n}{8} + \eta^2 n^2 L \right) \left\| \nabla F(\boldsymbol{x}_0^k) \right\|^2 + 2\eta^3 n^3 L^2 G^2 + \eta^4 n^4 L^3 G^2,
\end{aligned}
$$

where the last inequality used $ab \le \frac{a^2}{2} + \frac{b^2}{2}$ for $a = \frac{\eta^{1/2} n^{1/2}}{2} \left\| \nabla F(\boldsymbol{x}_0^k) \right\|$ and $b = 2\eta^{3/2} n^{3/2} L G$. Now, since $\eta n L \le 1/5$, we have

$$
-\frac{7\eta n}{8} + \eta^2 n^2 L \le -\frac{\eta n}{2},
$$

because $z \mapsto \frac{3}{8} z - z^2$ is nonnegative when $0 \le z \le 1/5$. In conclusion, we have

$$
F(\boldsymbol{x}_0^{k+1}) - F(\boldsymbol{x}_0^k) \le -\frac{\eta n}{2} \left\| \nabla F(\boldsymbol{x}_0^k) \right\|^2 + 3\eta^3 n^3 L^2 G^2.
$$

Applying the $\mu$-PŁ inequality, we have

$$
F(\boldsymbol{x}_0^{k+1}) - F^* \le (1 - \eta n \mu)(F(\boldsymbol{x}_0^k) - F^*) + 3\eta^3 n^3 L^2 G^2.
$$

Unrolling the inequalities and substituting $\eta = \frac{2 \log(n^{1/2} K)}{\mu n K}$, we get

$$
\begin{aligned}
\min_{k \in [K+1]} F(\boldsymbol{x}_0^k) - F^* &\le F(\boldsymbol{x}_0^{K+1}) - F^* \\
&\le \frac{F(\boldsymbol{x}_0) - F^*}{nK^2} + \frac{3\eta^2 n^2 L^2 G^2}{\mu} \le \frac{F(\boldsymbol{x}_0) - F^*}{nK^2} + \mathcal{O}\left( \frac{L^2 G^2}{\mu^3} \frac{\log^2(nK)}{K^2} \right).
\end{aligned}
$$

**The second case.**   Now consider the case where some end-of-epoch iterates satisfy $\boldsymbol{x}_0^k \notin \mathcal{S}_{\boldsymbol{x}_0}$. We can apply the same argument as the second case of Theorem 1 here.

Let $k'$ be the first such index. Then, this means that $F(\boldsymbol{x}_0^{k'})$ is greater than $F(\boldsymbol{x}_0^{k'-1})$, which holds only if

$$
-\eta n \mu (F(\boldsymbol{x}_0^{k'-1}) - F^*) + 3\eta^3 n^2 L^2 G^2 > 0.
$$

Then, this implies that

$$
\min_{k \in [K+1]} F(\boldsymbol{x}_0^k) - F^* \le F(\boldsymbol{x}_0^{k'-1}) - F^* < \frac{3\eta^2 n^2 L^2 G^2}{\mu} = \mathcal{O}\left( \frac{L^2 G^2}{\mu^3} \frac{\log^2(nK)}{K^2} \right).
$$

# B   Analysis on RANDOMSHUFFLE for quadratics (Proof of Theorem 2)

## B.1   Additional notation on matrices

Prior to the proofs, we introduce additional notation on matrices. For a matrix $\boldsymbol{A}$, $\|\boldsymbol{A}\|$ denotes its spectral norm. For matrices indexed $\boldsymbol{M}_1, \boldsymbol{M}_2, \ldots, \boldsymbol{M}_k$ and for any $1 \le i \le j \le k$, we use the shorthand notation for products $\boldsymbol{M}_{j:i} = \boldsymbol{M}_j \boldsymbol{M}_{j-1} \ldots \boldsymbol{M}_{i+1} \boldsymbol{M}_i$. In case where $i > j$, we define $\boldsymbol{M}_{j:i} = \boldsymbol{I}$. Similarly, $\boldsymbol{M}_{j:i}^T$ denotes the product $\boldsymbol{M}_i^T \boldsymbol{M}_{i+1}^T \ldots \boldsymbol{M}_{j-1}^T \boldsymbol{M}_j^T$.

The proofs of Theorems 2 and C.1 involve polynomials of matrices. We define the following noncommutative elementary symmetric polynomials, which will prove useful in the proof. For a permutation $\sigma : [n] \to [n]$ and integers $l, r$ and $m$ satisfying $1 \le l \le r \le n$ and $m \in [0 : n]$,

$$
e_m(\boldsymbol{A}_1, \ldots, \boldsymbol{A}_n; \sigma, l, r) := \sum_{l \le t_1 < t_2 < \cdots < t_m \le r} \boldsymbol{A}_{\sigma(t_m)} \boldsymbol{A}_{\sigma(t_{m-1})} \cdots \boldsymbol{A}_{\sigma(t_1)}. \tag{B.1}
$$

Whenever it is clear from the context that the arguments are $\boldsymbol{A}_1, \ldots, \boldsymbol{A}_n$ and permutation is $\sigma$, we use a shorthand $\boldsymbol{A}_{\sigma[n]}$. Also, the default value of $l$ and $r$ are $l = 1$ and $r = n$; so, $e_m(\boldsymbol{A}_{\sigma[n]}) := e_m(\boldsymbol{A}_1, \ldots, \boldsymbol{A}_n; \sigma, 1, n)$.

## B.2 Proof outline

Recall the definitions

$$f_i(\boldsymbol{x}) := \tfrac{1}{2}\boldsymbol{x}^T \boldsymbol{A}_i \boldsymbol{x} + \boldsymbol{b}_i^T \boldsymbol{x}, \ F(\boldsymbol{x}) := \frac{1}{n}\sum_{i=1}^n f_i(\boldsymbol{x}) = \tfrac{1}{2}\boldsymbol{x}^T \boldsymbol{A}\boldsymbol{x},$$

where $f_i$'s are $L$-smooth and $F$ is $\mu$ strongly convex. This is equivalent to saying that $\|\boldsymbol{A}_i\| \leq L$ and $\boldsymbol{A} := \frac{1}{n}\sum_{i=1}^n \boldsymbol{A}_i \succeq \mu \boldsymbol{I}$. Also note that $F$ is minimized at $\boldsymbol{x}^* = \boldsymbol{0}$ and $\sum_{i=1}^n \boldsymbol{b}_i = \boldsymbol{0}$. We let $G := \max_{i\in[n]} \|\boldsymbol{b}_i\|$.

The proof goes as follows. We first recursively apply the update equations over all iterations and obtain an equation that expresses the last iterate $\boldsymbol{x}_0^{K+1}$ in terms of the initialization $\boldsymbol{x}_0^1 = \boldsymbol{x}_0$. Using such an equation, we will directly bound $\mathbb{E}[\|\boldsymbol{x}_0^{K+1} - \boldsymbol{x}^*\|^2] = \mathbb{E}[\|\boldsymbol{x}_0^{K+1} - \boldsymbol{0}\|^2]$ to get our desired result.

Compute the update equation of $\boldsymbol{x}_1^k$ in terms of the initial iterate $\boldsymbol{x}_0^k$ of the $k$-th epoch:

$$\begin{aligned} \boldsymbol{x}_1^k &= \boldsymbol{x}_0^k - \eta \nabla f_{\sigma_k(1)}(\boldsymbol{x}_0^k) = \boldsymbol{x}_0^k - \eta(\boldsymbol{A}_{\sigma_k(1)}\boldsymbol{x}_0^k + \boldsymbol{b}_{\sigma_k(1)}) \\ &= (\boldsymbol{I} - \eta \boldsymbol{A}_{\sigma_k(1)})\boldsymbol{x}_0^k - \eta \boldsymbol{b}_{\sigma_k(1)}. \end{aligned}$$

Substituting this to the update equation of $\boldsymbol{x}_2^k$, we get

$$\begin{aligned} \boldsymbol{x}_2^k &= \boldsymbol{x}_1^k - \eta \nabla f_{\sigma_k(2)}(\boldsymbol{x}_1^k) \\ &= (\boldsymbol{I} - \eta \boldsymbol{A}_{\sigma_k(1)})\boldsymbol{x}_0^k - \eta \boldsymbol{b}_{\sigma_k(1)} - \eta(\boldsymbol{A}_{\sigma_k(2)}((\boldsymbol{I} - \eta \boldsymbol{A}_{\sigma_k(1)})\boldsymbol{x}_0^k - \eta \boldsymbol{b}_{\sigma_k(1)}) + \boldsymbol{b}_{\sigma_k(2)}) \\ &= (\boldsymbol{I} - \eta \boldsymbol{A}_{\sigma_k(2)})(\boldsymbol{I} - \eta \boldsymbol{A}_{\sigma_k(1)})\boldsymbol{x}_0^k - \eta \boldsymbol{b}_{\sigma_k(2)} - \eta(\boldsymbol{I} - \eta \boldsymbol{A}_{\sigma_k(2)})\boldsymbol{b}_{\sigma_k(1)}. \end{aligned}$$

Repeating this, one can write the last iterate $\boldsymbol{x}_n^k$ (or equivalently, $\boldsymbol{x}_0^{k+1}$) of the $k$-th epoch as the following:

$$\begin{aligned} \boldsymbol{x}_0^{k+1} &= \underbrace{\left[\prod_{t=n}^1 (\boldsymbol{I} - \eta \boldsymbol{A}_{\sigma_k(t)})\right]}_{=:\boldsymbol{S}_k} \boldsymbol{x}_0^k - \eta \underbrace{\left[\sum_{j=1}^n \left(\prod_{t=n}^{j+1}(\boldsymbol{I} - \eta \boldsymbol{A}_{\sigma_k(t)})\right) \boldsymbol{b}_{\sigma_k(j)}\right]}_{=:\boldsymbol{t}_k} \\ &= \boldsymbol{S}_k \boldsymbol{x}_0^k - \eta \boldsymbol{t}_k. \end{aligned} \tag{B.2}$$

Note that $\boldsymbol{S}_k$ and $\boldsymbol{t}_k$ are random variables that solely depend on the $k$-th permutation $\sigma_k$. Now, repeating this $K$ times, we get the equation for the iterate after $K$ epochs, which is the output of the algorithm we consider in Theorem 2:

$$\boldsymbol{x}_0^{K+1} = \left(\prod_{k=K}^1 \boldsymbol{S}_k\right)\boldsymbol{x}_0^1 - \eta \sum_{k=1}^K \left(\prod_{t=K}^{k+1} \boldsymbol{S}_t\right)\boldsymbol{t}_k = \boldsymbol{S}_{K:1}\boldsymbol{x}_0^1 - \eta \sum_{k=1}^K \boldsymbol{S}_{K:k+1}\boldsymbol{t}_k.$$

We aim to provide an upper bound on $\mathbb{E}[\|\boldsymbol{x}_0^{K+1}\|^2]$, where the expectation is over the randomness of permutation $\sigma_1, \ldots, \sigma_K$. To this end, using $\|\boldsymbol{a} + \boldsymbol{b}\|^2 \leq 2\|\boldsymbol{a}\|^2 + 2\|\boldsymbol{b}\|^2$,

$$\left\|\boldsymbol{x}_0^{K+1}\right\|^2 \leq 2\left\|\boldsymbol{S}_{K:1}\boldsymbol{x}_0^1\right\|^2 + 2\eta^2 \left\|\sum_{k=1}^K \boldsymbol{S}_{K:k+1}\boldsymbol{t}_k\right\|^2$$

where the second term on the RHS can be further decomposed into:

$$\left\|\sum_{k=1}^K \boldsymbol{S}_{K:k+1}\boldsymbol{t}_k\right\|^2 = \sum_{k=1}^K \|\boldsymbol{S}_{K:k+1}\boldsymbol{t}_k\|^2 + 2\sum_{1\leq k<k'\leq K}\langle \boldsymbol{S}_{K:k+1}\boldsymbol{t}_k, \boldsymbol{S}_{K:k'+1}\boldsymbol{t}_{k'}\rangle.$$

The remaining proof bounds each of the terms, which we state as the following three lemmas. The proofs of Lemmas B.1, B.2, and B.3 are deferred to Sections B.3, B.4, and B.5, respectively.

**Lemma B.1** (1st contraction bound). *For any* $0 \leq \eta \leq \frac{3}{16nL} \min\{1, \sqrt{\frac{n}{\kappa}}\}$ *and* $k \in [K]$,

$$\left\| \mathbb{E}\left[\boldsymbol{S}_k^T \boldsymbol{S}_k\right]\right\| \leq 1 - \eta n \mu.$$

**Lemma B.2.** *For any* $0 \leq \eta \leq \frac{3}{16nL} \min\{1, \sqrt{\frac{n}{\kappa}}\}$ *and* $k \in [K]$,

$$\mathbb{E}\left[\left\|\boldsymbol{S}_{K:k+1}\boldsymbol{t}_k\right\|^2\right] \leq 18(1 - \eta n \mu)^{K-k}\eta^2 n^3 L^2 G^2 \log n.$$

**Lemma B.3.** *For any* $0 \leq \eta \leq \frac{3}{16nL} \min\{1, \sqrt{\frac{n}{\kappa}}\}$ *and* $k, k' \in [K]$ ($k < k'$),

$$\mathbb{E}\left[\langle \boldsymbol{S}_{K:k+1}\boldsymbol{t}_k, \boldsymbol{S}_{K:k'+1}\boldsymbol{t}_{k'}\rangle\right] \leq 40\left(1 - \frac{\eta n \mu}{2}\right)^{2K-k'-k-1}\eta^2 n^2 L^2 G^2.$$

**Remark B.1** (Our contraction bounds and the matrix AM-GM inequality conjecture). Before we continue with the proof, a side remark on the contraction bounds is in order. In this paper, we prove a number of contraction bounds (Lemmas B.1, B.4, and C.2) that circumvents the need for the conjectured matrix AM-GM inequality [11], which was proven to be false [5]. The bounds we provide can be seen as "weaker" versions of the AM-GM inequalities, which hold for any number $n$ of matrices but with $\eta$ diminishing with $n$. Whether these weak AM-GM inequalities hold for a broader range of $\eta$ (e.g. $\eta \leq 1/L$) or not is left to future investigation.

By Lemma B.1, we have $\boldsymbol{0} \preceq \mathbb{E}[\boldsymbol{S}_k^T \boldsymbol{S}_k] \preceq (1 - \eta n \mu)\boldsymbol{I}$ for appropriately chosen step size $\eta$. Since any $\boldsymbol{S}_k^T \boldsymbol{S}_k$ is independent of $\sigma_1, \ldots, \sigma_{k-1}$, we have

$$
\begin{aligned}
\mathbb{E}\left[\left\|\boldsymbol{S}_{K:1}\boldsymbol{x}_0^1\right\|^2\right] &= \mathbb{E}\left[\left(\boldsymbol{S}_{K:1}\boldsymbol{x}_0^1\right)^T \left(\boldsymbol{S}_{K:1}\boldsymbol{x}_0^1\right)\right] \\
&= \mathbb{E}\left[\left(\boldsymbol{S}_{K-1:1}\boldsymbol{x}_0^1\right)^T \mathbb{E}\left[\boldsymbol{S}_K^T \boldsymbol{S}_K\right] \left(\boldsymbol{S}_{K-1:1}\boldsymbol{x}_0^1\right)\right] \\
&\leq (1 - \eta n \mu)\mathbb{E}\left[\left(\boldsymbol{S}_{K-1:1}\boldsymbol{x}_0^1\right)^T \left(\boldsymbol{S}_{K-1:1}\boldsymbol{x}_0^1\right)\right] \\
&\leq (1 - \eta n \mu)^2 \mathbb{E}\left[\left(\boldsymbol{S}_{K-2:1}\boldsymbol{x}_0^1\right)^T \left(\boldsymbol{S}_{K-2:1}\boldsymbol{x}_0^1\right)\right] \\
&\leq \cdots \leq (1 - \eta n \mu)^K \left\|\boldsymbol{x}_0^1\right\|^2.
\end{aligned}
$$

By Lemma B.2, we have

$$\sum_{k=1}^K \mathbb{E}\left[\left\|\boldsymbol{S}_{K:k+1}\boldsymbol{t}_k\right\|^2\right] \leq 18\eta^2 n^3 L^2 G^2 \log n \sum_{k=1}^K (1 - \eta n \mu)^{K-k} \leq \frac{18\eta n^2 L^2 G^2 \log n}{\mu},$$

and Lemma B.3 implies that

$$\sum_{1 \leq k < k' \leq K} \mathbb{E}\left[\langle \boldsymbol{S}_{K:k+1}\boldsymbol{t}_k, \boldsymbol{S}_{K:k'+1}\boldsymbol{t}_{k'}\rangle\right] \leq 40\eta^2 n^2 L^2 G^2 \sum_{1 \leq k < k' \leq K}\left(1 - \frac{\eta n \mu}{2}\right)^{2K-k'-k-1} \leq \frac{160 L^2 G^2}{\mu^2}.$$

Putting the bounds together, we get

$$\mathbb{E}[\left\|\boldsymbol{x}_0^{K+1}\right\|^2] \leq 2(1 - \eta n \mu)^K \left\|\boldsymbol{x}_0^1\right\|^2 + \frac{36\eta^3 n^2 L^2 G^2 \log n}{\mu} + \frac{640\eta^2 L^2 G^2}{\mu^2}.$$

Substituting the step size $\eta = \frac{2\log(nK)}{\mu nK}$ into the bound gives

$$\mathbb{E}[\left\|\boldsymbol{x}_0^{K+1}\right\|^2] \leq \frac{2\left\|\boldsymbol{x}_0^1\right\|^2}{n^2 K^2} + \mathcal{O}\left(\frac{L^2 G^2}{\mu^4}\left(\frac{\log^4(nK)}{nK^3} + \frac{\log^2(nK)}{n^2 K^2}\right)\right),$$

and in terms of the cost values,

$$\mathbb{E}[F(\boldsymbol{x}_0^{K+1}) - F^*] \leq \frac{2L\left\|\boldsymbol{x}_0^1\right\|^2}{n^2 K^2} + \mathcal{O}\left(\frac{L^3 G^2}{\mu^4}\left(\frac{\log^4(nK)}{nK^3} + \frac{\log^2(nK)}{n^2 K^2}\right)\right).$$

Recall that these bounds hold for $\eta \leq \frac{3}{16nL} \min\{1, \sqrt{\frac{n}{\kappa}}\}$, so $K$ must be large enough so that

$$\frac{2\log(nK)}{\mu nK} \leq \frac{3}{16nL} \min\left\{1, \sqrt{\frac{n}{\kappa}}\right\}.$$

This gives us the epoch requirement $K \geq \frac{32}{3}\kappa \max\{1, \sqrt{\frac{\kappa}{n}}\}\log(nK)$.

## B.3  Proof of the first contraction bound (Lemma B.1)

### B.3.1  Decomposition into elementary polynomials

For any permutation $\sigma_k$, note that we can expand $\boldsymbol{S}_k$ in the following way:

$$\boldsymbol{S}_k = \prod_{t=n}^{1}(\boldsymbol{I} - \eta\boldsymbol{A}_{\sigma_k(t)}) = \sum_{m=0}^{n}(-\eta)^m \sum_{1 \le t_1 < \cdots < t_m \le n} \boldsymbol{A}_{\sigma_k(t_m)}\cdots\boldsymbol{A}_{\sigma_k(t_1)} =: \sum_{m=0}^{n}(-\eta)^m e_m(\boldsymbol{A}_{\sigma_k[n]}),$$

where the noncommutative elementary symmetric polynomial $e_m$ was defined in (B.1). Using this, we can write

$$\boldsymbol{S}_k^T \boldsymbol{S}_k = \sum_{m=0}^{2n}(-\eta)^m \underbrace{\sum_{\substack{0 \le m_1 \le n \\ 0 \le m_2 \le n \\ m_1 + m_2 = m}} e_{m_1}(\boldsymbol{A}_{\sigma_k[n]})^T e_{m_2}(\boldsymbol{A}_{\sigma_k[n]})}_{=:\boldsymbol{C}_m}. \tag{B.3}$$

Note $\mathbb{E}[\boldsymbol{S}_k^T \boldsymbol{S}_k] = \sum_{m=0}^{2n}(-\eta)^m\mathbb{E}[\boldsymbol{C}_m]$. In what follows, we will examine the expectation $\mathbb{E}[\boldsymbol{C}_m]$ closely, and decompose $\mathbb{E}[\boldsymbol{S}_k^T \boldsymbol{S}_k]$ into the sum of $\sum_{m=0}^{n}\frac{(-2\eta n\boldsymbol{A})^m}{m!}$ and remainder terms. By bounding the spectral norm of $\sum_{m=0}^{n}\frac{(-2\eta n\boldsymbol{A})^m}{m!}$ and the remainder terms, we will get the desired bound on the spectral norm of $\mathbb{E}[\boldsymbol{S}_k^T \boldsymbol{S}_k]$.

**Cases $0 \le m \le 2$.** It is easy to check that $\boldsymbol{C}_0 = \boldsymbol{I}$ and $\boldsymbol{C}_1 = 2e_1(\boldsymbol{A}_{\sigma_k[n]}) = 2\sum_{i=1}^{n}\boldsymbol{A}_i = 2n\boldsymbol{A}$, regardless of $\sigma_k$. For $\boldsymbol{C}_2$, we have

$$\boldsymbol{C}_2 = e_2(\boldsymbol{A}_{\sigma_k[n]})^T e_0(\boldsymbol{A}_{\sigma_k[n]}) + e_1(\boldsymbol{A}_{\sigma_k[n]})^T e_1(\boldsymbol{A}_{\sigma_k[n]}) + e_0(\boldsymbol{A}_{\sigma_k[n]})^T e_2(\boldsymbol{A}_{\sigma_k[n]})$$

$$= \sum_{1 \le t_1 < t_2 \le n}\boldsymbol{A}_{\sigma_k(t_1)}\boldsymbol{A}_{\sigma_k(t_2)} + \left(\sum_{i=1}^{n}\boldsymbol{A}_i\right)^2 + \sum_{1 \le t_1 < t_2 \le n}\boldsymbol{A}_{\sigma_k(t_2)}\boldsymbol{A}_{\sigma_k(t_1)}$$

$$= \sum_{i \ne j}\boldsymbol{A}_i\boldsymbol{A}_j + \left(\sum_{i=1}^{n}\boldsymbol{A}_i\right)^2 = 2\left(\sum_{i=1}^{n}\boldsymbol{A}_i\right)^2 - \sum_{i=1}^{n}\boldsymbol{A}_i^2 = 2(n\boldsymbol{A})^2 - \sum_{i=1}^{n}\boldsymbol{A}_i^2,$$

again regardless of $\sigma_k$. Note that each $\boldsymbol{A}_i^2$ is positive semidefinite even when $\boldsymbol{A}_i$ is not.

**Cases $3 \le m \le n$: decomposition of $\boldsymbol{C}_m$.** In a similar way, for $m = 3, \ldots, n$, we will take expectation $\mathbb{E}[\boldsymbol{C}_m]$ and express it as the sum of $\frac{(2n\boldsymbol{A})^m}{m!}$ and the remainder terms. Now fix any $m \in [3 : n]$, and consider any $m_1$ and $m_2$ satisfying $m_1 + m_2 = m$. Then, the product of elementary polynomials $e_{m_1}(\boldsymbol{A}_{\sigma_k[n]})^T e_{m_2}(\boldsymbol{A}_{\sigma_k[n]})$ consists of $\binom{n}{m_1}\binom{n}{m_2}$ terms of the following form:

$$\prod_{i=1}^{m_1}\boldsymbol{A}_{\sigma_k(s_i)}\prod_{i=m_2}^{1}\boldsymbol{A}_{\sigma_k(t_i)}, \text{ where } 1 \le s_1 < \cdots < s_{m_1} \le n, \ 1 \le t_1 < \cdots < t_{m_2} \le n. \tag{B.4}$$

Among them, $\binom{n}{m_1}\binom{n-m_1}{m_2}$ terms have the property that each of the $s_1, \ldots, s_{m_1}$ and $t_1 \ldots, t_{m_2}$ is unique; in other words, $\{s_1, \ldots, s_{m_1}\} \cap \{t_1, \ldots, t_{m_2}\} = \emptyset$. The remaining $\binom{n}{m_1}(\binom{n}{m_2} - \binom{n-m_1}{m_2})$ terms have overlapping indices.

Using this observation, we decompose $\boldsymbol{C}_m$ into two terms $\boldsymbol{C}_m = \boldsymbol{D}_m + \boldsymbol{R}_m$. Here, $\boldsymbol{D}_m$ is a sum of terms in $\boldsymbol{C}_m$ with distinct indices $s_1, \ldots, s_{m_1}, t_1, \ldots, t_{m_2}$ and $\boldsymbol{R}_m$ is the sum of the remaining terms.

$$\boldsymbol{D}_m := \sum_{\substack{0 \le m_1 \le n \\ 0 \le m_2 \le n \\ m_1 + m_2 = m}} \sum_{\substack{1 \le s_1 < \cdots < s_{m_1} \le n \\ 1 \le t_1 < \cdots < t_{m_2} \le n \\ s_i, t_i \text{ unique}}} \mathbb{E}\left[\prod_{i=1}^{m_1}\boldsymbol{A}_{\sigma_k(s_i)}\prod_{i=m_2}^{1}\boldsymbol{A}_{\sigma_k(t_i)}\right], \quad \boldsymbol{R}_m := \boldsymbol{C}_m - \boldsymbol{D}_m. \tag{B.5}$$

The matrix $\boldsymbol{C}_m$ is a summation of

$$\sum_{\substack{0 \leq m_1 \leq n \\ 0 \leq m_2 \leq n \\ m_1 + m_2 = m}} \binom{n}{m_1}\binom{n}{m_2} = \binom{2n}{m}$$

terms of the form in (B.4). The number of terms in $\boldsymbol{D}_m$ is

$$\sum_{\substack{0 \leq m_1 \leq n \\ 0 \leq m_2 \leq n \\ m_1 + m_2 = m}} \binom{n}{m_1}\binom{n - m_1}{m_2} = 2^m \binom{n}{m},$$

and consequently, $\boldsymbol{R}_m$ consists of $\binom{2n}{m} - 2^m \binom{n}{m}$ terms.

**Cases** $3 \leq m \leq n$**: expectation of terms in** $\boldsymbol{D}_m$**.** For any $s_1, s_2, \ldots, s_{m_1}, t_1, t_2, \ldots, t_{m_2}$ such that each of $s_i$ or $t_i$ is unique, we have

$$\mathbb{E}\left[\prod_{i=1}^{m_1} \boldsymbol{A}_{\sigma_k(s_i)} \prod_{i=m_2}^{1} \boldsymbol{A}_{\sigma_k(t_i)}\right] = \mathbb{E}\left[\prod_{i=1}^{m} \boldsymbol{A}_{\sigma_k(i)}\right],$$

due to taking expectation. We can expand this expectation using the law of total expectation.

$$\mathbb{E}\left[\prod_{i=1}^{m} \boldsymbol{A}_{\sigma_k(i)}\right]$$

$$= \sum_{j_1 \in [n]} \boldsymbol{A}_{j_1} \mathbb{E}\left[\prod_{i=2}^{m} \boldsymbol{A}_{\sigma_k(i)} \mid \sigma_k(1) = j_1\right] \mathbb{P}[\sigma_k(1) = j_1]$$

$$= \frac{1}{n} \sum_{j_1 \in [n]} \boldsymbol{A}_{j_1} \mathbb{E}\left[\prod_{i=2}^{m} \boldsymbol{A}_{\sigma_k(i)} \mid \sigma_k(1) = j_1\right]$$

$$= \frac{1}{n(n-1)} \sum_{j_1 \in [n]} \sum_{j_2 \in [n] \setminus \{j_1\}} \boldsymbol{A}_{j_1} \boldsymbol{A}_{j_2} \mathbb{E}\left[\prod_{i=3}^{m} \boldsymbol{A}_{\sigma_k(i)} \mid \sigma_k(1) = j_1, \sigma_k(2) = j_2\right]$$

$$= \cdots = \frac{(n-m)!}{n!} \sum_{j_1 \in [n]} \sum_{j_2 \in [n] \setminus \{j_1\}} \cdots \sum_{j_m \in [n] \setminus \{j_1, \ldots, j_{m-1}\}} \prod_{i=1}^{m} \boldsymbol{A}_{j_i}$$

$$= \frac{(n-m)!}{n!} \sum_{\substack{j_1, \ldots, j_m \in [n] \\ j_1, \ldots, j_m \text{ unique}}} \prod_{i=1}^{m} \boldsymbol{A}_{j_i}$$

$$= \frac{(n-m)!}{n!} \left(\sum_{i=1}^{n} \boldsymbol{A}_i\right)^m - \frac{(n-m)!}{n!} \underbrace{\sum_{\substack{j_1, \ldots, j_m \in [n] \\ j_1, \ldots, j_m \text{ not unique}}} \prod_{i=1}^{m} \boldsymbol{A}_{j_i}}_{=:\boldsymbol{N}_m}. \tag{B.6}$$

Here, we decompose the expectation of $\prod_{i=1}^{m} \boldsymbol{A}_{\sigma_k(i)}$ into the difference of $(n\boldsymbol{A})^m$ and $\boldsymbol{N}_m$. Note that all $2^m \binom{n}{m}$ terms in $\boldsymbol{D}_m$ have the same expectation, identical to the one evaluated above. Also note that $\boldsymbol{N}_m$ is a sum of $n^m - \frac{n!}{(n-m)!}$ terms.

To summarize, we have decomposed the expectation of $\boldsymbol{C}_m$ twice, in the following way:

$$\mathbb{E}[\boldsymbol{C}_m] = \mathbb{E}[\boldsymbol{D}_m] + \mathbb{E}[\boldsymbol{R}_m]$$

$$= 2^m \binom{n}{m} \frac{(n-m)!}{n!} \left((n\boldsymbol{A})^m - \boldsymbol{N}_m\right) + \mathbb{E}[\boldsymbol{R}_m]$$

$$= \frac{(2n\boldsymbol{A})^m}{m!} - \frac{2^m}{m!} \boldsymbol{N}_m + \mathbb{E}[\boldsymbol{R}_m].$$

**Spectral norm bound.** Up to this point, we obtained the following equations for $\boldsymbol{C}_m$'s:

$$\boldsymbol{C}_0 = \boldsymbol{I},$$
$$\boldsymbol{C}_1 = 2n\boldsymbol{A},$$
$$\boldsymbol{C}_2 = 2(n\boldsymbol{A})^2 - \sum_{i=1}^n \boldsymbol{A}_i^2,$$
$$\mathbb{E}[\boldsymbol{C}_m] = \frac{(2n\boldsymbol{A})^m}{m!} - \frac{2^m}{m!}\boldsymbol{N}_m + \mathbb{E}[\boldsymbol{R}_m], \text{ for } m = 3, \ldots, n.$$

We substitute these to $\mathbb{E}[\boldsymbol{S}_k^T \boldsymbol{S}_k] = \sum_{m=0}^{2n}(-\eta)^m \mathbb{E}[\boldsymbol{C}_m]$ and get

$$\mathbb{E}[\boldsymbol{S}_k^T \boldsymbol{S}_k] = \sum_{m=0}^n \frac{(-2\eta n\boldsymbol{A})^m}{m!} - \eta^2 \sum_{i=1}^n \boldsymbol{A}_i^2 + \sum_{m=3}^n (-\eta)^m \left(\mathbb{E}[\boldsymbol{R}_m] - \frac{2^m}{m!}\boldsymbol{N}_m\right)$$
$$+ \sum_{m=n+1}^{2n} (-\eta)^m \mathbb{E}[\boldsymbol{C}_m], \tag{B.7}$$

and consequently,

$$\left\|\mathbb{E}[\boldsymbol{S}_k^T \boldsymbol{S}_k]\right\| \leq \left\|\sum_{m=0}^n \frac{(-2\eta n\boldsymbol{A})^m}{m!}\right\| + \sum_{m=3}^n \eta^m \left(\|\mathbb{E}[\boldsymbol{R}_m]\| + \frac{2^m}{m!}\|\boldsymbol{N}_m\|\right)$$
$$+ \sum_{m=n+1}^{2n} \eta^m \|\mathbb{E}[\boldsymbol{C}_m]\|.$$

In what follows, we will bound each of the norms to get an upper bound.

### B.3.2 Bounding each term of the spectral norm bound

We first start with $\left\|\sum_{m=0}^n \frac{(-2\eta n\boldsymbol{A})^m}{m!}\right\|$. Note that for any eigenvalue $s$ of the positive definite matrix $\boldsymbol{A}$, the corresponding eigenvalue of $\sum_{m=0}^n \frac{(-2\eta n\boldsymbol{A})^m}{m!}$ is $\sum_{m=0}^n \frac{(-2\eta n s)^m}{m!}$. Recall $\eta \leq \frac{3}{16nL}\min\{1, \sqrt{\frac{n}{\kappa}}\} \leq \frac{1}{4nL}$, so $0 \leq 2\eta n s \leq 1/2$ for any eigenvalue $s$ of $\boldsymbol{A}$. Since $t \mapsto \sum_{m=0}^n \frac{(-t)^m}{m!}$ is a positive and decreasing function on $[0, 0.5]$ for any $n \geq 2$, the matrix $\sum_{m=0}^n \frac{(-2\eta n\boldsymbol{A})^m}{m!}$ is positive definite and its maximum singular value (i.e., spectral norm) comes from the minimum eigenvalue of $\boldsymbol{A}$, hence

$$\left\|\sum_{m=0}^n \frac{(-2\eta n\boldsymbol{A})^m}{m!}\right\| \leq \sum_{m=0}^n \frac{(-2\eta n\mu)^m}{m!}.$$

As for $\|\mathbb{E}[\boldsymbol{R}_m]\|$, where $m = 3, \ldots, n$, recall that $\boldsymbol{R}_m$ is a sum of $\binom{2n}{m} - 2^m \binom{n}{m}$ terms, and each of the terms has spectral norm bounded above by $L^m$. Thus,

$$\|\mathbb{E}[\boldsymbol{R}_m]\| \leq \mathbb{E}[\|\boldsymbol{R}_m\|] \leq \left(\binom{2n}{m} - 2^m \binom{n}{m}\right) L^m \leq (2n)^{m-1} L^m, \tag{B.8}$$

due to Lemma B.7. Similarly, $\boldsymbol{N}_m$ is a sum of $n^m - \frac{n!}{(n-m)!}$ elements, so using the same lemma,

$$\frac{2^m}{m!}\|\boldsymbol{N}_m\| \leq \frac{2^m}{m!}\left(n^m - \frac{n!}{(n-m)!}\right) L^m = \left(\frac{(2n)^m}{m!} - 2^m \binom{n}{m}\right) L^m \leq (2n)^{m-1} L^m. \tag{B.9}$$

Finally, we consider $\|\mathbb{E}[\boldsymbol{C}_m]\|$ for $m = n+1, \ldots, 2n$. It contains $\binom{2n}{m} = \binom{2n}{2n-m}$ terms, and each of the terms have spectral norm bounded above by $L^m$. This leads to

$$\|\mathbb{E}[\boldsymbol{C}_m]\| \leq \binom{2n}{2n-m} L^m \leq (2n)^{2n-m} L^m \leq (2n)^{m-1} L^m, \tag{B.10}$$

where the last bound used $2n - m \leq m - 1$, which holds for $m = n+1, \ldots, 2n$.

### B.3.3 Concluding the proof

Putting the bounds together, we get

$$\left\|\mathbb{E}[\boldsymbol{S}_k^T \boldsymbol{S}_k]\right\| \leq \sum_{m=0}^{n} \frac{(-2\eta n\mu)^m}{m!} + 2\sum_{m=3}^{n} \eta^m (2n)^{m-1} L^m + \sum_{m=n+1}^{2n} \eta^m (2n)^{m-1} L^m$$

$$\leq \sum_{m=0}^{2} \frac{(-2\eta n\mu)^m}{m!} + \frac{1}{n}\sum_{m=3}^{2n} (2\eta n L)^m$$

$$\leq \sum_{m=0}^{2} \frac{(-2\eta n\mu)^m}{m!} + \frac{1}{n}\frac{(2\eta n L)^3}{1 - 2\eta n L}$$

$$\leq 1 - 2\eta n\mu + \frac{1}{2}(2\eta n\mu)^2 + \frac{2}{n}(2\eta n L)^3.$$

Here, we used $2\eta n L \leq 1/2$, and the fact that $1 - t + \frac{t^2}{2} \geq \sum_{m=0}^{n} \frac{(-t)^m}{m!}$ for all $t \in [0, 0.5]$ and $n \geq 2$. The remaining step is to show that the right hand side of the inequality is bounded above by $1 - \eta n\mu$ for $0 \leq \eta \leq \frac{3}{16nL}\min\{1, \sqrt{\frac{n}{\kappa}}\}$.

Define $z = 2\eta n L$. Using this, we have

$$1 - 2\eta n\mu + \frac{1}{2}(2\eta n\mu)^2 + \frac{2}{n}(2\eta n L)^3 \leq 1 - \eta n\mu \text{ for } 0 \leq \eta \leq \frac{3}{16nL}\min\left\{1, \sqrt{\frac{n}{\kappa}}\right\}$$

$$\Leftrightarrow g(z) := \frac{z}{2\kappa} - \frac{z^2}{2\kappa^2} - \frac{2z^3}{n} \geq 0 \text{ for } 0 \leq z \leq \frac{3}{8}\min\left\{1, \sqrt{\frac{n}{\kappa}}\right\},$$

so it suffices to show the latter. One can check that $g(0) = 0$, $g'(0) > 0$ and $g'(z)$ is monotonically decreasing in $z \geq 0$, so $g(z) \geq 0$ holds for $z \in [0, c]$ for some $c > 0$. This also means that if we have $g(c) \geq 0$ for some $c > 0$, $g(z) \geq 0$ for all $z \in [0, c]$.

First, consider the case $\kappa \leq n$. Then, $n/\kappa \geq 1$ and $\kappa \geq 1$, so

$$\frac{z}{2\kappa} - \frac{z^2}{2\kappa^2} - \frac{2z^3}{n} = \frac{1}{2\kappa}\left(z - \frac{z^2}{\kappa} - \frac{4z^3}{n/\kappa}\right) \geq \frac{1}{2\kappa}\left(z - z^2 - 4z^3\right).$$

We can check that the function $z \mapsto z - z^2 - 4z^3$ is strictly positive at $z = \frac{3}{8}$. This means that $g(\frac{3}{8}) > 0$, hence $g(z) \geq 0$ for $0 \leq z \leq \frac{3}{8}$.

Next, consider the case $\kappa \geq n$. In this case, set $z = c\sqrt{\frac{n}{\kappa}}$ where $c = \frac{3}{8}$. Then,

$$\frac{z}{2\kappa} - \frac{z^2}{2\kappa^2} - \frac{2z^3}{n} = \frac{1}{2\kappa}\left(c\sqrt{\frac{n}{\kappa}} - \frac{c^2 n}{\kappa^2} - 4c^3\sqrt{\frac{n}{\kappa}}\right) \geq \frac{1}{2\kappa}\left((c - 4c^3)\sqrt{\frac{n}{\kappa}} - c^2\frac{n}{\kappa}\right).$$

Note that $\sqrt{\frac{n}{\kappa}} \leq 1$, and the function $t \mapsto (c - 4c^3)t - c^2 t^2 = \frac{21}{128}t - \frac{9}{64}t^2$ is nonnegative on $[0, 1]$. Therefore, we have $g(\frac{3}{8}\sqrt{\frac{n}{\kappa}}) \geq 0$, so $g(z) \geq 0$ for $0 \leq z \leq \frac{3}{8}\sqrt{\frac{n}{\kappa}}$.

### B.4 Proof of Lemma B.2

First, note that since $0 \leq \eta \leq \frac{3}{16nL}\min\{1, \sqrt{\frac{n}{\kappa}}\}$, Lemma B.1 holds and it gives

$$\mathbb{E}\left[\|\boldsymbol{S}_{K:k+1}\boldsymbol{t}_k\|^2\right] = \mathbb{E}\left[(\boldsymbol{S}_{K:k+1}\boldsymbol{t}_k)^T(\boldsymbol{S}_{K:k+1}\boldsymbol{t}_k)\right]$$

$$\leq (1 - \eta n\mu)\mathbb{E}\left[(\boldsymbol{S}_{K-1:k+1}\boldsymbol{t}_k)^T(\boldsymbol{S}_{K-1:k+1}\boldsymbol{t}_k)\right] \leq \cdots \leq (1 - \eta n\mu)^{K-k}\mathbb{E}[\|\boldsymbol{t}_k\|^2].$$

Now, it is left to bound $\mathbb{E}[\|\boldsymbol{t}_k\|^2]$. The proof technique follows that of [12]. We express $\|\boldsymbol{t}_k\|$ as a summation of norms of partial sums of $\boldsymbol{b}_{\sigma_k(j)}$ and use a vector-valued version of the Hoeffding-Serfling inequality due to [13].

Due to summation by parts, the following identity holds:

$$\sum_{j=1}^{n} a_j b_j = a_n \sum_{j=1}^{n} b_j - \sum_{i=1}^{n-1}(a_{i+1} - a_i)\sum_{j=1}^{i} b_j.$$

We can apply the identity to $\boldsymbol{t}_k$, by substituting $a_j = \prod_{t=n}^{j+1}(\boldsymbol{I} - \eta\boldsymbol{A}_{\sigma_k(t)})$ and $b_j = \boldsymbol{b}_{\sigma_k(j)}$:

$$
\begin{aligned}
\|\boldsymbol{t}_k\| &= \left\| \sum_{j=1}^{n} \left( \prod_{t=n}^{j+1}(\boldsymbol{I} - \eta\boldsymbol{A}_{\sigma_k(t)}) \right) \boldsymbol{b}_{\sigma_k(j)} \right\| \\
&= \left\| \sum_{j=1}^{n} \boldsymbol{b}_{\sigma_k(j)} - \sum_{i=1}^{n-1} \left( \prod_{t=n}^{i+2}(\boldsymbol{I} - \eta\boldsymbol{A}_{\sigma_k(t)}) - \prod_{t=n}^{i+1}(\boldsymbol{I} - \eta\boldsymbol{A}_{\sigma_k(t)}) \right) \sum_{j=1}^{i} \boldsymbol{b}_{\sigma_k(j)} \right\| \\
&= \left\| \eta \sum_{i=1}^{n-1} \left( \prod_{t=n}^{i+2}(\boldsymbol{I} - \eta\boldsymbol{A}_{\sigma_k(t)}) \right) \boldsymbol{A}_{\sigma_k(i+1)} \sum_{j=1}^{i} \boldsymbol{b}_{\sigma_k(j)} \right\| \\
&\le \eta \sum_{i=1}^{n-1} \left\| \left( \prod_{t=n}^{i+2}(\boldsymbol{I} - \eta\boldsymbol{A}_{\sigma_k(t)}) \right) \boldsymbol{A}_{\sigma_k(i+1)} \sum_{j=1}^{i} \boldsymbol{b}_{\sigma_k(j)} \right\| \le \eta L(1+\eta L)^n \sum_{i=1}^{n-1} \left\| \sum_{j=1}^{i} \boldsymbol{b}_{\sigma_k(j)} \right\|,
\end{aligned}
$$
(B.11)

where the last step used $\left\| \boldsymbol{A}_{\sigma_k(j)} \right\| \le L$. Recall that $\eta \le \frac{1}{4nL}$, which implies $(1+\eta L)^n \le e^{1/4}$. Now, we use Lemma A.1, the Hoeffding-Serfling inequality for bounded random vectors. We restate the lemma for readers' convenience.

**Lemma A.1** ([13, Theorem 2]). *Suppose $n \ge 2$. Let $\boldsymbol{v}_1, \boldsymbol{v}_2, \ldots, \boldsymbol{v}_n \in \mathbb{R}^d$ satisfy $\|\boldsymbol{v}_j\| \le G$ for all $j$. Let $\bar{\boldsymbol{v}} = \frac{1}{n}\sum_{j=1}^{n} \boldsymbol{v}_j$. Let $\sigma \in \mathcal{S}_n$ be a uniform random permutation of $n$ elements. Then, for $i \le n$, with probability at least $1 - \delta$, we have*

$$
\left\| \frac{1}{i} \sum_{j=1}^{i} \boldsymbol{v}_{\sigma(j)} - \bar{\boldsymbol{v}} \right\| \le G\sqrt{\frac{8(1 - \frac{i-1}{n})\log\frac{2}{\delta}}{i}}.
$$

Recall that the mean $\bar{\boldsymbol{v}} = \frac{1}{n}\sum_i \boldsymbol{b}_i = \boldsymbol{0}$ for our setting, so with probability at least $1 - \delta$, we have

$$
\left\| \sum_{j=1}^{i} \boldsymbol{b}_{\sigma_k(j)} \right\| \le G\sqrt{8i\log\frac{2}{\delta}}.
$$

Using the union bound for all $i = 1, \ldots, n-1$, we have with probability at least $1 - \delta$,

$$
\sum_{i=1}^{n-1} \left\| \sum_{j=1}^{i} \boldsymbol{b}_{\sigma_k(j)} \right\| \le G\sqrt{8\log\frac{2n}{\delta}} \sum_{i=1}^{n-1} \sqrt{i} \le G\sqrt{8\log\frac{2n}{\delta}} \int_1^n \sqrt{y}\, dy
$$

$$
\le \frac{2G}{3}\sqrt{8\log\frac{2n}{\delta}} n^{3/2}.
$$
(B.12)

Substituting this to (B.11) then leads to

$$
\|\boldsymbol{t}_k\|^2 \le \frac{32e^{1/2}}{9} \eta^2 n^3 L^2 G^2 \log\frac{2n}{\delta},
$$

which holds with probability at least $1 - \delta$.

Now, set $\delta = 1/n$, and let $E$ be the probabilistic event that (B.12) holds. Let $E^c$ be the complement of $E$. Given $E^c$, directly bounding (B.11) yields

$$
\mathbb{E}\left[ \|\boldsymbol{t}_k\|^2 \mid E^c \right] \le \mathbb{E}\left[ \left( e^{1/4}\eta L \sum_{i=1}^{n-1} \left\| \sum_{j=1}^{i} \boldsymbol{b}_{\sigma_k(j)} \right\| \right)^2 \mid E^c \right] \le \frac{e^{1/2}\eta^2 n^4 L^2 G^2}{4}.
$$

Finally, putting everything together and using $\log(2n^2) \le 3\log n$ (due to $n \ge 2$),

$$
\begin{aligned}
\mathbb{E}\left[ \|\boldsymbol{t}_k\|^2 \right] &= \mathbb{E}\left[ \|\boldsymbol{t}_k\|^2 \mid E \right] \mathbb{P}[E] + \mathbb{E}\left[ \|\boldsymbol{t}_k\|^2 \mid E^c \right] \mathbb{P}[E^c] \\
&\le \frac{32e^{1/2}}{3} \eta^2 n^3 L^2 G^2 \log n + \frac{e^{1/2}\eta^2 n^4 L^2 G^2}{4} \frac{1}{n} \\
&\le 18\eta^2 n^3 L^2 G^2 \log n.
\end{aligned}
$$

## B.5 Proof of Lemma B.3

Recall that $\boldsymbol{S}_t$ and $\boldsymbol{t}_t$ depend only on the permutation $\sigma_t$. Hence, for any $t' \neq t$, $\boldsymbol{S}_t$ and $\boldsymbol{t}_t$ are independent of $\boldsymbol{S}_{t'}$ and $\boldsymbol{t}_{t'}$. Recall $k < k'$. Using independence, we can decompose the dot product.

$$
\begin{aligned}
\mathbb{E}\left[\langle \boldsymbol{S}_{K:k+1}\boldsymbol{t}_k, \boldsymbol{S}_{K:k'+1}\boldsymbol{t}_{k'}\rangle\right] &= \mathbb{E}\left[\boldsymbol{t}_k^T \boldsymbol{S}_{K:k+1}^T \boldsymbol{S}_{K:k'+1}\boldsymbol{t}_{k'}\right] \\
&= \mathbb{E}[\boldsymbol{t}_k]^T \mathbb{E}[\boldsymbol{S}_{k'-1:k+1}]^T \mathbb{E}\left[\boldsymbol{S}_{K:k'}^T \boldsymbol{S}_{K:k'+1}\boldsymbol{t}_{k'}\right] \\
&\leq \left\|\mathbb{E}[\boldsymbol{S}_{k'-1:k+1}]\mathbb{E}[\boldsymbol{t}_k]\right\| \left\|\mathbb{E}\left[\boldsymbol{S}_{K:k'}^T \boldsymbol{S}_{K:k'+1}\boldsymbol{t}_{k'}\right]\right\| \\
&\leq \left\|\mathbb{E}[\boldsymbol{S}_1]\right\|^{k'-k-1} \left\|\mathbb{E}[\boldsymbol{t}_k]\right\| \left\|\mathbb{E}\left[\boldsymbol{S}_{K:k'}^T \boldsymbol{S}_{K:k'+1}\boldsymbol{t}_{k'}\right]\right\|,
\end{aligned}
$$

where we used Cauchy-Schwarz inequality.

For the remainder of the proof, we use the following three technical lemmas that bound each of the terms in the product and get to the conclusion. The proofs of Lemmas B.4, B.5, and B.6 are deferred to Sections B.6, B.7, and B.8, respectively.

**Lemma B.4** (2nd contraction bound). *For any $0 \leq \eta \leq \frac{3}{16nL}\min\{1, \sqrt{\frac{n}{\kappa}}\}$ and any $k \in [K]$,*

$$
\|\mathbb{E}[\boldsymbol{S}_k]\| \leq 1 - \frac{\eta n \mu}{2}.
$$

**Lemma B.5.** *For any $0 \leq \eta \leq \frac{1}{2nL}$ and any $k \in [K]$,*

$$
\|\mathbb{E}[\boldsymbol{t}_k]\| \leq 4\eta nLG.
$$

**Lemma B.6.** *For any $0 \leq \eta \leq \frac{3}{16nL}\min\{1, \sqrt{\frac{n}{\kappa}}\}$ and any $k \in [K]$,*

$$
\left\|\mathbb{E}\left[\boldsymbol{S}_{K:k}^T \boldsymbol{S}_{K:k+1}\boldsymbol{t}_k\right]\right\| \leq 10(1 - \eta n \mu)^{K-k}\eta nLG.
$$

Given these lemmas, we get the desired bound:

$$
\begin{aligned}
\mathbb{E}\left[\langle \boldsymbol{S}_{K:k+1}\boldsymbol{t}_k, \boldsymbol{S}_{K:k'+1}\boldsymbol{t}_{k'}\rangle\right] &\leq 40\left(1 - \frac{\eta n \mu}{2}\right)^{k'-k-1}(1 - \eta n \mu)^{K-k'}\eta^2 n^2 L^2 G^2 \\
&\leq 40\left(1 - \frac{\eta n \mu}{2}\right)^{2K-k'-k-1}\eta^2 n^2 L^2 G^2.
\end{aligned}
$$

## B.6 Proof of the second contraction bound (Lemma B.4)

The proof goes in a similar way as the first contraction bound (Lemma B.1), but is simpler than Lemma B.1. Nevertheless, we recommend the readers to first go over Section B.3 before reading this section, because this section borrows quantities defined in Section B.3.

### B.6.1 Decomposition into elementary polynomials

For any permutation $\sigma_k$, recall that we can expand $\boldsymbol{S}_k$ in the following way:

$$
\boldsymbol{S}_k = \prod_{t=n}^{1}(\boldsymbol{I} - \eta \boldsymbol{A}_{\sigma_k(t)}) = \sum_{m=0}^{n}(-\eta)^m \sum_{1 \leq t_1 < \cdots < t_m \leq n} \boldsymbol{A}_{\sigma_k(t_m)} \cdots \boldsymbol{A}_{\sigma_k(t_1)} =: \sum_{m=0}^{n}(-\eta)^m e_m(\boldsymbol{A}_{\sigma_k[n]}),
$$

where the noncommutative elementary symmetric polynomial $e_m$ was defined in (B.1). In what follows, we will examine the expectation $\mathbb{E}[e_m(\boldsymbol{A}_{\sigma_k[n]})]$ closely and decompose $\mathbb{E}[S_k]$ into the sum of $\sum_{m=0}^{n} \frac{(-\eta n \boldsymbol{A})^m}{m!}$ and remainder terms.

**Cases $0 \leq m \leq 1$.** By definition, $e_0(\boldsymbol{A}_{\sigma_k[n]}) = \boldsymbol{I}$ and $e_1(\boldsymbol{A}_{\sigma_k[n]}) = \sum_{i=1}^{n} \boldsymbol{A}_i = n\boldsymbol{A}$, regardless of $\sigma_k$.

**Cases $2 \leq m \leq n$.** Note that each elementary symmetric polynomial $e_m(\boldsymbol{A}_{\sigma_k[n]})$ contains $\binom{n}{m}$ terms, and each term is of the form

$$
\prod_{i=m}^{1} \boldsymbol{A}_{\sigma_k(t_i)}, \text{ where } 1 \leq t_1 < \cdots < t_m \leq n.
$$

Since the indices $t_1, \ldots, t_m$ are guaranteed to be distinct, we have

$$\mathbb{E}\left[\prod_{i=m}^{1} \boldsymbol{A}_{\sigma_k(t_i)}\right] = \mathbb{E}\left[\prod_{i=1}^{m} \boldsymbol{A}_{\sigma_k(i)}\right].$$

This expectation was evaluated in (B.6):

$$\mathbb{E}\left[\prod_{i=1}^{m} \boldsymbol{A}_{\sigma_k(i)}\right] = \frac{(n-m)!}{n!}\left(\sum_{i=1}^{n} \boldsymbol{A}_i\right)^m - \frac{(n-m)!}{n!} \sum_{\substack{j_1,\ldots,j_m \in [n] \\ j_1,\ldots,j_m \text{ not unique}}} \prod_{i=1}^{m} \boldsymbol{A}_{j_i}$$

$$=: \frac{(n-m)!}{n!}(n\boldsymbol{A})^m - \frac{(n-m)!}{n!}\boldsymbol{N}_m.$$

Here, we decompose the expectation of $\prod_{i=1}^{m} \boldsymbol{A}_{\sigma_k(i)}$ into the difference of $(n\boldsymbol{A})^m$ and $\boldsymbol{N}_m$. Note that all $\binom{n}{m}$ terms in $e_m(\boldsymbol{A}_{\sigma_k[n]})$ have the same expectation, identical to the one evaluated above. Therefore, we have

$$\mathbb{E}[e_m(\boldsymbol{A}_{\sigma_k[n]})] = \binom{n}{m}\frac{(n-m)!}{n!}((n\boldsymbol{A})^m - \boldsymbol{N}_m) = \frac{(n\boldsymbol{A})^m}{m!} - \frac{\boldsymbol{N}_m}{m!}.$$

Here, note one special case, $m = 2$:

$$\boldsymbol{N}_2 := \sum_{\substack{j_1,j_2 \in [n] \\ j_1,j_2 \text{ not unique}}} \boldsymbol{A}_{j_1} \boldsymbol{A}_{j_2} = \sum_{i=1}^{n} \boldsymbol{A}_i^2,$$

which is a sum of positive semi-definite matrices.

**Spectral norm bound.** Up to this point, we obtained the following equations for $e_m(\boldsymbol{A}_{\sigma_k[n]})$'s:

$$e_0(\boldsymbol{A}_{\sigma_k[n]}) = \boldsymbol{I},$$
$$e_1(\boldsymbol{A}_{\sigma_k[n]}) = n\boldsymbol{A},$$
$$\mathbb{E}[e_2(\boldsymbol{A}_{\sigma_k[n]})] = \tfrac{1}{2}(n\boldsymbol{A})^2 - \tfrac{1}{2}\sum_{i=1}^{n} \boldsymbol{A}_i^2,$$
$$\mathbb{E}[e_m(\boldsymbol{A}_{\sigma_k[n]})] = \frac{(n\boldsymbol{A})^m}{m!} - \frac{\boldsymbol{N}_m}{m!}, \text{ for } m = 3, \ldots, n.$$

We substitute these to $\mathbb{E}[\boldsymbol{S}_k] = \sum_{m=0}^{n}(-\eta)^m \mathbb{E}[e_m(\boldsymbol{A}_{\sigma_k[n]})]$ and get

$$\mathbb{E}[\boldsymbol{S}_k] = \sum_{m=0}^{n} \frac{(-\eta n\boldsymbol{A})^m}{m!} - \frac{\eta^2}{2}\sum_{i=1}^{n} \boldsymbol{A}_i^2 - \sum_{m=3}^{n}(-\eta)^m \frac{\boldsymbol{N}_m}{m!},$$

and consequently,

$$\|\mathbb{E}[\boldsymbol{S}_k]\| \leq \left\|\sum_{m=0}^{n} \frac{(-\eta n\boldsymbol{A})^m}{m!}\right\| + \sum_{m=3}^{n} \frac{\eta^m}{m!} \|\boldsymbol{N}_m\|.$$

In what follows, we will bound each of the norms to get an upper bound.

### B.6.2 Bounding each term of the spectral norm bound

We first start with $\left\|\sum_{m=0}^{n} \frac{(-\eta n\boldsymbol{A})^m}{m!}\right\|$. Note that for any eigenvalue $s$ of the positive definite matrix $\boldsymbol{A}$, the corresponding eigenvalue of $\sum_{m=0}^{n} \frac{(-\eta n\boldsymbol{A})^m}{m!}$ is $\sum_{m=0}^{n} \frac{(-\eta ns)^m}{m!}$. Recall $\eta \leq \frac{3}{16nL} \min\{1, \sqrt{\frac{n}{\kappa}}\} \leq \frac{1}{4nL}$, so $0 \leq \eta ns \leq 1/4$ for any eigenvalue $s$ of $\boldsymbol{A}$. Since $t \mapsto \sum_{m=0}^{n} \frac{(-t)^m}{m!}$ is a positive and decreasing function on $[0, 0.25]$ for any $n \geq 2$, the maximum singular value (i.e., spectral norm) of $\sum_{m=0}^{n} \frac{(-\eta n\boldsymbol{A})^m}{m!}$ comes from the minimum eigenvalue of $\boldsymbol{A}$, hence

$$\left\|\sum_{m=0}^{n} \frac{(-\eta n\boldsymbol{A})^m}{m!}\right\| \leq \sum_{m=0}^{n} \frac{(-\eta n\mu)^m}{m!}.$$

As for $\|\boldsymbol{N}_m\|$ where $m = 3, \ldots, n$, recall that $\boldsymbol{N}_m$ is a sum of $n^m - \frac{n!}{(n-m)!}$ terms, and each of the terms has spectral norm bounded above by $L^m$. Thus,

$$\frac{1}{m!}\|\boldsymbol{N}_m\| \leq \frac{1}{m!}\left(n^m - \frac{n!}{(n-m)!}\right)L^m = \left(\frac{n^m}{m!} - \binom{n}{m}\right)L^m \leq \frac{1}{2}n^{m-1}L^m,$$

due to Lemma B.8.

### B.6.3 Concluding the proof

Putting the bounds together, we get

$$\|\mathbb{E}[\boldsymbol{S}_k]\| \leq \sum_{m=0}^{n} \frac{(-\eta n\mu)^m}{m!} + \frac{1}{2n}\sum_{m=3}^{n}(\eta n L)^m$$

$$\leq \sum_{m=0}^{2} \frac{(-\eta n\mu)^m}{m!} + \frac{1}{2n}\frac{(\eta n L)^3}{1 - \eta n L}$$

$$\leq 1 - \eta n\mu + \frac{1}{2}(\eta n\mu)^2 + \frac{2}{3n}(\eta n L)^3.$$

Here, we used $\eta n L \leq 1/4$, and the fact that $1 - t + \frac{t^2}{2} \geq \sum_{m=0}^{n} \frac{(-t)^m}{m!}$ for all $t \in [0, 0.25]$ and $n \geq 2$. The remaining step is to show that the right hand side of the inequality is bounded above by $1 - \frac{\eta n\mu}{2}$ for $0 \leq \eta \leq \frac{3}{16nL}\min\{1, \sqrt{\frac{n}{\kappa}}\}$.

Define $z = \eta n L$. Using this, we have

$$1 - \eta n\mu + \frac{1}{2}(\eta n\mu)^2 + \frac{2}{3n}(\eta n L)^3 \leq 1 - \frac{\eta n\mu}{2} \text{ for } 0 \leq \eta \leq \frac{3}{16nL}\min\left\{1, \sqrt{\frac{n}{\kappa}}\right\}$$

$$\Leftrightarrow g(z) := \frac{z}{2\kappa} - \frac{z^2}{2\kappa^2} - \frac{2z^3}{3n} \geq 0 \text{ for } 0 \leq z \leq \frac{3}{16}\min\left\{1, \sqrt{\frac{n}{\kappa}}\right\},$$

so it suffices to show the latter. One can check that $g(0) = 0$, $g'(0) > 0$ and $g'(z)$ is monotonically decreasing in $z \geq 0$, so $g(z) \geq 0$ holds for $z \in [0, c]$ for some $c > 0$. This also means that if we have $g(c) \geq 0$ for some $c > 0$, $g(z) \geq 0$ for all $z \in [0, c]$.

First, consider the case $\kappa \leq n$. Then, $n/\kappa \geq 1$ and $\kappa \geq 1$, so

$$\frac{z}{2\kappa} - \frac{z^2}{2\kappa^2} - \frac{2z^3}{3n} = \frac{1}{2\kappa}\left(z - \frac{z^2}{\kappa} - \frac{4z^3}{3n/\kappa}\right) \geq \frac{1}{2\kappa}\left(z - z^2 - \frac{4}{3}z^3\right).$$

We can check that the function $z \mapsto z - z^2 - \frac{4}{3}z^3$ is strictly positive at $z = \frac{3}{16}$. This means that $g(\frac{3}{8}) > 0$, hence $g(z) \geq 0$ for $0 \leq z \leq \frac{3}{8}$.

Next, consider the case $\kappa \geq n$. In this case, set $z = c\sqrt{\frac{n}{\kappa}}$ where $c = \frac{3}{16}$. Then,

$$\frac{z}{2\kappa} - \frac{z^2}{2\kappa^2} - \frac{2z^3}{3n} = \frac{1}{2\kappa}\left(c\sqrt{\frac{n}{\kappa}} - \frac{c^2 n}{\kappa^2} - \frac{4c^3}{3}\sqrt{\frac{n}{\kappa}}\right) \geq \frac{1}{2\kappa}\left(\left(c - \frac{4c^3}{3}\right)\sqrt{\frac{n}{\kappa}} - c^2\frac{n}{\kappa}\right).$$

Note that $\sqrt{\frac{n}{\kappa}} \leq 1$, and the function $t \mapsto (c - \frac{4}{3}c^3)t - c^2 t^2 = \frac{183}{1024}t - \frac{9}{256}t^2$ is nonnegative on $[0, 1]$. Therefore, we have $g(\frac{3}{16}\sqrt{\frac{n}{\kappa}}) \geq 0$, so $g(z) \geq 0$ for $0 \leq z \leq \frac{3}{16}\sqrt{\frac{n}{\kappa}}$.

## B.7 Proof of Lemma B.5

For this lemma, the proof is an extension of Lemma 8 in [12] from one dimension to higher dimensions. We use the law of total expectation to unwind the expectation $\mathbb{E}[\boldsymbol{t}_k]$, and use $\sum_{i=1}^{n} \boldsymbol{b}_i = \boldsymbol{0}$ to write

$$\sum_{i_{m+1} \in [n]\setminus\{i_1, \ldots, i_m\}} \boldsymbol{b}_{i_{m+1}} = -\sum_{i_{m+1} \in \{i_1, \ldots, i_m\}} \boldsymbol{b}_{i_{m+1}},$$

which turns a sum of $n - m$ terms into $m$ terms. This trick reduces the bound by a factor of $n$.

Now, expand the expectation of $\boldsymbol{t}_k$ as

$$\mathbb{E}[\boldsymbol{t}_k] = \mathbb{E}\left[\sum_{j=1}^{n}\left(\prod_{t=n}^{j+1}(\boldsymbol{I}-\eta\boldsymbol{A}_{\sigma_k(t)})\right)\boldsymbol{b}_{\sigma_k(j)}\right] = \sum_{j=1}^{n}\mathbb{E}\left[\left(\prod_{t=n}^{j+1}(\boldsymbol{I}-\eta\boldsymbol{A}_{\sigma_k(t)})\right)\boldsymbol{b}_{\sigma_k(j)}\right]$$

$$= \sum_{j=1}^{n}\mathbb{E}\left[\boldsymbol{b}_{\sigma_k(j)} + \sum_{m=1}^{n-j}(-\eta)^m \sum_{j+1\leq t_1<\cdots<t_m\leq n}\left(\prod_{i=m}^{1}\boldsymbol{A}_{\sigma_k(t_i)}\right)\boldsymbol{b}_{\sigma_k(j)}\right]$$

$$= \sum_{j=1}^{n}\sum_{m=1}^{n-j}(-\eta)^m\mathbb{E}\left[e_m(\boldsymbol{A}_{\sigma_k[n]};j+1,n)\boldsymbol{b}_{\sigma_k(j)}\right],$$

where the elementary polynomial $e_m$ is defined in (B.1). Now, fix any $t_1,\ldots,t_m$ satisfying $j+1\leq t_1<\cdots<t_m\leq n$. Since all the indices $j,t_1,\ldots,t_m$ in the product are unique, the expectation is the same for all $\binom{n-j}{m}$ such terms:

$$\mathbb{E}\left[\left(\prod_{i=m}^{1}\boldsymbol{A}_{\sigma_k(t_i)}\right)\boldsymbol{b}_{\sigma_k(j)}\right] = \mathbb{E}\left[\left(\prod_{i=1}^{m}\boldsymbol{A}_{\sigma_k(i)}\right)\boldsymbol{b}_{\sigma_k(m+1)}\right].$$

We can calculate the expectation using the law of total expectation.

$$\mathbb{E}\left[\boldsymbol{A}_{\sigma_k(1)}\boldsymbol{A}_{\sigma_k(t_2)}\ldots\boldsymbol{A}_{\sigma_k(m)}\boldsymbol{b}_{\sigma_k(m+1)}\right]$$

$$= \sum_{i_1\in[n]}\boldsymbol{A}_{i_1}\mathbb{E}\left[\boldsymbol{A}_{\sigma_k(2)}\ldots\boldsymbol{A}_{\sigma_k(m)}\boldsymbol{b}_{\sigma_k(m+1)}\mid\sigma_k(1)=i_1\right]\mathbb{P}[\sigma_k(t_1)=i_1]$$

$$= \frac{1}{n}\sum_{i_1\in[n]}\boldsymbol{A}_{i_1}\mathbb{E}\left[\boldsymbol{A}_{\sigma_k(2)}\ldots\boldsymbol{A}_{\sigma_k(m)}\boldsymbol{b}_{\sigma_k(m+1)}\mid\sigma_k(1)=i_1\right]$$

$$= \frac{1}{n(n-1)}\sum_{i_1\in[n]}\sum_{i_2\in[n]\setminus\{i_1\}}\boldsymbol{A}_{i_1}\boldsymbol{A}_{i_2}\mathbb{E}\left[\boldsymbol{A}_{\sigma_k(3)}\ldots\boldsymbol{A}_{\sigma_k(m)}\boldsymbol{b}_{\sigma_k(m+1)}\mid\sigma_k(1)=i_1,\sigma_k(2)=i_2\right]$$

$$= \frac{(n-m)!}{n!}\sum_{i_1\in[n]}\cdots\sum_{i_m\in[n]\setminus\{i_1,\ldots,i_{m-1}\}}\left(\prod_{l=1}^{m}\boldsymbol{A}_{i_l}\right)\mathbb{E}\left[\boldsymbol{b}_{\sigma_k(m+1)}\mid\sigma_k(1)=i_1,\ldots,\sigma_k(m)=i_m\right]$$

$$= \frac{(n-m)!}{n!}\sum_{i_1\in[n]}\cdots\sum_{i_m\in[n]\setminus\{i_1,\ldots,i_{m-1}\}}\left(\prod_{l=1}^{m}\boldsymbol{A}_{i_l}\right)\frac{1}{n-m}\sum_{i_{m+1}\in[n]\setminus\{i_1,\ldots,i_m\}}\boldsymbol{b}_{i_{m+1}}$$

$$= -\frac{(n-m)!}{n!}\sum_{i_1\in[n]}\cdots\sum_{i_m\in[n]\setminus\{i_1,\ldots,i_{m-1}\}}\left(\prod_{l=1}^{m}\boldsymbol{A}_{i_l}\right)\frac{1}{n-m}\sum_{i_{m+1}\in\{i_1,\ldots,i_m\}}\boldsymbol{b}_{i_{m+1}}.$$

As a consequence, we get

$$\left\|\mathbb{E}\left[\left(\prod_{i=m}^{1}\boldsymbol{A}_{\sigma_k(t_i)}\right)\boldsymbol{b}_{\sigma_k(j)}\right]\right\| \leq \frac{m}{n-m}L^mG,$$

for each term in $e_m(\boldsymbol{A}_{\sigma_k[n]};j+1,n)\boldsymbol{b}_{\sigma_k(j)}$. Applying this to the norm of $\mathbb{E}[\boldsymbol{t}_k]$ gives

$$\|\mathbb{E}[\boldsymbol{t}_k]\| \leq \sum_{j=1}^{n}\sum_{m=1}^{n-j}\eta^m\left\|\mathbb{E}\left[e_m(\boldsymbol{A}_{\sigma_k[n]};j+1,n)\boldsymbol{b}_{\sigma_k(j)}\right]\right\|$$

$$\leq \sum_{j=1}^{n}\sum_{m=1}^{n-j}\eta^m\binom{n-j}{m}\frac{m}{n-m}L^mG \leq \sum_{j=1}^{n}\sum_{m=1}^{n-1}\eta^m\binom{n}{m}\frac{m}{n-m}L^mG$$

$$= \sum_{j=1}^{n}\sum_{m=1}^{n-1}\eta^m\binom{n}{m-1}\frac{n-m+1}{n-m}L^mG \leq 2G\sum_{j=1}^{n}\sum_{m=1}^{n-1}\eta^m n^{m-1}L^m$$

$$\leq 2nG\frac{\eta L}{1-\eta nL} \leq 4\eta nLG,$$

where the last steps used $\eta nL\leq 0.5$.

## B.8 Proof of Lemma B.6

### B.8.1 Proof outline

First, recall that $\boldsymbol{S}_t$ for $t > k$ is independent of $\sigma_k$. So

$$\mathbb{E}\left[\boldsymbol{S}_{K:k}^T \boldsymbol{S}_{K:k+1} \boldsymbol{t}_k\right] = \mathbb{E}\left[\boldsymbol{S}_k^T \underbrace{\mathbb{E}\left[\boldsymbol{S}_{K:k+1}^T \boldsymbol{S}_{K:k+1}\right]}_{=:\boldsymbol{M}} \boldsymbol{t}_k\right] = \mathbb{E}\left[\boldsymbol{S}_k^T \boldsymbol{M} \boldsymbol{t}_k\right],$$

where $\boldsymbol{M}$ is a matrix satisfying $\|\boldsymbol{M}\| \leq (1 - \eta n \mu)^{K-k}$ (due to Lemma B.1) that does not depend on $\sigma_k$. Recall that

$$\boldsymbol{S}_k^T = \prod_{t=1}^{n}(\boldsymbol{I} - \eta \boldsymbol{A}_{\sigma_k(t)}) = \sum_{m=0}^{n}(-\eta)^m e_m(\boldsymbol{A}_{\sigma_k[n]})^T,$$

$$\boldsymbol{t}_k = \sum_{j=1}^{n}\left(\prod_{t=n}^{j+1}(\boldsymbol{I} - \eta \boldsymbol{A}_{\sigma_k(t)})\right)\boldsymbol{b}_{\sigma_k(j)} = \sum_{j=1}^{n}\sum_{m=1}^{n-j}(-\eta)^m e_m(\boldsymbol{A}_{\sigma_k[n]}; j+1, n)\boldsymbol{b}_{\sigma_k(j)},$$

where the elementary polynomial $e_m$ is defined in (B.1). Substituting these into $\boldsymbol{S}_k^T \boldsymbol{M} \boldsymbol{t}_k$ gives

$$\boldsymbol{S}_k^T \boldsymbol{M} \boldsymbol{t}_k = \sum_{j=1}^{n}\sum_{m=1}^{2n-j}(-\eta)^m \underbrace{\sum_{\substack{0 \leq m_1 \leq n \\ 1 \leq m_2 \leq n-j \\ m_1+m_2=m}} e_{m_1}(\boldsymbol{A}_{\sigma_k[n]})^T \boldsymbol{M} e_{m_2}(\boldsymbol{A}_{\sigma_k[n]}; j+1, n)\boldsymbol{b}_{\sigma_k(j)}}_{=:\boldsymbol{c}_{j,m}}.$$

The rest of the proof is decomposing and bounding the vector $\boldsymbol{c}_{j,m}$ for each $j = 1, \ldots, n$ and $m = 1, \ldots, 2n-j$ to get the desired bound on the norm of $\mathbb{E}[\boldsymbol{S}_k^T \boldsymbol{M} \boldsymbol{t}_k]$.

### B.8.2 Decomposing the terms in the vector $\mathbf{c}_{j,m}$ into three categories

Now fix any $j \in [n]$ and $m \in [2n-j]$, and consider any $m_1$ and $m_2$ satisfying $m_1 + m_2 = m$. Then, the product $e_{m_1}(\boldsymbol{A}_{\sigma_k[n]})^T \boldsymbol{M} e_{m_2}(\boldsymbol{A}_{\sigma_k[n]}; j+1, n)\boldsymbol{b}_{\sigma_k(j)}$ in $\boldsymbol{c}_{j,m}$ consists of $\binom{n}{m_1}\binom{n-j}{m_2}$ terms of the following form:

$$\left(\prod_{i=1}^{m_1} \boldsymbol{A}_{\sigma_k(s_i)}\right) \boldsymbol{M} \left(\prod_{i=m_2}^{1} \boldsymbol{A}_{\sigma_k(t_i)}\right) \boldsymbol{b}_{\sigma_k(j)},$$

where $1 \leq s_1 < \cdots < s_{m_1} \leq n$, and $j+1 \leq t_1 < \cdots < t_{m_2} \leq n$.

Among them, $\binom{n-1}{m_1}\binom{n-j}{m_2}$ terms have the property that $j \notin \{s_1, \ldots, s_{m_1}\}$. The remaining $\binom{n-1}{m_1-1}\binom{n-j}{m_2}$ terms satisfy $j \in \{s_1, \ldots, s_{m_1}\}$.

Using this observation, we decompose $\boldsymbol{c}_{j,m}$ into two terms $\boldsymbol{c}_{j,m} = \boldsymbol{d}_{j,m} + \boldsymbol{r}_{j,m}$. Here, $\boldsymbol{d}_{j,m}$ is a sum of terms in $\boldsymbol{c}_{j,m}$ with $s_1, \ldots, s_{m_1}$ that satisfies $j \notin \{s_1, \ldots, s_{m_1}\}$, and $\boldsymbol{r}_{j,m}$ is the sum of the remaining terms.

$$\boldsymbol{d}_{j,m} := \sum_{\substack{0 \leq m_1 \leq n-1 \\ 1 \leq m_2 \leq n-j \\ m_1+m_2=m}} \sum_{\substack{1 \leq s_1 < \cdots < s_{m_1} \leq n \\ j+1 \leq t_1 < \cdots < t_{m_2} \leq n \\ j \notin \{s_1, \ldots, s_{m_1}\}}} \left(\prod_{i=1}^{m_1} \boldsymbol{A}_{\sigma_k(s_i)}\right) \boldsymbol{M} \left(\prod_{i=m_2}^{1} \boldsymbol{A}_{\sigma_k(t_i)}\right) \boldsymbol{b}_{\sigma_k(j)},$$

$$\boldsymbol{r}_{j,m} := \boldsymbol{c}_{j,m} - \boldsymbol{d}_{j,m}.$$

Then, we will bound the sum of terms in $\boldsymbol{d}_{j,m}$ and $\boldsymbol{r}_{j,m}$ separately. There are three categories we consider:

1. Bounding $\boldsymbol{r}_{j,m}$,
2. Bounding $\boldsymbol{d}_{j,m}$, for $m \geq n/2$,
3. Bounding $\boldsymbol{d}_{j,m}$, for $m < n/2$.

The first two categories are straightforward, and the last category requires the law of total expectation trick. We will first state the bounds for the first two and then move on to the third.

### B.8.3 Directly bounding the first two categories

For the first category, the norm of each term in $\boldsymbol{r}_{j,m}$ can be easily bounded:

$$\left\| \left( \prod_{i=1}^{m_1} \boldsymbol{A}_{\sigma_k(s_i)} \right) \boldsymbol{M} \left( \prod_{i=m_2}^{1} \boldsymbol{A}_{\sigma_k(t_i)} \right) \boldsymbol{b}_{\sigma_k(j)} \right\| \le (1 - \eta n \mu)^{K-k} L^m G.$$

Since there are $\binom{n-1}{m_1-1}\binom{n-j}{m_2}$ terms in $\boldsymbol{r}_{j,m}$ for each $m_1$, $m_2$ satisfying $m_1 + m_2 = m$, we have

$$\|\boldsymbol{r}_{j,m}\| \le (1 - \eta n \mu)^{K-k} L^m G \sum_{\substack{1 \le m_1 \le n \\ 1 \le m_2 \le n-j \\ m_1 + m_2 = m}} \binom{n-1}{m_1 - 1}\binom{n-j}{m_2}$$

$$\le (1 - \eta n \mu)^{K-k} L^m G \sum_{\substack{0 \le m_1 \le n-1 \\ 0 \le m_2 \le n-j \\ m_1 + m_2 = m-1}} \binom{n-1}{m_1}\binom{n-j}{m_2}$$

$$= (1 - \eta n \mu)^{K-k} \binom{2n-j-1}{m-1} L^m G \le (1 - \eta n \mu)^{K-k} (2n)^{m-1} L^m G. \tag{B.13}$$

For the second category where $m \ge n/2$, the norm of each term in $\boldsymbol{d}_{j,m}$ can be bounded by $(1 - \eta n \mu)^{K-k} L^m G$ in the same way. Now, since there are $\binom{n-1}{m_1}\binom{n-j}{m_2}$ terms for each $m_1$ and $m_2$, we have

$$\|\boldsymbol{d}_{j,m}\| \le (1 - \eta n \mu)^{K-k} L^m G \sum_{\substack{0 \le m_1 \le n-1 \\ 1 \le m_2 \le n-j \\ m_1 + m_2 = m}} \binom{n-1}{m_1}\binom{n-j}{m_2}$$

$$\le (1 - \eta n \mu)^{K-k} L^m G \sum_{\substack{0 \le m_1 \le n-1 \\ 0 \le m_2 \le n-j \\ m_1 + m_2 = m}} \binom{n-1}{m_1}\binom{n-j}{m_2}$$

$$= (1 - \eta n \mu)^{K-k} \binom{2n-j-1}{m} L^m G \le (1 - \eta n \mu)^{K-k} \binom{2n}{m} L^m G.$$

Since $m \ge n/2$, we can upper-bound $\binom{2n}{m}$ with a constant multiple of $\binom{2n}{m-1}$:

$$\binom{2n}{m} = \frac{2n-m+1}{m}\binom{2n}{m-1} \le 4\binom{2n}{m-1},$$

where the inequality holds because

$$m \ge n/2 \text{ and } n \ge 2 \Rightarrow 5m \ge 2n+1 \Leftrightarrow 4m \ge 2n - m + 1.$$

Therefore, if $m \ge n/2$,

$$\|\boldsymbol{d}_{j,m}\| \le 4(1 - \eta n \mu)^{K-k} (2n)^{m-1} L^m G. \tag{B.14}$$

### B.8.4 Bounding the third category using the law of total expectation

We will show a similar bound for $\|\mathbb{E}[\boldsymbol{d}_{j,m}]\|$ in case of $m < n/2$ as well, but the third category requires a bit more care. For $m < n/2$, we need to use the law of total expectation to exploit the fact that $\sum_i \boldsymbol{b}_i = \boldsymbol{0}$ and reduce a factor of $n$.

Now consider the expectation for a term in $\boldsymbol{d}_{j,m}$:

$$\mathbb{E}\left[ \left( \prod_{i=1}^{m_1} \boldsymbol{A}_{\sigma_k(s_i)} \right) \boldsymbol{M} \left( \prod_{i=m_2}^{1} \boldsymbol{A}_{\sigma_k(t_i)} \right) \boldsymbol{b}_{\sigma_k(j)} \right].$$

We will use the law of total expectation to bound the norm of this expectation. One thing we should be careful of is that there may be overlapping indices between $\{s_1, \ldots, s_{m_1}\}$ and $\{t_1, \ldots, t_{m_2}\}$. For

now, let us assume that there are no overlapping indices; hence, $s_1, \ldots, s_{m_1}, t_1, \ldots, t_{m_2}, j$ are all distinct. Then, the expectation can be expanded as the following.

$$
\mathbb{E}\left[\left(\prod_{i=1}^{m_1} \boldsymbol{A}_{\sigma_k(s_i)}\right) \boldsymbol{M}\left(\prod_{i=m_2}^{1} \boldsymbol{A}_{\sigma_k(t_i)}\right) \boldsymbol{b}_{\sigma_k(j)}\right]
$$

$$
= \sum_{i_1 \in [n]} \mathbb{E}\left[\left(\prod_{i=1}^{m_1} \boldsymbol{A}_{\sigma_k(s_i)}\right) \boldsymbol{M}\left(\prod_{i=m_2}^{1} \boldsymbol{A}_{\sigma_k(t_i)}\right) \boldsymbol{b}_{\sigma_k(j)} \mid \sigma_k(s_1) = i_1\right] \mathbb{P}[\sigma_k(s_1) = i_1]
$$

$$
= \frac{1}{n} \sum_{i_1 \in [n]} \boldsymbol{A}_{i_1} \mathbb{E}\left[\left(\prod_{i=2}^{m_1} \boldsymbol{A}_{\sigma_k(s_i)}\right) \boldsymbol{M}\left(\prod_{i=m_2}^{1} \boldsymbol{A}_{\sigma_k(t_i)}\right) \boldsymbol{b}_{\sigma_k(j)} \mid \sigma_k(s_1) = i_1\right]
$$

$$
= \frac{1}{n(n-1)} \sum_{i_1 \in [n]} \sum_{i_2 \in [n]\setminus\{i_1\}} \boldsymbol{A}_{i_1} \boldsymbol{A}_{i_2} \mathbb{E}\left[\left(\prod_{i=3}^{m_1} \boldsymbol{A}_{\sigma_k(s_i)}\right) \boldsymbol{M}\left(\prod_{i=m_2}^{1} \boldsymbol{A}_{\sigma_k(t_i)}\right) \boldsymbol{b}_{\sigma_k(j)} \mid \sigma_k(s_1) = i_1, \sigma_k(s_2) = i_2\right]
$$

$$
= \frac{(n-m)!}{n!} \sum_{i_1 \in [n]} \cdots \sum_{i_{m_1} \in [n]\setminus\{i_1,\ldots,i_{m_1-1}\}} \sum_{l_{m_2} \in [n]\setminus\{i_1,\ldots,i_{m_1}\}} \cdots \sum_{l_1 \in [n]\setminus\{i_1,\ldots,i_{m_1},l_2,\ldots,l_{m_2}\}}
$$

$$
\left(\prod_{t=1}^{m_1} \boldsymbol{A}_{i_t}\right) \boldsymbol{M}\left(\prod_{t=m_2}^{1} \boldsymbol{A}_{l_t}\right) \mathbb{E}\left[\boldsymbol{b}_{\sigma_k(j)} \mid \sigma_k(s_1) = i_1, \ldots, \sigma_k(t_{m_2}) = l_{m_2}\right].
$$

Here, by $\sum_t \boldsymbol{b}_t = \boldsymbol{0}$,

$$
\mathbb{E}\left[\boldsymbol{b}_{\sigma_k(j)} \mid \sigma_k(s_1) = i_1, \ldots, \sigma_k(t_{m_2}) = l_{m_2}\right] = \frac{1}{n-m} \sum_{t \in [n]\setminus\{i_1,\ldots,i_{m_1},l_1,\ldots,l_{m_2}\}} \boldsymbol{b}_t
$$

$$
= -\frac{1}{n-m} \sum_{t \in \{i_1,\ldots,i_{m_1},l_1,\ldots,l_{m_2}\}} \boldsymbol{b}_t.
$$

Putting these together, we can get a bound on the norm of the expectation:

$$
\left\|\mathbb{E}\left[\left(\prod_{i=1}^{m_1} \boldsymbol{A}_{\sigma_k(s_i)}\right) \boldsymbol{M}\left(\prod_{i=m_2}^{1} \boldsymbol{A}_{\sigma_k(t_i)}\right) \boldsymbol{b}_{\sigma_k(j)}\right]\right\|
$$

$$
\leq \frac{(n-m)!}{n!} \sum_{i_1 \in [n]} \cdots \sum_{i_{m_1} \in [n]\setminus\{i_1,\ldots,i_{m_1-1}\}} \sum_{l_{m_2} \in [n]\setminus\{i_1,\ldots,i_{m_1}\}} \cdots \sum_{l_1 \in [n]\setminus\{i_1,\ldots,i_{m_1},l_2,\ldots,l_{m_2}\}}
$$

$$
\left\|\left(\prod_{t=1}^{m_1} \boldsymbol{A}_{i_t}\right) \boldsymbol{M}\left(\prod_{t=m_2}^{1} \boldsymbol{A}_{l_t}\right) \frac{1}{n-m} \sum_{t \in \{i_1,\ldots,i_{m_1},l_1,\ldots,l_{m_2}\}} \boldsymbol{b}_t\right\|
$$

$$
\leq (1-\eta n\mu)^{K-k} \frac{m}{n-m} L^m G.
$$

What if there are overlapping indices between $\{s_1, \ldots, s_{m_1}\}$ and $\{t_1, \ldots, t_{m_2}\}$? Suppose the union of the two sets has $\tilde{m} < m$ elements. Notice that even in this case, $j$ does not overlap with any $s_i$ or $t_i$. So, we can do a similar calculation and use the $\sum_t \boldsymbol{b}_t = \boldsymbol{0}$ trick at the end. This gives

$$
\left\|\mathbb{E}\left[\left(\prod_{i=1}^{m_1} \boldsymbol{A}_{\sigma_k(s_i)}\right) \boldsymbol{M}\left(\prod_{i=m_2}^{1} \boldsymbol{A}_{\sigma_k(t_i)}\right) \boldsymbol{b}_{\sigma_k(j)}\right]\right\| \leq (1-\eta n\mu)^{K-k} \frac{\tilde{m}}{n-\tilde{m}} L^m G
$$

$$
\leq (1-\eta n\mu)^{K-k} \frac{m}{n-m} L^m G,
$$

so the same upper bound holds even for the terms with overlapping indices. Now, since there are $\binom{n-1}{m_1}\binom{n-j}{m_2}$ such terms for each $m_1$ and $m_2$, we have

$$
\|\mathbb{E}[\boldsymbol{d}_{j,m}]\| \leq (1-\eta n\mu)^{K-k} \frac{m}{n-m} L^m G \sum_{\substack{0 \leq m_1 \leq n-1 \\ 1 \leq m_2 \leq n-j \\ m_1+m_2=m}} \binom{n-1}{m_1}\binom{n-j}{m_2}
$$

$$\leq (1 - \eta n \mu)^{K-k} \frac{m}{n-m} \binom{2n-j-1}{m} L^m G.$$

Note that

$$\frac{m}{n-m} \binom{2n-j-1}{m} \leq \frac{m}{n-m} \binom{2n}{m} = \frac{2n-m+1}{n-m} \binom{2n}{m-1} \leq 3 \binom{2n}{m-1},$$

this is because

$$\begin{aligned}
m < n/2 &\Rightarrow 2m + 1 \leq n \\
&\Leftrightarrow 2m + 1 + (2n - 3m) \leq n + (2n - 3m) \\
&\Leftrightarrow \frac{2m + 1 + (2n - 3m)}{n + (2n - 3m)} = \frac{2n - m + 1}{3n - 3m} \leq 1.
\end{aligned}$$

Thus, for $m < n/2$, we obtain the following bound on $\|\mathbb{E}[d_{j,m}]\|$:

$$\|\mathbb{E}[d_{j,m}]\| \leq 3(1 - \eta n \mu)^{K-k} (2n)^{m-1} L^m G. \tag{B.15}$$

### B.8.5 Concluding the proof

Finally, using the bounds (B.13), (B.14), and (B.15), we get

$$\begin{aligned}
\|\mathbb{E}[S_k^T M t_k]\| &\leq \sum_{j=1}^{n} \sum_{m=1}^{2n-j} \eta^m \|\mathbb{E}[c_{j,m}]\| \leq \sum_{j=1}^{n} \sum_{m=1}^{2n-j} \eta^m (\|\mathbb{E}[d_{j,m}]\| + \|\mathbb{E}[r_{j,m}]\|) \\
&\leq \sum_{j=1}^{n} \sum_{m=1}^{2n-j} 5(1 - \eta n \mu)^{K-k} (2n)^{m-1} \eta^m L^m G \\
&\leq 5(1 - \eta n \mu)^{K-k} nG \frac{\eta L}{1 - 2\eta n L} \leq 10(1 - \eta n \mu)^{K-k} \eta n L G,
\end{aligned}$$

where the last inequality used $\eta \leq \frac{1}{4nL}$.

### B.9 Technical lemmas on binomial coefficients

**Lemma B.7.** *For any $n \in \mathbb{N}$ and $2 \leq m \leq n$,*

$$\binom{2n}{m} - 2^m \binom{n}{m} \leq \frac{(2n)^m}{m!} - 2^m \binom{n}{m} \leq \frac{(2n)^{m-1}}{(m-2)!}.$$

**Proof**   The first inequality is straightforward from

$$\binom{2n}{m} = \frac{2n(2n-1)\dots(2n-m+1)}{m!} \leq \frac{(2n)^m}{m!}.$$

The remaining inequality is shown with mathematical induction. For the base case ($m = 2$),

$$\frac{(2n)^2}{2!} - 2^2 \binom{n}{2} = 2n^2 - 4\frac{n(n-1)}{2} = 2n = \frac{(2n)^{2-1}}{(2-2)!},$$

so the inequality holds with equality. For the inductive case, suppose

$$\frac{(2n)^m}{m!} - 2^m \binom{n}{m} \leq \frac{(2n)^{m-1}}{(m-2)!}$$

holds, where $2 \leq m \leq n - 1$. Then,

$$\begin{aligned}
\frac{(2n)^{m+1}}{(m+1)!} - 2^{m+1} \binom{n}{m+1} &= \frac{2n}{m+1} \frac{(2n)^m}{m!} - \frac{2(n-m)}{m+1} 2^m \binom{n}{m} \\
&= \frac{2m}{m+1} \frac{(2n)^m}{m!} + \frac{2(n-m)}{m+1} \left( \frac{(2n)^m}{m!} - 2^m \binom{n}{m} \right)
\end{aligned}$$

$$\leq \frac{2m}{m+1} \frac{(2n)^m}{m!} + \frac{2(n-m)}{m+1} \frac{(2n)^{m-1}}{(m-2)!}$$

$$= \frac{(2n)^{m-1}}{(m+1)(m-2)!} \left( \frac{4n}{m-1} + 2(n-m) \right)$$

$$= \frac{(2n)^{m-1}}{(m+1)(m-2)!} \frac{2mn + 2n - 2m^2 + 2m}{m-1}$$

$$= \frac{(2n)^m}{(m-1)!} \frac{mn + n - m^2 + m}{n(m+1)}.$$

It now suffices to check that $\frac{mn+n-m^2+m}{n(m+1)} \leq 1$.

$$\frac{mn + n - m^2 + m}{n(m+1)} \leq 1 \Leftrightarrow mn + n - m^2 + m \leq mn + n \Leftrightarrow m \leq m^2.$$

Since $m \geq 2$, the inequality holds. This finishes the proof. $\qquad\square$

**Lemma B.8.** *For any $n \in \mathbb{N}$ and $2 \leq m \leq n$,*

$$\frac{n^m}{m!} - \binom{n}{m} \leq \frac{n^{m-1}}{2(m-2)!}.$$

**Proof** The is shown with mathematical induction. For the base case ($m = 2$),

$$\frac{n^2}{2!} - \binom{n}{2} = \frac{n^2}{2} - \frac{n(n-1)}{2} = \frac{n}{2},$$

so the inequality holds with equality. For the inductive case, suppose

$$\frac{n^m}{m!} - \binom{n}{m} \leq \frac{n^{m-1}}{2(m-2)!}$$

holds, where $2 \leq m \leq n - 1$. Then,

$$\frac{n^{m+1}}{(m+1)!} - \binom{n}{m+1} = \frac{n}{m+1} \frac{n^m}{m!} - \frac{n-m}{m+1} \binom{n}{m}$$

$$= \frac{m}{m+1} \frac{n^m}{m!} + \frac{n-m}{m+1} \left( \frac{n^m}{m!} - \binom{n}{m} \right)$$

$$\leq \frac{1}{m+1} \frac{n^m}{(m-1)!} + \frac{n-m}{m+1} \frac{n^{m-1}}{2(m-2)!}$$

$$= \frac{n^{m-1}}{2(m+1)(m-2)!} \left( \frac{2n}{m-1} + n - m \right)$$

$$= \frac{n^{m-1}}{2(m+1)(m-2)!} \frac{mn + n - m^2 + m}{m-1}$$

$$= \frac{n^m}{2(m-1)!} \frac{mn + n - m^2 + m}{n(m+1)}.$$

It now suffices to check that $\frac{mn+n-m^2+m}{n(m+1)} \leq 1$.

$$\frac{mn + n - m^2 + m}{n(m+1)} \leq 1 \Leftrightarrow mn + n - m^2 + m \leq mn + n \Leftrightarrow m \leq m^2.$$

Since $m \geq 2$, the inequality holds. This finishes the proof. $\qquad\square$

# C RANDOMSHUFFLE: Tail average bound for strongly convex quadratics

In this section, we provide details for Remark 3. We first state the theorem for the tail average iterate, which improves the leading constants of Theorem 2 by a factor of $\kappa$. We will then provide the proof for Theorem C.1 in the subsequent subsections.

**Theorem C.1** (Tail averaging). *Assume that* $F(\boldsymbol{x}) := \frac{1}{n} \sum_{i=1}^{n} f_i(\boldsymbol{x}) = \frac{1}{2} \boldsymbol{x}^T \boldsymbol{A} \boldsymbol{x}$ *and* $F$ *is* $\mu$-*strongly convex. Let* $f_i(\boldsymbol{x}) := \frac{1}{2} \boldsymbol{x}^T \boldsymbol{A}_i \boldsymbol{x} + \boldsymbol{b}_i^T \boldsymbol{x}$ *and* $f_i \in C_L^1(\mathbb{R}^d)$. *Consider* RANDOMSHUFFLE *for the number of epochs* $K$ *satisfying* $K \geq 128\kappa \max\{1, \sqrt{\frac{\kappa}{n}}\} \log(nK)$, *step size* $\eta_i^k = \eta := \frac{16 \log(nK)}{\mu n K}$, *and initialization* $\boldsymbol{x}_0$. *Then for* $G := \max_{i \in [n]} \|\boldsymbol{b}_i\|$ *and some constant* $c = O(\kappa^3)$,

$$\mathbb{E}[F(\bar{\boldsymbol{x}})] - F^* \leq \frac{\mu \|\boldsymbol{x}_0\|^2}{16 n^2 K^2} + \frac{c \cdot G^2 \cdot \log^2(nK)}{n^2 K^2} + \frac{c \cdot G^2 \cdot \log^4(nK)}{n K^3},$$

*where* $\bar{\boldsymbol{x}}$ *is the tail average of the iterates* $\bar{\boldsymbol{x}} = \frac{\sum_{k=\lceil K/2 \rceil}^{K} \boldsymbol{x}_0^k}{K - \lceil K/2 \rceil + 1}$.

## C.1 Proof outline

Recall the definitions

$$f_i(\boldsymbol{x}) := \tfrac{1}{2} \boldsymbol{x}^T \boldsymbol{A}_i \boldsymbol{x} + \boldsymbol{b}_i^T \boldsymbol{x}, \; F(\boldsymbol{x}) := \frac{1}{n} \sum_{i=1}^{n} f_i(\boldsymbol{x}) = \tfrac{1}{2} \boldsymbol{x}^T \boldsymbol{A} \boldsymbol{x},$$

where $f_i$'s are $L$-smooth and $F$ is $\mu$ strongly convex. This is equivalent to saying that $\|\boldsymbol{A}_i\| \leq L$ and $\boldsymbol{A} := \frac{1}{n} \sum_{i=1}^{n} \boldsymbol{A}_i \succeq \mu \boldsymbol{I}$. Also note that $F$ is minimized at $\boldsymbol{x}^* = \boldsymbol{0}$ and $\sum_{i=1}^{n} \boldsymbol{b}_i = \boldsymbol{0}$. We let $G := \max_{i \in [n]} \|\boldsymbol{b}_i\|$.

In order to get a bound for tail average of the iterates, we need to modify our proof technique a bit. Instead of unrolling all the update equations (as done in Theorem 2), we only consider one epoch, and derive a per-epoch improvement bound. In the Proof of Theorem 2, we derived the epoch update equation:

$$\boldsymbol{x}_0^{k+1} = \underbrace{\left[ \prod_{t=n}^{1} (\boldsymbol{I} - \eta \boldsymbol{A}_{\sigma_k(t)}) \right]}_{=: \boldsymbol{S}_k} \boldsymbol{x}_0^k - \eta \underbrace{\left[ \sum_{j=1}^{n} \left( \prod_{t=n}^{j+1} (\boldsymbol{I} - \eta \boldsymbol{A}_{\sigma_k(t)}) \right) \boldsymbol{b}_{\sigma_k(j)} \right]}_{=: \boldsymbol{t}_k}$$

$$= \boldsymbol{S}_k \boldsymbol{x}_0^k - \eta \boldsymbol{t}_k.$$

Using this update, the expected distance to the optimum squared $\|\boldsymbol{x}_0^{k+1}\|^2$ given $\boldsymbol{x}_0^k$ is

$$\mathbb{E}[\|\boldsymbol{x}_0^{k+1}\|^2] = \mathbb{E}[\|\boldsymbol{S}_k \boldsymbol{x}_0^k\|^2] - 2\eta \mathbb{E}[\langle \boldsymbol{S}_k \boldsymbol{x}_0^k, \boldsymbol{t}_k \rangle] + \eta^2 \mathbb{E}[\|\boldsymbol{t}_k\|^2]$$

$$= \mathbb{E}[\|\boldsymbol{S}_k \boldsymbol{x}_0^k\|^2] - 2\eta \langle \boldsymbol{x}_0^k, \mathbb{E}[\boldsymbol{S}_k^T \boldsymbol{t}_k] \rangle + \eta^2 \mathbb{E}[\|\boldsymbol{t}_k\|^2]$$

$$\leq \mathbb{E}[\|\boldsymbol{S}_k \boldsymbol{x}_0^k\|^2] + 2\eta \|\boldsymbol{x}_0^k\| \|\mathbb{E}[\boldsymbol{S}_k^T \boldsymbol{t}_k]\| + \eta^2 \mathbb{E}[\|\boldsymbol{t}_k\|^2],$$

where the last inequality is due to Cauchy-Schwarz. We now bound each term in the right hand side. The first term can be bounded by a slight refinement of the first contraction bound (Lemma B.1).

**Lemma C.2** (3rd contraction bound). *For any* $0 \leq \eta \leq \frac{1}{8nL} \min\{1, \sqrt{\frac{n}{\kappa}}\}$,

$$\mathbb{E}[\|\boldsymbol{S}_k \boldsymbol{x}_0^k\|^2] \leq \left(1 - \frac{\eta n \mu}{2}\right) \|\boldsymbol{x}_0^k\|^2 - 2\eta n F(\boldsymbol{x}_0^k).$$

The next two terms can be bounded using Lemmas B.6 and B.2:

$$\|\mathbb{E}[\boldsymbol{S}_k^T \boldsymbol{t}_k]\| \leq 10 \eta n L G, \;\; \mathbb{E}[\|\boldsymbol{t}_k\|^2] \leq 18 \eta^2 n^3 L^2 G^2 \log n.$$

Substituting these bounds, we get

$$\mathbb{E}[\|\boldsymbol{x}_0^{k+1}\|^2] \leq \left(1 - \frac{\eta n \mu}{2}\right) \|\boldsymbol{x}_0^k\|^2 - 2\eta n F(\boldsymbol{x}_0^k) + 20 \eta^2 n L G \|\boldsymbol{x}_0^k\| + 18 \eta^4 n^3 L^2 G^2 \log n.$$

We then use the AM-GM inequality $ab \leq \frac{a^2+b^2}{2}$ on $a = \frac{20\sqrt{2}\eta^{3/2}n^{1/2}LG}{\mu^{1/2}}$ and $b = \sqrt{\frac{\eta n \mu}{2}} \left\| \boldsymbol{x}_0^k \right\|$, and get

$$\mathbb{E}[\left\| \boldsymbol{x}_0^{k+1} \right\|^2] \leq \left(1 - \frac{\eta n \mu}{4}\right) \left\| \boldsymbol{x}_0^k \right\|^2 - 2\eta n F(\boldsymbol{x}_0^k) + \frac{400\eta^3 n L^2 G^2}{\mu} + 18\eta^4 n^3 L^2 G^2 \log n. \quad \text{(C.1)}$$

Now, consider the following rearrangement of (C.1)

$$2\eta n \mathbb{E}[F(\boldsymbol{x}_0^k)] \leq \left(1 - \frac{\eta n \mu}{4}\right) \mathbb{E}[\left\| \boldsymbol{x}_0^k \right\|^2] - \mathbb{E}[\left\| \boldsymbol{x}_0^{k+1} \right\|^2] + \frac{400\eta^3 n L^2 G^2}{\mu} + 18\eta^4 n^3 L^2 G^2 \log n.$$

Summing up both sides of the inequality for $k = \lceil K/2 \rceil, \ldots, K$ gives

$$2\eta n \sum_{k=\lceil K/2 \rceil}^{K} \mathbb{E}[F(\boldsymbol{x}_0^k)] \leq \left(1 - \frac{\eta n \mu}{4}\right) \mathbb{E}\left[\left\| \boldsymbol{x}_0^{\lceil K/2 \rceil} \right\|^2\right]$$

$$+ \left(K - \left\lceil \frac{K}{2} \right\rceil + 1\right) \left(\frac{400\eta^3 n L^2 G^2}{\mu} + 18\eta^4 n^3 L^2 G^2 \log n\right).$$

Unwinding the recursion (C.1) from $k = \lceil K/2 \rceil - 1$ until $k = 1$ (while using $F(x) \geq 0$), we obtain

$$\mathbb{E}\left[\left\| \boldsymbol{x}_0^{\lceil K/2 \rceil} \right\|^2\right] \leq \left(1 - \frac{\eta n \mu}{4}\right)^{\lceil K/2 \rceil - 1} \left\| \boldsymbol{x}_0^1 \right\|^2$$

$$+ \left(\left\lceil \frac{K}{2} \right\rceil - 1\right) \left(\frac{400\eta^3 n L^2 G^2}{\mu} + 18\eta^4 n^3 L^2 G^2 \log n\right),$$

so by substitution we have

$$2\eta n \sum_{k=\lceil K/2 \rceil}^{K} \mathbb{E}[F(\boldsymbol{x}_0^k)] \leq \left(1 - \frac{\eta n \mu}{4}\right)^{\lceil K/2 \rceil} \left\| \boldsymbol{x}_0^1 \right\|^2 + \frac{400\eta^3 n K L^2 G^2}{\mu} + 18\eta^4 n^3 K L^2 G^2 \log n.$$

Now, we take the average of both sides by dividing both sides by $K - \lceil K/2 \rceil + 1$. We then further divide both sides by $2\eta n$ and apply Jensen's inequality to get a bound on the tail average $\bar{\boldsymbol{x}} := \frac{\sum_{k=\lceil K/2 \rceil}^{K} \boldsymbol{x}_0^k}{K - \lceil K/2 \rceil + 1}$.

$$\mathbb{E}[F(\bar{\boldsymbol{x}})] \leq \frac{\sum_{k=\lceil K/2 \rceil}^{K} \mathbb{E}[F(\boldsymbol{x}_0^k)]}{K - \lceil K/2 \rceil + 1}$$

$$\leq \frac{1}{\eta n K} \left(1 - \frac{\eta n \mu}{4}\right)^{\lceil K/2 \rceil} \left\| \boldsymbol{x}_0^1 \right\|^2 + \frac{400\eta^2 L^2 G^2}{\mu} + 18\eta^3 n^2 L^2 G^2 \log n,$$

where the last inequality used $K - \lceil K/2 \rceil + 1 \geq K/2$. Lastly, substituting $\eta = \frac{16 \log nK}{\mu n K}$ gives us

$$\left(1 - \frac{\eta n \mu}{4}\right)^{\lceil K/2 \rceil} = \left(1 - \frac{4 \log nK}{K}\right)^{\lceil K/2 \rceil} \leq \left(1 - \frac{2 \log nK}{\lceil K/2 \rceil}\right)^{\lceil K/2 \rceil} \leq \frac{1}{n^2 K^2}.$$

This results in the bound

$$\mathbb{E}[F(\bar{\boldsymbol{x}}) - F^*] \leq \frac{\mu \left\| \boldsymbol{x}_0^1 \right\|^2}{16 n^2 K^2} + \mathcal{O}\left(\frac{L^2 G^2}{\mu^3} \left(\frac{\log^2(nK)}{n^2 K^2} + \frac{\log^4(nK)}{n K^3}\right)\right),$$

as desired. Recall that the bound holds for $\eta \leq \frac{1}{8nL} \min\{1, \sqrt{\frac{n}{\kappa}}\}$, so $K$ must be large enough so that

$$\frac{16 \log nK}{\mu n K} \leq \frac{1}{8nL} \min\left\{1, \sqrt{\frac{n}{\kappa}}\right\}.$$

This gives us the epoch requirement $K \geq 128\kappa \max\{1, \sqrt{\frac{\kappa}{n}}\} \log nK$.

## C.2 Proof of the third contraction bound (Lemma C.2)

The proof is an extension of the proof of Lemma B.1, so we recommend the authors to read Section B.3 before reading this subsection. From the definiton $F(\boldsymbol{x}_0^k) = \frac{1}{2}(\boldsymbol{x}_0^k)^T \boldsymbol{A}\boldsymbol{x}_0^k$, we have

$$
\begin{aligned}
\mathbb{E}\left[\left\|\boldsymbol{S}_k\boldsymbol{x}_0^k\right\|^2\right] &= (\boldsymbol{x}_0^k)^T \mathbb{E}[\boldsymbol{S}_k^T \boldsymbol{S}_k]\boldsymbol{x}_0^k \\
&= (\boldsymbol{x}_0^k)^T \left(\mathbb{E}[\boldsymbol{S}_k^T \boldsymbol{S}_k] + \eta n\boldsymbol{A}\right)\boldsymbol{x}_0^k - \eta n(\boldsymbol{x}_0^k)^T \boldsymbol{A}\boldsymbol{x}_0^k \\
&\leq \left\|\mathbb{E}[\boldsymbol{S}_k^T \boldsymbol{S}_k] + \eta n\boldsymbol{A}\right\| \left\|\boldsymbol{x}_0^k\right\|^2 - 2\eta n F(\boldsymbol{x}_0^k).
\end{aligned}
$$

The remainder of the proof is to bound $\left\|\mathbb{E}[\boldsymbol{S}_k^T \boldsymbol{S}_k] + \eta n\boldsymbol{A}\right\| \leq 1 - \frac{\eta n\mu}{2}$ for $0 \leq \eta \leq \frac{1}{8nL}\min\{1, \sqrt{\frac{n}{\kappa}}\}$.

As seen in (B.7) (Section B.3), the expectation of $\boldsymbol{S}_k^T \boldsymbol{S}_k$ reads

$$
\begin{aligned}
\mathbb{E}[\boldsymbol{S}_k^T \boldsymbol{S}_k] = \sum_{m=0}^{n} \frac{(-2\eta n\boldsymbol{A})^m}{m!} - \eta^2 \sum_{i=1}^{n} \boldsymbol{A}_i^2 + \sum_{m=3}^{n} (-\eta)^m \left(\mathbb{E}[\boldsymbol{R}_m] - \frac{2^m}{m!}\boldsymbol{N}_m\right) \\
+ \sum_{m=n+1}^{2n} (-\eta)^m \mathbb{E}[\boldsymbol{C}_m],
\end{aligned}
$$

where $\boldsymbol{C}_m, \boldsymbol{R}_m, \boldsymbol{N}_m$ are defined in (B.3), (B.5) and (B.6). Then,

$$
\begin{aligned}
\left\|\mathbb{E}[\boldsymbol{S}_k^T \boldsymbol{S}_k] + \eta n\boldsymbol{A}\right\| &\leq \left\|\eta n\boldsymbol{A} + \sum_{m=0}^{n} \frac{(-2\eta n\boldsymbol{A})^m}{m!}\right\| + \sum_{m=3}^{n} \eta^m \left(\|\mathbb{E}[\boldsymbol{R}_m]\| + \frac{2^m}{m!}\|\boldsymbol{N}_m\|\right) \\
&\quad + \sum_{m=n+1}^{2n} \eta^m \|\mathbb{E}[\boldsymbol{C}_m]\| \\
&\leq \left\|\eta n\boldsymbol{A} + \sum_{m=0}^{n} \frac{(-2\eta n\boldsymbol{A})^m}{m!}\right\| + \frac{1}{n}\sum_{m=3}^{2n} (2\eta nL)^m,
\end{aligned}
$$

where the bounds for $\|\mathbb{E}[\boldsymbol{R}_m]\|, \|\boldsymbol{N}_m\|, \|\mathbb{E}[\boldsymbol{C}_m]\|$ are from Eqs (B.8), (B.9), and (B.10). So, it is left to bound the term $\left\|\eta n\boldsymbol{A} + \sum_{m=0}^{n} \frac{(-2\eta n\boldsymbol{A})^m}{m!}\right\|$.

Note that for any eigenvalue $s$ of the positive definite matrix $\boldsymbol{A}$, the corresponding eigenvalue of $\eta n\boldsymbol{A} + \sum_{m=0}^{n} \frac{(-2\eta n\boldsymbol{A})^m}{m!}$ is $\eta ns + \sum_{m=0}^{n} \frac{(-2\eta ns)^m}{m!}$. Recall $\eta \leq \frac{1}{8nL}\min\{1, \sqrt{\frac{n}{\kappa}}\} \leq \frac{1}{8nL}$, so $0 \leq 2\eta ns \leq 1/4$ for any eigenvalue $s$ of $\boldsymbol{A}$. Since $t \mapsto \frac{t}{2} + \sum_{m=0}^{n} \frac{(-t)^m}{m!}$ is a positive and decreasing function on $[0, 1/4]$ for any $n \geq 2$, the matrix $\eta n\boldsymbol{A} + \sum_{m=0}^{n} \frac{(-2\eta n\boldsymbol{A})^m}{m!}$ is positive definite and its maximum singular value (i.e., spectral norm) comes from the minimum eigenvalue of $\boldsymbol{A}$, hence

$$
\left\|\eta n\boldsymbol{A} + \sum_{m=0}^{n} \frac{(-2\eta n\boldsymbol{A})^m}{m!}\right\| \leq \eta n\mu + \sum_{m=0}^{n} \frac{(-2\eta n\mu)^m}{m!}.
$$

Putting the bounds together, we get

$$
\begin{aligned}
\left\|\mathbb{E}[\boldsymbol{S}_k^T \boldsymbol{S}_k] + \eta n\boldsymbol{A}\right\| &\leq \eta n\mu + \sum_{m=0}^{n} \frac{(-2\eta n\mu)^m}{m!} + \frac{1}{n}\sum_{m=3}^{2n} (2\eta nL)^m \\
&\leq \eta n\mu + \sum_{m=0}^{2} \frac{(-2\eta n\mu)^m}{m!} + \frac{1}{n}\frac{(2\eta nL)^3}{1 - 2\eta nL} \\
&\leq 1 - \eta n\mu + \frac{1}{2}(2\eta n\mu)^2 + \frac{2}{n}(2\eta nL)^3.
\end{aligned}
$$

Here, we used $2\eta nL \leq 1/2$, and the fact that $1 - t + \frac{t^2}{2} \geq \sum_{m=0}^{n} \frac{(-t)^m}{m!}$ for all $t \in [0, 1/4]$ and $n \geq 2$. The remaining step is to show that the right hand side of the inequality is bounded above by $1 - \frac{\eta n\mu}{2}$ for $0 \leq \eta \leq \frac{1}{8nL}\min\{1, \sqrt{\frac{n}{\kappa}}\}$.

Define $z = 2\eta nL$. Using this, we have

$$1 - \eta n\mu + \frac{1}{2}(2\eta n\mu)^2 + \frac{2}{n}(2\eta nL)^3 \leq 1 - \frac{\eta n\mu}{2} \text{ for } 0 \leq \eta \leq \frac{1}{8nL}\min\left\{1, \sqrt{\frac{n}{\kappa}}\right\}$$

$$\Leftrightarrow g(z) := \frac{z}{4\kappa} - \frac{z^2}{2\kappa^2} - \frac{2z^3}{n} \geq 0 \text{ for } 0 \leq z \leq \frac{1}{4}\min\left\{1, \sqrt{\frac{n}{\kappa}}\right\},$$

so it suffices to show the latter. One can check that $g(0) = 0$, $g'(0) > 0$ and $g'(z)$ is monotonically decreasing in $z \geq 0$, so $g(z) \geq 0$ holds for $z \in [0, c]$ for some $c > 0$. This also means that if we have $g(c) \geq 0$ for some $c > 0$, $g(z) \geq 0$ for all $z \in [0, c]$.

First, consider the case $\kappa \leq n$. Then, $n/\kappa \geq 1$ and $\kappa \geq 1$, so

$$\frac{z}{4\kappa} - \frac{z^2}{2\kappa^2} - \frac{2z^3}{n} = \frac{1}{4\kappa}\left(z - \frac{2z^2}{\kappa} - \frac{8z^3}{n/\kappa}\right) \geq \frac{1}{4\kappa}\left(z - 2z^2 - 8z^3\right).$$

We can check that the function $z \mapsto z - 2z^2 - 8z^3$ is zero at $z = \frac{1}{4}$. This means that $g(z) \geq 0$ for $0 \leq z \leq \frac{1}{4}$.

Next, consider the case $\kappa \geq n$. In this case, set $z = c\sqrt{\frac{n}{\kappa}}$ where $c = \frac{1}{4}$. Then,

$$\frac{z}{4\kappa} - \frac{z^2}{2\kappa^2} - \frac{2z^3}{n} = \frac{1}{4\kappa}\left(c\sqrt{\frac{n}{\kappa}} - \frac{2c^2 n}{\kappa^2} - 8c^3\sqrt{\frac{n}{\kappa}}\right) \geq \frac{1}{4\kappa}\left((c - 8c^3)\sqrt{\frac{n}{\kappa}} - 2c^2\frac{n}{\kappa}\right).$$

Note that $\sqrt{\frac{n}{\kappa}} \leq 1$, and the function $t \mapsto (c - 8c^3)t - 2c^2t^2 = \frac{t}{8} - \frac{t^2}{8}$ is nonnegative on $[0, 1]$. Therefore, we have $g(\frac{1}{4}\sqrt{\frac{n}{\kappa}}) \geq 0$, so $g(z) \geq 0$ for $0 \leq z \leq \frac{1}{4}\sqrt{\frac{n}{\kappa}}$.

# D    Analysis of varying step sizes (Proofs of Theorems 3 and 4)

Throughout this section, since Theorems 3 and 4 assume the bounded iterates assumption (Assumption 1) and the $L$-smoothness of $f_i$'s, one can assume that $f_i$'s are Lipschitz continuous. In particular, one can assume that there exists $G > 0$ such that $\left\|\nabla f_i(\boldsymbol{x}_j^k)\right\| \leq G$ for all $i, j \in [n]$ and $k \geq 1$.

## D.1    Preliminaries: existing per-iteration/-epoch bounds

We first review the progress bounds for RANDOMSHUFFLE developed in Nagaraj, Jain, and Netrapalli [8], which are crucial for our varying step sizes analysis. Note that for RANDOMSHUFFLE, there are two different types of analyses:

1. Per-iteration analysis where one characterizes the progress made at each iteration.

2. Per-epoch analysis where one characterizes the aggregate progress made over one epoch.

For per-iteration analysis, [8] develops coupling arguments to prove that the progress made by RANDOMSHUFFLE is *not* worse than SGD. In particular, their coupling arguments demonstrate the closeness in expectation between the iterates of without- and with-replacement SGD. The following is a consequence of their coupling argument:

**Proposition D.1** (Per-iteration analysis [8, implicit in Section A.1])**.** *Assume for $L, G, \mu > 0$ that each component function $f_i$ is convex, $G$-Lipschitz and $L$-smooth and the cost function $F$ is $\mu$-strongly convex. Then, for any step size for the $(i + 1)$-th iteration of the $k$-th epoch such that $\eta_{i+1}^k \leq \frac{2}{L}$, the following bound holds between the adjacent iterates:*

$$\mathbb{E}\left\|\boldsymbol{x}_{i+1}^k - \boldsymbol{x}^*\right\|^2 \leq \left(1 - \eta_{i+1}^k\mu/2\right) \cdot \mathbb{E}\left\|\boldsymbol{x}_i^k - \boldsymbol{x}^*\right\|^2 + 3(\eta_{i+1}^k)^2 G^2 + 4(\eta_{i+1}^k)^3\kappa LG^2. \quad \text{(D.1)}$$

*where the expectation is taken over the randomness within the $k$-th epoch.*

However, with the above analysis, one can only obtain results comparable to SGD, as manifested in [8, Theorem 2]. In order to characterize better progress, one needs to characterize the aggregate progress made over one epoch as a whole:

**Proposition D.2** (Per-epoch analysis [8, implicit Section 5.1]). *Under the same setting as Proposition D.1, let $\eta_k \leq \frac{2}{L}$ be the step size for the $k$-th epoch, i.e., $\eta_i^k \equiv \eta_k$ for $i = 1, 2, \ldots, n$. Then, the following bound holds between the output of the $k$-th and $(k-1)$-th epochs $\boldsymbol{x}_0^{k+1}$ and $\boldsymbol{x}_0^k$:*

$$
\begin{aligned}
\mathbb{E}\big\|\boldsymbol{x}_0^{k+1} - \boldsymbol{x}^*\big\|^2 &\leq \left(1 - 3n\eta_k\mu/4 + n^2(\eta_k)^2 L^2\right) \cdot \big\|\boldsymbol{x}_0^k - \boldsymbol{x}^*\big\|^2 \\
&\quad - 2n\eta_k \left(1 - 4n\eta_k\kappa L\right) \cdot \left(\mathbb{E}F(\boldsymbol{x}_0^k) - F(\boldsymbol{x}^*)\right) + 20n^2(\eta_k)^3\kappa LG^2 + 5n^3(\eta_k)^4 L^2 G^2 \,.
\end{aligned}
\tag{D.2}
$$

*where the expectation is taken over the randomness within the $k$-th epoch.*

Having these per-iteration/-epoch progress bounds, the final ingredient of the non-asymptotic convergence rate analysis is to turn these bounds into *across-epochs* global convergence bounds.

## D.2   Chung's lemma: an analytic tool for varying stepsize

To illustrate our varying step sizes analysis, let us warm up with the per-iteration progress bound in Proposition D.1. Since Proposition D.1 works for any iterations, one can disregard the epoch structure and simply denote by $x_t$ the $t$-th iterate and by $\eta_t$ the step size used for the $t$-th iteration. Choosing $\eta_t = \frac{2\alpha}{\mu} \cdot \frac{1}{k_0+t}$ for all $t \geq 1$ with the initial index $k_0$, where we choose $k_0 = \alpha \cdot \kappa$ to ensure $\eta_t \leq \frac{2}{L}$, the per-iteration bound (D.1) becomes (we also use $(\eta_t)^3 \leq (\eta_t)^2 \frac{L}{2}$):

$$
\mathbb{E}\|\boldsymbol{x}_{t+1} - \boldsymbol{x}^*\|^2 \leq \left(1 - \frac{\alpha}{k_0 + t + 1}\right) \cdot \mathbb{E}\|\boldsymbol{x}_t - \boldsymbol{x}^*\|^2 + \frac{\alpha^2 G^2 (12\mu^{-2} + 32\kappa^3)}{(k_0 + t + 1)^2} \,.
\tag{D.3}
$$

In fact, for the bounds of type (D.3), there are suitable tools for obtaining convergence rates: versions of *Chung's lemma* [1], developed in the stochastic approximation literature. Among the various versions of Chung's lemma, there is a non-asymptotic version [1, Lemma 1]:

**Lemma D.3** (Non-asymptotic Chung's lemma). *Let $\{\xi_k\}_{k\geq 0}$ be a sequence of positive real numbers. Suppose that there exist an initial index $k_0 > 0$ and real numbers $A > 0$, $\alpha > \beta > 0$ such that $\xi_{k+1}$ satisfies the following inequality:*

$$
\xi_{k+1} \leq \exp\left(-\frac{\alpha}{k_0 + k + 1}\right) \xi_k + \frac{A}{(k_0 + k + 1)^{\beta+1}} \quad \text{for any } k \geq 0 \,.
\tag{D.4}
$$

*Then, for any $K \geq 1$ we have the following bound:*

$$
\xi_K \leq \exp\left(-\alpha \cdot \sum_{i=1}^{K} \frac{1}{k_0 + i}\right) \cdot \xi_0 + \frac{\frac{1}{\alpha-\beta} e^{\frac{\alpha}{k_0+1}} \cdot A}{(k_0 + K)^\beta} + \frac{e^{\frac{\alpha}{k_0+1}} \cdot A}{(k_0 + K)^{\beta+1}}
\tag{D.5}
$$

$$
\leq \frac{(k_0 + 1)^\alpha \cdot \xi_0}{(k_0 + K)^\alpha} + \frac{\frac{1}{\alpha-\beta} e^{\frac{\alpha}{k_0+1}} \cdot A}{(k_0 + K)^\beta} + \frac{e^{\frac{\alpha}{k_0+1}} \cdot A}{(k_0 + K)^{\beta+1}} \,.
\tag{D.6}
$$

**Proof**   Unfortunately, the original "proof" contains some errors as pointed out by Fabian [2, Discussion above Lemma 4.2]. We are able to correct the original proof; for this, see Section E.   □

Let us apply Lemma D.3 to (D.3) as a warm-up. From (D.3), one can see that $A$ in Lemma D.3 can be chosen as $G^2(12\mu^{-2} + 32\kappa^3)$. Hence, we obtain:

**Corollary D.4.** *Under the setting of Proposition D.1, let $\alpha > 1$ be a constant, and consider the step size $\eta_i^k = \frac{2\alpha/\mu}{k_0 + n(k-1)+i}$ for $k_0 := \alpha \cdot \kappa$. Then the following convergence rate holds for any $K \geq 1$:*

$$
\mathbb{E}\big\|\boldsymbol{x}_0^K - \boldsymbol{x}^*\big\|^2 \leq \frac{(k_0+1)^\alpha \|\boldsymbol{x}_0 - \boldsymbol{x}^*\|^2}{(k_0 + nK)^\alpha} + \frac{\frac{e}{\alpha-1}\alpha^2 G^2(12\mu^{-2} + 32\kappa^3)}{k_0 + nK} + \frac{e\alpha^2 G^2(12\mu^{-2} + 32\kappa^3)}{(k_0 + nK)^2} \,.
\tag{D.7}
$$

Notably, Corollary D.4 is an improvement over [8, Theorem 2] as it gets rid of extra poly-logarithmic terms.

## D.3 An illustrative failed attempt using Chung's lemma

Now, let us apply Lemma D.3 to the per-epoch progress bound (Proposition D.2). For illustrative purpose, consider an ideal situation where instead of the actual progress bound (D.2), a nicer epoch progress bound of the following form holds (that is to say, the coefficient of $\mathbb{E}\|y_k - \boldsymbol{x}^*\|^2$ does not contain the higher order error terms):

$$\mathbb{E}\big\|\boldsymbol{x}_0^{k+1} - \boldsymbol{x}^*\big\|^2 \leq (1 - n\eta_k\mu/2) \cdot \mathbb{E}\big\|\boldsymbol{x}_0^k - \boldsymbol{x}^*\big\|^2 + 20n^2(\eta_k)^3\kappa LG^2 + 5n^3(\eta_k)^4 L^2 G^2 \,. \quad \text{(D.8)}$$

Following the same principle as the previous section, let us take $\eta_k = \frac{2\alpha/\mu}{k_0 + nk}$ for some constant $\alpha > 2$. On the other hand, to make things simpler, let us assume that one can take $k_0 = 0$. Plugging this stepsize into (D.8), we obtain the following bound for some constants $c > 0$:

$$\mathbb{E}\big\|\boldsymbol{x}_0^{k+1} - \boldsymbol{x}^*\big\|^2 \leq \left(1 - \frac{\alpha}{k}\right) \cdot \mathbb{E}\big\|\boldsymbol{x}_0^k - \boldsymbol{x}^*\big\|^2 + \frac{c/n}{k^3}\,,$$

which then yields the following non-asymptotic bound due to Lemma D.3:

$$\mathbb{E}\big\|\boldsymbol{x}_0^{K+1} - \boldsymbol{x}^*\big\|^2 \leq O\left(\frac{1}{K^\alpha}\right) + O\left(\frac{1}{nK^2}\right) + O\left(\frac{1}{nK^3}\right)\,. \quad \text{(D.9)}$$

Although the last two terms in (D.9) are what we desire, the first term is undesirable. Even though we choose $\alpha$ large, this bound will still contain the term $O(1/K^\alpha)$ which does not match the rate in Theorem 3. Therefore, for the target convergence bound, one needs other versions of Lemma D.3.

## D.4 A variant of Chung's lemma

As we have seen in the previous section, Chung's lemma is not enough for capturing the desired convergence rate. In this section, to capture the right order for both $n$ and $K$, we develop a variant of Chung's lemma.

**Lemma D.5.** *Let $n > 0$ be an integer, and $\{\xi_k\}_{k \geq 0}$ be a sequence of positive real numbers. Suppose that there exist an initial index $k_0 > 0$ and real numbers $A_1, A_2 > 0$, $\alpha > \beta > 0$ and $\epsilon > 0$ such that the following are satisfied:*

$$\xi_1 \leq \exp\left(-\alpha \sum_{i=1}^n \frac{1}{k_0 + i}\right)\xi_0 + A_1 \quad and \quad \text{(D.10)}$$

$$\xi_{k+1} \leq \exp\left(-\alpha \sum_{i=1}^n \frac{1}{k_0 + nk + i} + \frac{\epsilon}{k^2}\right)\xi_k + \frac{A_2}{(k_0 + n(k+1))^{\beta+1}} \quad for\ any\ k \geq 1. \quad \text{(D.11)}$$

*Then, for any $K \geq 1$ we have the following bound for $c := e^{\epsilon\pi^2/6}$:*

$$\xi_K \leq \frac{c(k_0+1)^\alpha \cdot \xi_0}{(k_0 + nK)^\alpha} + \frac{c \cdot (k_0+n+1)^\alpha \cdot A_1}{(k_0+nK)^\alpha} + \frac{\frac{c}{\alpha-\beta}e^{\frac{\alpha}{k_0+n+1}} \cdot A_2}{n(k_0+nK)^\beta} + \frac{ce^{\frac{\alpha}{k_0+n+1}} \cdot A_2}{(k_0+nK)^{\beta+1}}\,. \quad \text{(D.12)}$$

**Proof**  See Section E.2.  □

## D.5 Sharper convergence rate for strongly convex costs (Proof of Theorem 3)

Now we use Lemma D.5 to obtain a sharper convergence rate. Let $\xi_k := \mathbb{E}\big\|\boldsymbol{x}_0^{k+1} - \boldsymbol{x}^*\big\|^2$ for $k \geq 1$ and $\xi_0 := \|\boldsymbol{x}_0 - \boldsymbol{x}^*\|^2$. Let $\alpha > 2$ be an arbitrarily chosen constant. For the first epoch, we take the following iteration-varying step size: $\eta_i^1 = \frac{2\alpha}{\mu} \cdot \frac{1}{k_0+i}$, where $k_0 = \alpha \cdot \kappa$ to ensure $\eta_i^1 \leq \frac{2}{L}$. Then, similarly to Corollary D.4, yet this time by using the bound (D.5) in Lemma D.3, one can derive the the following bound:

$$\xi_1 \leq \exp\left(-\alpha \cdot \sum_{i=1}^n \frac{1}{k_0 + i}\right) \cdot \xi_0 + \frac{a_1}{k_0 + n}\,, \quad \text{(D.13)}$$

where $a_1 := \alpha^2 G^2 \cdot [\frac{e}{\alpha-1}(12\mu^{-2} + 32\kappa^3) + e\alpha^2 G^2 (12\mu^{-1}L^{-1} + 32\kappa^2)]$, i.e., $a_1 = O\left(\kappa^3\right)$.

Next, let us establish bounds of the form (D.11) for the $k$-th epoch for $k \geq 2$. From the second epoch on, we use the same step size within an epoch. More specifically, for the $k$-th epoch we choose $\eta_{k,i} \equiv \eta_k = \frac{2\alpha/\mu}{k_0+nk}$. Let us recall the per-epoch progress bound from Proposition D.2:

$$
\begin{aligned}
\xi_k \leq{}& \left(1 - 3n\eta_k\mu/4 + n^2(\eta_k)^2 L^2\right) \cdot \xi_{k-1} \\
&- 2n\eta_k \left(1 - 4n\eta_k\kappa L\right) \cdot (\mathbb{E}F(\boldsymbol{x}_0^k) - F(\boldsymbol{x}^*)) + 20n^2(\eta_k)^3\kappa LG^2 + 5n^3(\eta_k)^4 L^2 G^2 \,.
\end{aligned}
\tag{D.14}
$$

Since $\mathbb{E}F(\boldsymbol{x}_0^k) - F(\boldsymbol{x}^*) > 0$, one can disregard the second term in the upper bound (D.2) as long as $4n\eta_k\kappa L < 1$. If $\eta_k$ small enough that $4n\eta_k\kappa L < 1$ holds, then since we also have $\frac{n\eta_k\mu}{4} > n^2(\eta_k)^2 L^2$, the per-epoch bound (D.2) becomes:

$$
\xi_k \leq (1 - n\eta_k\mu/2)\,\xi_{k-1} + 20n^2(\eta_k)^3\kappa LG^2 + 5n^3(\eta_k)^4 L^2 G^2 \,.
\tag{D.15}
$$

$$
\leq \exp\left(-n\eta_k\mu/2\right) \cdot \xi_k + 20n^2(\eta_k)^3\kappa LG^2 + 5n^3(\eta_k)^4 L^2 G^2 \,.
\tag{D.16}
$$

Since $4n\eta_k\kappa L < 1$ is fulfilled for $k \geq 8\alpha\kappa^2$ (note that for $k \geq 8\alpha\kappa^2$, $nk > 8\alpha\kappa^2 n = (2\alpha/\mu) \cdot 4n\kappa L$), we conclude that (D.16) holds for $k \geq 8\alpha\kappa^2$.

For $k < 8\alpha\kappa^2$, recursively applying Proposition D.1 with the fact $(n\eta_k)^{-1} \leq 4\kappa L + L/(2n)$ implies:

$$
\xi_k \leq \exp\left(-n\eta_k\mu/2\right) \cdot \xi_{k-1} + 3n^2(\eta_k)^3 G^2(4\kappa L + L/(2n)) + 4n(\eta_k)^3\kappa LG^2 \,.
\tag{D.17}
$$

Therefore, combining (D.16) and (D.17), we obtain the following bound which holds for any $k \geq 2$:

$$
\xi_k \leq \exp\left(-n\eta_k\mu/2\right) \cdot \xi_{k-1} + a_2 \cdot n^2(\eta_k)^3 \,,
\tag{D.18}
$$

where $a_2 := 12\kappa LG^2 + (3L/2 + 4\kappa LG^2)/n + 20\kappa LG^2 + 5\mu^2 G^2/8$, i.e., $a_2 = O\left(\kappa\right)$. Let us modify the coefficient of $\xi_k$ in (D.18) so that it fits into the form of (D.11) in Lemma D.5. First note that $\exp\left(-n\eta_k\mu/2\right) = \exp\left(-\alpha n/(k_0 + nk)\right)$. Now, this expression can be modified as

$$
\exp\left[-\alpha \cdot \sum_{i=1}^{n} \frac{1}{k_0 + n(k-1) + i} + \alpha \cdot \sum_{i=1}^{n}\left(\frac{1}{k_0 + n(k-1) + i} - \frac{1}{k_0 + nk}\right)\right] \,,
$$

which is then upper bounded by $\exp\left[-\alpha \cdot \sum_{i=1}^{n} \frac{1}{k_0+n(k-1)+i} + \frac{\alpha}{(k-1)^2}\right]$. Thus, (D.18) can be rewritten as:

$$
\xi_k \leq \exp\left(-\alpha \cdot \sum_{i=1}^{n} \frac{1}{k_0 + n(k-1) + i} + \frac{\alpha}{(k-1)^2}\right) \cdot \xi_{k-1} + \frac{8a_2\alpha^3 n^2\mu^{-3}}{(k_0 + nk)^3} \,.
\tag{D.19}
$$

Now applying Lemma D.5 with (D.13) and (D.19) implies the following result:

### D.6 Sharper convergence rate for quadratic costs (Proof of Theorem 4)

Now let us use again Lemma D.5 to obtain a sharper convergence rate. We follow the notations in Section D.5. Again, we use the following bound (which we derived in (D.13) in the main text) for the first recursive inequality (D.10) in Lemma D.5:

$$
\xi_1 \leq \exp\left(-\alpha \cdot \sum_{i=1}^{n} \frac{1}{k_0+i}\right) \cdot \xi_0 + \frac{a_1}{k_0+n} \,,
$$

where $a_1 := \alpha^2 G^2 \cdot [\frac{e}{\alpha-1}(12\mu^{-2} + 32\kappa^3) + e\alpha^2 G^2 (12\mu^{-1}L^{-1} + 32\kappa^2)]$, i.e., $a_1 = O\left(\kappa^3\right)$.

For the second recursive inequalities (D.11) in Lemma D.5, in order to obtain better convergence rate, we use the following improved per-epoch bound for quadratic costs due to Rajput, Gupta, and Papailiopoulos [10]:

**Proposition D.6** ([10], implicit in Appendix A). *Under the setting of Proposition D.1, assume further that $F$ is quadratic. Then for any step size for the $k$-th epoch $\eta_k \leq \frac{2}{L}$, the following bound holds between the output of the $k$-th and $k-1$-th epochs $\boldsymbol{x}_0^{k+1}$ and $\boldsymbol{x}_0^k$:*

$$
\begin{aligned}
\mathbb{E}\left\|\boldsymbol{x}_0^{k+1} - \boldsymbol{x}^*\right\|^2 \leq{}& \left(1 - 3n\eta_k\mu/2 + 5n^2(\eta_k)^2 L^2 + 8n^3(\eta_k)^3\kappa L^3\right) \left\|\boldsymbol{x}_0^k - \boldsymbol{x}^*\right\|^2 \\
&+ 10n^3(\eta_k)^4 L^2 G^2 + 40n^4(\eta_k)^5\kappa L^3 G^2 + 32n(\eta_k)^3\kappa LG^2 \,.
\end{aligned}
\tag{D.20}
$$

*where the expectation is taken over the randomness within the $k$-th epoch.*

For $k > 16\alpha\kappa^2$, we have $n\eta_k < \frac{1}{8}\frac{\mu}{L^2}$. Using this bound, it is straightforward to check that (D.20) can be simplified into:

$$\xi_k \leq \exp\left(-n\eta_k\mu/2\right)\xi_{k-1} + 15n^3(\eta_k)^4 L^2 G^2 + 32n(\eta_k)^3\kappa LG^2. \tag{D.21}$$

For $k < 16\alpha\kappa^2$, recursively applying Proposition D.1 with the fact $(n\eta_{k+1})^{-1} \leq 8\kappa L + L/(2n)$ implies:

$$\xi_k \leq \exp\left(-n\eta_k\mu/2\right)\cdot\xi_k + 3n^3(\eta_k)^4 G^2(8\kappa L + L/(2n))^2 + 4n(\eta_k)^3\kappa LG^2. \tag{D.22}$$

Therefore, combining (D.21) and (D.22), we obtain the following bound which holds for any $k \geq 1$:

$$\xi_k \leq \exp\left(-n\eta_k\mu/2\right)\cdot\xi_{k-1} + b_2\cdot n^3(\eta_k)^4 + b_3\cdot n(\eta_k)^3, \tag{D.23}$$

where $b_2 := 15L^2G^2 + 3G^2(8\kappa L + L/(2n))^2$ and $b_3 := 32\kappa LG^2$, i.e., $b_2 = O(\kappa^2)$ and $b_3 = O(\kappa)$. Following Section D.4, one can similarly modify the coefficient of $\xi_k$ in (D.23) to obtain the following for $k \geq 2$:

$$\xi_k \leq \exp\left(-\alpha\cdot\sum_{i=1}^{n}\frac{1}{k_0 + n(k-1) + i} + \frac{\alpha}{(k-1)^2}\right)\cdot\xi_k + \frac{16b_2\alpha^4 n^3\mu^{-4}}{(k_0 + nk)^4} + \frac{8b_3\alpha^3 n\mu^{-3}}{(k_0 + nk)^3} \tag{D.24}$$

However, one can notice that (D.24) is not quite of the form (D.11), and Lemma D.5 is not directly applicable to this bound. In fact, we need to make some modifications in Lemma D.5. First, for $A_3 > 0$ and $\gamma > 0$, there is an additional term to the recursive relations (D.11): for any $k \geq 1$, the new recursive relations now read

$$\xi_{k+1} \leq \exp\left(-\alpha\sum_{i=1}^{n}\frac{1}{k_0 + nk + i} + \frac{\epsilon}{k^2}\right)\xi_k + \frac{A_2}{(k_0 + n(k+1))^{\beta+1}} + \frac{A_3}{(k_0 + n(k+1))^{\gamma+1}}. \tag{D.25}$$

It turns out that for these additional terms in the recursive relations, one can use the same techniques to prove that the corresponding global convergence bound (D.12) has the following additional terms:

$$\frac{\frac{c}{\alpha-\beta}e^{\frac{\alpha}{k_0+n+1}}\cdot A_3}{n(k_0 + nK)^\gamma} + \frac{ce^{\frac{\alpha}{k_0+n+1}}\cdot A_3}{(k_0 + nK)^{\gamma+1}}. \tag{D.26}$$

Now using this modified version of Lemma D.5, the proof is completed.

# E  Proofs of the versions of Chung's lemma (Lemmas D.3 and D.5)

We begin by introducing an elementary fact that we will use throughout the proofs:

**Proposition E.1** (Integral approximation; see e.g. [6, Theorem 14.3])**.** *Let* $f : \mathrm{e}^+ \to \mathrm{e}^+$ *be a non-decreasing continuous function. Then, for any integers* $1 \leq m < n$, $\int_m^n f(x)dx + f(m) \leq \sum_{i=m}^n f(i) \leq \int_m^n f(x)dx + f(n)$*. Similarly, if* $f$ *is non-increasing, then for any integers* $1 \leq m < n$, $\int_m^n f(x)dx + f(n) \leq \sum_{i=m}^n f(i) \leq \int_m^n f(x)dx + f(m)$*.*

We first prove Lemma D.3, and hence proving the non-asymptotic Chung's lemma [1, Lemma 1] which has an incorrect original proof.

## E.1  A correct proof of Chung's lemma (Proof of Lemma D.3)

We first restate the lemma for reader's convenience:

**Lemma E.2** (Restatement from Section D.2)**.** *Let* $\{\xi_k\}_{k\geq 0}$ *be a sequence of positive real numbers. Suppose that there exist an initial index* $k_0 > 0$ *and real numbers* $A > 0$, $\alpha > \beta > 0$ *such that* $\xi_{k+1}$ *satisfies the following inequality:*

$$\xi_{k+1} \leq \exp\left(-\frac{\alpha}{k_0 + k + 1}\right)\xi_k + \frac{A}{(k_0 + k + 1)^{\beta+1}} \quad \text{for any } k \geq 0. \tag{E.1}$$

*Then, for any $K \geq 1$ we have the following bound:*

$$\xi_K \leq \exp\left(-\alpha \cdot \sum_{i=1}^{K} \frac{1}{k_0 + i}\right) \cdot \xi_0 + \frac{\frac{1}{\alpha-\beta} e^{\frac{\alpha}{k_0+1}} \cdot A}{(k_0 + K)^\beta} + \frac{e^{\frac{\alpha}{k_0+1}} \cdot A}{(k_0 + K)^{\beta+1}} \tag{E.2}$$

$$\leq \frac{(k_0 + 1)^\alpha \cdot \xi_0}{(k_0 + K)^\alpha} + \frac{\frac{1}{\alpha-\beta} e^{\frac{\alpha}{k_0+1}} \cdot A}{(k_0 + K)^\beta} + \frac{e^{\frac{\alpha}{k_0+1}} \cdot A}{(k_0 + K)^{\beta+1}} . \tag{E.3}$$

For simplicity, let us define the following quantities for $k \geq 1$:

$$a_k := \exp\left(-\frac{\alpha}{k_0 + k}\right) \quad \text{and} \quad c_k := \frac{A}{(k_0 + k)^{\beta+1}} .$$

Using these notations, the recursive relation (E.1) becomes:

$$\xi_{k+1} \leq a_{k+1} \cdot \xi_k + c_{k+1} \quad \text{for any integer } k \geq 1. \tag{E.4}$$

After recursively applying (E.4) for $k = 0, 1, 2, \ldots, K-1$, one obtains the following bound:

$$\xi_K \leq \xi_0 \prod_{j=1}^{K} a_j + \left(\prod_{j=1}^{K} a_j\right) \cdot \left[\sum_{k=1}^{K} \left(\prod_{j=1}^{k} a_j\right)^{-1} c_k\right] . \tag{E.5}$$

Now let us upper and lower bound the product of $a_j$'s. Note that

$$\prod_{j=1}^{k} a_j = \exp\left(-\alpha \sum_{i=1}^{k} \frac{1}{k_0 + i}\right) \quad \text{for any } k \geq 1.$$

Using Proposition E.1 with $f(x) = \frac{1}{k_0 + x}$, we get

$$\log \frac{k_0 + k}{k_0 + 1} \leq \sum_{i=1}^{k} \frac{1}{k_0 + i} \leq \log \frac{k_0 + k}{k_0 + 1} + \frac{1}{k_0 + 1} .$$

Using these upper and lower bounds, one can conclude:

$$e^{-\frac{\alpha}{k_0+1}} \left(\frac{k_0 + 1}{k_0 + k}\right)^\alpha \leq \prod_{j=1}^{k} a_j \leq \left(\frac{k_0 + 1}{k_0 + k}\right)^\alpha . \tag{E.6}$$

Therefore, we have

$$\sum_{k=1}^{K} \left(\prod_{j=1}^{k} a_j\right)^{-1} c_k \leq e^{\frac{\alpha}{k_0+1}} \sum_{k=1}^{K} \left(\frac{k_0 + k}{k_0 + 1}\right)^\alpha \cdot \frac{A}{(k_0 + k)^{\beta+1}} = \frac{e^{\frac{\alpha}{k_0+1}} \cdot A}{(k_0 + 1)^\alpha} \cdot \sum_{k=1}^{K} (k_0 + k)^{\alpha-\beta-1} .$$

Applying Proposition E.1 with $f(x) = (k_0 + x)^{\alpha-\beta-1}$ to the above, since $\frac{1}{\alpha-\beta}(k_0 + x)^{\alpha-\beta}$ is an anti-derivative of $f$, we obtain the following upper bounds:

$$\frac{e^{\frac{\alpha}{k_0+1}} \cdot A}{(k_0 + 1)^\alpha} \cdot \begin{cases} \frac{1}{\alpha-\beta}\left((k_0 + K)^{\alpha-\beta} - (k_0 + 1)^{\alpha-\beta}\right) + (k_0 + K)^{\alpha-\beta-1}, & \text{if } \alpha > \beta + 1, \\ K, & \text{if } \alpha = \beta + 1, \\ \frac{1}{\alpha-\beta}\left((k_0 + K)^{\alpha-\beta} - (k_0 + 1)^{\alpha-\beta}\right) + (k_0 + 1)^{\alpha-\beta-1} & \text{if } \alpha < \beta + 1. \end{cases}$$

Combining all three cases, we conclude:

$$\sum_{k=1}^{K} \left(\prod_{j=1}^{k} a_j\right)^{-1} c_k \leq \frac{e^{\frac{\alpha}{k_0+1}} \cdot A}{(k_0 + 1)^\alpha} \cdot \left(\frac{(k_0 + K)^{\alpha-\beta}}{\alpha - \beta} + (k_0 + K)^{\alpha-\beta-1}\right) . \tag{E.7}$$

Indeed, for the cases $\alpha > \beta - 1$ and $\alpha = \beta + 1$, the above upper bound follows immediately; for the case $\alpha < \beta + 1$, note (from the assumption $\alpha > \beta$) that $\alpha - \beta \in (0, 1)$, which implies

$-\frac{1}{\alpha-\beta}(k_0+1)^{\alpha-\beta}+(k_0+1)^{\alpha-\beta-1} < -(k_0+1)^{\alpha-\beta}+(k_0+1)^{\alpha-\beta-1} = -(k_0+1)^{\alpha-\beta-1} \cdot k_0 < 0,$
which then implies the desired upper bound.

Plugging (E.7) back to (E.5) and using (E.6) to upper bound $\prod_{j=1}^{K} a_j$, we obtain:

$$\xi_K \le \xi_0 \prod_{j=1}^{K} a_j + \left(\prod_{j=1}^{K} a_j\right) \cdot \frac{e^{\frac{\alpha}{k_0+1}} \cdot A}{(k_0+1)^\alpha} \cdot \left(\frac{(k_0+K)^{\alpha-\beta}}{\alpha-\beta} + (k_0+K)^{\alpha-\beta-1}\right)$$

$$\le \xi_0 \prod_{j=1}^{K} a_j + \left(\frac{k_0+1}{k_0+K}\right)^\alpha \cdot \frac{e^{\frac{\alpha}{k_0+1}} \cdot A}{(k_0+1)^\alpha} \cdot \left(\frac{(k_0+K)^{\alpha-\beta}}{\alpha-\beta} + (k_0+K)^{\alpha-\beta-1}\right)$$

$$\le \exp\left(-\alpha \cdot \sum_{i=1}^{K} \frac{1}{k_0+i}\right) \cdot \xi_0 + \frac{\frac{1}{\alpha-\beta} e^{\frac{\alpha}{k_0+1}} \cdot A}{(k_0+K)^\beta} + \frac{e^{\frac{\alpha}{k_0+1}} \cdot A}{(k_0+K)^{\beta+1}},$$

which is precisely (E.2). Using (E.6) once again to upper bound the term $\exp(-\alpha \cdot \sum_{i=1}^{K} \frac{1}{k_0+i})$, we obtain (E.3), which completes the proof.

## E.2 Proof of Lemma D.5

We first restate the lemma for reader's convenience:

**Lemma E.3** (Restatement from Section D.4). *Let $n > 0$ be an integer, and $\{\xi_k\}_{k\ge 0}$ be a sequence of positive real numbers. Suppose that there exist an initial index $k_0 > 0$ and real numbers $A_1, A_2 > 0$, $\alpha > \beta > 0$ and $\epsilon > 0$ such that the following are satisfied:*

$$\xi_1 \le \exp\left(-\alpha \sum_{i=1}^{n} \frac{1}{k_0+i}\right)\xi_0 + A_1 \quad and \tag{E.8}$$

$$\xi_{k+1} \le \exp\left(-\alpha \sum_{i=1}^{n} \frac{1}{k_0+nk+i} + \frac{\epsilon}{k^2}\right)\xi_k + \frac{A_2}{(k_0+n(k+1))^{\beta+1}} \quad for\ any\ k \ge 1. \tag{E.9}$$

*Then, for any $K \ge 1$ we have the following bound for $c := e^{\epsilon\pi^2/6}$:*

$$\xi_K \le \frac{c(k_0+1)^\alpha \cdot \xi_0}{(k_0+nK)^\alpha} + \frac{c \cdot (k_0+n+1)^\alpha \cdot A_1}{(k_0+nK)^\alpha} + \frac{\frac{c}{\alpha-\beta} e^{\frac{\alpha}{k_0+n+1}} \cdot A_2}{n(k_0+nK)^\beta} + \frac{ce^{\frac{\alpha}{k_0+n+1}} \cdot A_2}{(k_0+nK)^{\beta+1}}. \tag{E.10}$$

The proof is generally analogous to that of Lemma D.3, while some distinctions are required so that the final bound captures the desired dependencies on the two parameters $n$ and $K$. To simplify notations, let us define the following quantities for $k \ge 1$:

$$a_k := \exp\left(-\alpha \cdot \sum_{i=1}^{n} \frac{1}{k_0+n(k-1)+i}\right), \quad b_k := \exp\left(\frac{\epsilon}{(k-1)^2}\right), \quad and \quad c_k := \frac{A_2}{(k_0+nk)^{\beta+1}}.$$

Using these notations, the recursive relations (D.10) and (D.11) become:

$$\xi_1 \le a_1 \cdot \xi_0 + A_1 \tag{E.11}$$
$$\xi_{k+1} \le a_{k+1}b_{k+1} \cdot \xi_k + c_{k+1} \quad for\ any\ integer\ k \ge 1. \tag{E.12}$$

Recursively applying (E.12) for $k = 1, 2, \ldots, K-1$ and then (E.11), we obtain:

$$\xi_K \le a_1\xi_0 \prod_{j=2}^{K} a_jb_j + \left(\prod_{j=2}^{K} a_jb_j\right) \cdot \left[A_1 + \sum_{k=2}^{K} \left(\prod_{j=2}^{k} a_jb_j\right)^{-1} c_k\right]. \tag{E.13}$$

Note taht from the fact $\sum_{i\ge 1} i^{-2} = \frac{\pi^2}{6}$, one can upper and lower bound the product of $b_j$'s:

$$1 \le \prod_{i=2}^{K} b_i \le \exp\left(\sum_{i=2}^{K} \frac{\epsilon}{(i-1)^2}\right) \le \exp\left(\epsilon\pi^2/6\right). \tag{E.14}$$

Applying (E.14) to (E.13), we obtain the following bound (recall that $c := e^{\epsilon \pi^2/6}$):

$$\xi_K \le c\xi_0 \prod_{j=1}^{K} a_j + c \prod_{j=2}^{K} a_j \cdot \left[ A_1 + \sum_{k=2}^{K} \left( \prod_{j=2}^{k} a_j \right)^{-1} c_k \right]. \tag{E.15}$$

To obtain upper and lower bounds on the product of $a_j$'s, again note that for any $2 \le k$,

$$\prod_{j=2}^{k} a_j = \exp \left( -\alpha \cdot \sum_{i=1}^{(k-1)n} \frac{1}{k_0 + n + i} \right),$$

which can then be estimated as follows using Proposition E.1 similarly to (E.6):

$$e^{-\frac{\alpha}{k_0+n+1}} \left( \frac{k_0 + n + 1}{k_0 + nk} \right)^{\alpha} \le \prod_{j=2}^{k} a_j \le \left( \frac{k_0 + n + 1}{k_0 + nk} \right)^{\alpha}. \tag{E.16}$$

Therefore, we have

$$\sum_{k=2}^{K} \left( \prod_{j=2}^{k} a_j \right)^{-1} c_k \le e^{\frac{\alpha}{k_0+n+1}} \sum_{k=2}^{K} \left( \frac{k_0 + nk}{k_0 + n + 1} \right)^{\alpha} \cdot \frac{A_2}{(k_0 + nk)^{\beta+1}}$$

$$= \frac{e^{\frac{\alpha}{k_0+n+1}} \cdot A_2}{(k_0 + n + 1)^{\alpha}} \cdot \sum_{k=2}^{K} (k_0 + nk)^{\alpha-\beta-1}.$$

Applying Proposition E.1 with $f(x) = (k_0 + nx)^{\alpha-\beta-1}$ to the above, since $\frac{1}{n(\alpha-\beta)}(k_0 + nx)^{\alpha-\beta}$ is an anti-derivative of $f$, we obtain the following upper bounds:

$$\frac{e^{\frac{\alpha}{k_0+n+1}} \cdot A_2}{(k_0 + n + 1)^{\alpha}} \cdot \begin{cases} \frac{1}{n(\alpha-\beta)} \left( (k_0 + nK)^{\alpha-\beta} - (k_0 + 2n)^{\alpha-\beta} \right) + (k_0 + nK)^{\alpha-\beta-1}, & \text{if } \alpha > \beta + 1, \\ K - 1, & \text{if } \alpha = \beta + 1, \\ \frac{1}{n(\alpha-\beta)} \left( (k_0 + nK)^{\alpha-\beta} - (k_0 + 2n)^{\alpha-\beta} \right) + (k_0 + 2n)^{\alpha-\beta-1} & \text{if } \alpha < \beta + 1. \end{cases}$$

Akin to (E.7), one can combining all three cases and conclude:

$$\sum_{k=2}^{K} \left( \prod_{j=2}^{k} a_j \right)^{-1} c_k \le \frac{e^{\frac{\alpha}{k_0+n+1}} \cdot A_2}{(k_0 + n + 1)^{\alpha}} \cdot \left( \frac{(k_0 + nK)^{\alpha-\beta}}{n(\alpha - \beta)} + (k_0 + nK)^{\alpha-\beta-1} \right).$$

Plugging this back to (E.15), and using (E.16) to upper bound the product of $a_j$'s, we obtain:

$$\xi_K \le c\xi_0 \prod_{j=1}^{K} a_j + c \prod_{j=2}^{K} a_j \cdot \left[ A_1 + \frac{e^{\frac{\alpha}{k_0+n+1}} \cdot A_2}{(k_0 + n + 1)^{\alpha}} \cdot \left( \frac{(k_0 + nK)^{\alpha-\beta}}{n(\alpha - \beta)} + (k_0 + nK)^{\alpha-\beta-1} \right) \right]$$

$$\le c\xi_0 \prod_{j=1}^{K} a_j + c \left( \frac{k_0 + n + 1}{k_0 + nK} \right)^{\alpha} \cdot \left[ A_1 + \frac{e^{\frac{\alpha}{k_0+n+1}} \cdot A_2}{(k_0 + n + 1)^{\alpha}} \cdot \left( \frac{(k_0 + nK)^{\alpha-\beta}}{n(\alpha - \beta)} + (k_0 + nK)^{\alpha-\beta-1} \right) \right]$$

$$= c\xi_0 \prod_{j=1}^{K} a_j + \frac{c \cdot (k_0 + n + 1)^{\alpha} \cdot A_1}{(k_0 + nK)^{\alpha}} + \frac{\frac{c}{\alpha-\beta} e^{\frac{\alpha}{k_0+n+1}} \cdot A_2}{n(k_0 + nK)^{\beta}} + \frac{ce^{\frac{\alpha}{k_0+n+1}} \cdot A_2}{(k_0 + nK)^{\beta+1}}.$$

Now similarly to (E.16), one obtains the upper bound $\prod_{j=1}^{K} a_j \le \left( \frac{k_0+n+1}{k_0+nK} \right)^{\alpha}$, which together with the last expression deduces (E.10) and hence completes the proof.

## F  Tight convergence bound for SINGLESHUFFLE

In this section, we provide a tight convergence bound for SINGLESHUFFLE on smooth strongly convex functions, which also holds for strongly convex quadratic functions.

**Theorem F.1** (Strongly convex costs). *Assume that $F$ is $\mu$-strongly convex and each $f_i \in C_L^1(\mathbb{R}^d)$. Consider* SINGLESHUFFLE *for the number of epochs $K$ satisfying $K \geq 10\kappa^2 \log(n^{1/2}K)$, step size $\eta_i^k = \eta := \frac{2 \log(n^{1/2}K)}{\mu n K}$, and initialization $\boldsymbol{x}_0$. Then for $G := \max_{i \in [n]} \|\nabla f_i(\boldsymbol{x}^*)\|$ and some constant $c = O(\kappa^4)$,*

$$\mathbb{E}[F(\boldsymbol{x}_0^{K+1})] - F^* \leq \frac{2L \|\boldsymbol{x}_0 - \boldsymbol{x}^*\|^2}{nK^2} + \frac{c \cdot G^2 \cdot \log^3(nK)}{nK^2} \, .$$

**Proof:** The proof technique builds on the proof of Theorem 2, using the idea of the end-to-end analysis from [12]. See the subsequent subsections for the full proof. □

**Optimality of convergence rate.** Theorem F.1 provides a tight (up to poly-log factors) bound that matches the known lower bound $\Omega\left(1/nK^2\right)$ [12], which was proven for strongly convex quadratic functions. Since Theorem F.1 applies to subclasses of smooth strongly convex functions, it also gives the minimax optimal rate (up to log factors) for strongly convex quadratic functions (see Table A). Note that the theorem does *not* require the convexity of component functions or bounded iterates assumption (Assumption 1), in the same spirit as our RANDOMSHUFFLE results (Theorems 1 and 2).

**Remark F.1** (RANDOMSHUFFLE v.s. SINGLESHUFFLE). It is often conjectured that RANDOMSHUFFLE performs better than SINGLESHUFFLE due to multiple shuffling. The class of strongly convex quadratic functions aligns with this intuition, because there is a gap between optimal convergence rates $\widetilde{\mathcal{O}}(1/(nK)^2 + 1/nK^3)$ (RANDOMSHUFFLE) and $\widetilde{\mathcal{O}}(1/nK^2)$ (SINGLESHUFFLE) for quadratic functions. In contrast, for a broader class of smooth strongly convex functions, Theorems 1 and F.1 reveal a rather surprising fact: the optimal rates of RANDOMSHUFFLE and SINGLESHUFFLE have the same dependence on $n$ and $K$. Although Theorem F.1 shows the same rate in $n$ and $K$ as Theorem 1, we note that its epoch requirement $K \gtrsim \kappa^2 \log(n^{1/2}K)$ is worse than Theorem 1 by a factor of $\kappa$; however, it matches the epoch requirement of the existing bound for RANDOMSHUFFLE [8].

**Remark F.2** (Proof techniques). The Hoeffding-Serfling inequality used in the proof of Theorem 1 for RANDOMSHUFFLE requires that the vectors $\nabla f_i(\boldsymbol{x}_0^k)$'s, to which we apply the Hoeffding-Serfling inequality, have to be independent of the permutation $\sigma_k$. This is no longer true for SINGLESHUFFLE, because in SINGLESHUFFLE, once a permutation $\sigma$ is fixed, it is used over and over again. The iterates become dependent on the choice of $\sigma$, hence rendering a direct extension of Theorem 1 to SINGLESHUFFLE impossible. For the proof of Theorem F.1, we instead take an end-to-end approach following [12]. Taking this approach, we apply the Hoeffding-Serfling inequality to the vectors $\nabla f_i(\boldsymbol{x}^*)$'s, i.e., gradients evaluated at the global minimum $\boldsymbol{x}^*$, which are independent of permutations sampled by the algorithm. This way, we can prove a bound for SINGLESHUFFLE. In fact, this proof technique can be easily extended to any reshuffling schemes that lie between RANDOMSHUFFLE and SINGLESHUFFLE, modulo some additional union bounds. For instance, our proof can be extended to the scheme where the components are reshuffled every 5 epochs.

**Remark F.3** (Possible improvements for quadratics). Notice that if the component functions $f_i$'s are quadratic, then their Hessians are constant, which implies that the matrix $\boldsymbol{S}_k$ (B.2) that appears in the update equation of RANDOMSHUFFLE is now constant ($\boldsymbol{S}_k = \boldsymbol{S}$) over epochs of SINGLESHUFFLE. We believe that leveraging this fact could lead to a tighter epoch requirement than Theorem F.1. However, proving such an epoch requirement demands deriving a contraction bound that is more involved than the ones proven for RANDOMSHUFFLE (e.g., Lemma B.1), because one has to now bound $\left\|\mathbb{E}[(\boldsymbol{S}^K)^T \boldsymbol{S}^K]\right\|$, in place of $\left\|\mathbb{E}[\boldsymbol{S}^T \boldsymbol{S}]\right\|$. We leave this refinement for future work.

## F.1 Proof outline

The proof of Theorem F.1 builds on the proof of Theorem 2 presented in Section B. We first recursively apply the update equations over all iterations and obtain an equation that expresses the last iterate $\boldsymbol{x}_0^{K+1}$ in terms of the initialization $\boldsymbol{x}_0^1 = \boldsymbol{x}_0$. By proving new lemmas in a similar flavor to the ones developed in Section B, we will bound $\mathbb{E}[\|\boldsymbol{x}_0^{K+1} - \boldsymbol{x}^*\|^2]$ to get our desired result.

Since the algorithm is SINGLESHUFFLE, we fix the permutation $\sigma$ and use it for all epochs. If the component functions $f_i$'s were quadratic functions as in Theorem 2, $\boldsymbol{S}_k$ and $\boldsymbol{t}_k$ (B.2) defined in the proof of Theorem 2 would have been *constant* over epochs of SINGLESHUFFLE, given the choice of $\sigma$; however, this is *not* true in the non-quadratic case, because the Hessians of $f_i$'s are not constant. We have to take this into account in the proof.

Table A: A summary of existing convergence rates and our results for SINGLESHUFFLE. All the convergence rates are with respect to the suboptimality of objective function value. Note that since the function classes become more restrictive as we go down the table, the noted lower bounds are also valid for upper rows, and the upper bounds are also valid for lower rows. In the "Assumptions" column, inequalities such as $K \gtrsim \kappa^\alpha$ mark the requirements $K \geq C\kappa^\alpha \log(nK)$ for the bounds to hold, and (A1) denotes the assumption that all the iterates remain in a bounded set (see Assumption 1). Also, (LB) stands for "lower bound."

| Convergence rates for SINGLESHUFFLE | | | | |
|---|---|---|---|---|
| Settings | | References | Convergence rates | Assumptions |
| (1) $F$ PŁ | $f_i$ smooth Lipschitz | Nguyen et al. [9] | $O\left(\frac{1}{K^2}\right)$ | $K \geq 1$ |
| | | Safran and Shamir [12] | $\Omega\left(\frac{1}{nK^2}\right)$ (LB) | const. step size |
| (2) $F$ strongly convex | $f_i$ smooth | Nguyen et al. [9] | $O\left(\frac{1}{K^2}\right)$ | $K \geq 1$ |
| | | **Ours** (Thm F.1) | $O\left(\frac{\log^3(nK)}{nK^2}\right)$ | $K \gtrsim \kappa^2$ |
| | $f_i$ smooth convex | Gürbüzbalaban et al. [3] | $O\left(\frac{1}{K^2}\right)$ | asymptotic & (A1) |
| | | Mishchenko et al. [7]$^\dagger$ | $O\left(e^{-\frac{nK}{\kappa}} + \frac{\log^2(nK)}{nK^2}\right)$ | $K \geq 1$ |
| | | Safran and Shamir [12] | $\Omega\left(\frac{1}{nK^2}\right)$ (LB) | const. step size |
| (3) $F$ strongly convex quadratic | $f_i$ smooth | **Ours** (Thm F.1) | $O\left(\frac{\log^3(nK)}{nK^2}\right)$ | $K \gtrsim \kappa^2$ |
| | $f_i$ smooth quadratic convex | Gürbüzbalaban et al. [3]$^\ddagger$ | $O\left(\frac{1}{K^2}\right)$ | asymptotic |
| | | Safran and Shamir [12] | $O\left(\frac{\log^4(nK)}{nK^2}\right)$ | $d=1$, $K \gtrsim \kappa/n$ |
| | | Safran and Shamir [12] | $\Omega\left(\frac{1}{nK^2}\right)$ (LB) | const. step size |

$^\dagger$ additionally requires $\mu$-*strong convexity* of $f_i$'s.
$^\ddagger$ does not require that $f_i$'s are convex.

Throughout the proof, we assume without loss of generality that the global minimum is achieved at $\boldsymbol{x}^* = \boldsymbol{0}$. That is, $\sum_{i=1}^n \nabla f_i(\boldsymbol{0}) = \boldsymbol{0}$. We define $G := \max_{i \in [n]} \|\nabla f_i(\boldsymbol{0})\|$.

We first decompose the gradient estimate $\nabla f_{\sigma(i)}(\boldsymbol{x}_{i-1}^k)$ at the $i$-th iteration of the $k$-th epoch ($i \in [n], k \in [K]$) into a sum of three different parts:

$$\nabla f_{\sigma(i)}(\boldsymbol{x}_{i-1}^k) = \nabla f_{\sigma(i)}(\boldsymbol{0}) + \nabla f_{\sigma(i)}(\boldsymbol{x}_0^k) - \nabla f_{\sigma(i)}(\boldsymbol{0}) + \nabla f_{\sigma(i)}(\boldsymbol{x}_{i-1}^k) - \nabla f_{\sigma(i)}(\boldsymbol{x}_0^k)$$

$$= \nabla f_{\sigma(i)}(\boldsymbol{0}) + \underbrace{\left[\int_0^1 \nabla^2 f_{\sigma(i)}(t\boldsymbol{x}_0^k)dt\right]}_{=:\boldsymbol{A}_{\sigma(i)}^k} \boldsymbol{x}_0^k + \underbrace{\left[\int_0^1 \nabla^2 f_{\sigma(i)}(\boldsymbol{x}_0^k + t(\boldsymbol{x}_{i-1}^k - \boldsymbol{x}_0^k))dt\right]}_{=:\boldsymbol{B}_{\sigma(i)}^k}(\boldsymbol{x}_{i-1}^k - \boldsymbol{x}_0^k)$$

$$= \nabla f_{\sigma(i)}(\boldsymbol{0}) + \boldsymbol{A}_{\sigma(i)}^k \boldsymbol{x}_0^k + \boldsymbol{B}_{\sigma(i)}^k(\boldsymbol{x}_{i-1}^k - \boldsymbol{x}_0^k).$$

As discussed in Section A.2, the integrals $\boldsymbol{A}_{\sigma(i)}^k$ and $\boldsymbol{B}_{\sigma(i)}^k$ exist due to smoothness of $f_{\sigma(i)}$'s. Note that $\|\boldsymbol{A}_{\sigma(i)}^k\| \leq L$ and $\|\boldsymbol{B}_{\sigma(i)}^k\| \leq L$ due to $L$-smoothness of $f_{\sigma(i)}$'s, and $\frac{1}{n}\sum_{i=1}^n \boldsymbol{A}_{\sigma(i)}^k \succeq \mu\boldsymbol{I}$ due to $\mu$-strong convexity of $F$.

Plugging this into the update equation of $\boldsymbol{x}_1^k$, we get

$$\boldsymbol{x}_1^k = \boldsymbol{x}_0^k - \eta \nabla f_{\sigma(1)}(\boldsymbol{x}_0^k) = \boldsymbol{x}_0^k - \eta(\nabla f_{\sigma(1)}(\boldsymbol{0}) + \boldsymbol{A}_{\sigma(1)}^k \boldsymbol{x}_0^k)$$

$$= (\boldsymbol{I} - \eta\boldsymbol{A}_{\sigma(1)}^k)\boldsymbol{x}_0^k - \eta \nabla f_{\sigma(1)}(\boldsymbol{0}).$$

Substituting this to the update equation of $\boldsymbol{x}_2^k$,

$$\boldsymbol{x}_2^k = \boldsymbol{x}_1^k - \eta \nabla f_{\sigma(2)}(\boldsymbol{x}_1^k)$$

$$= \boldsymbol{x}_1^k - \eta(\nabla f_{\sigma(2)}(\boldsymbol{0}) + \boldsymbol{A}_{\sigma(2)}^k \boldsymbol{x}_0^k + \boldsymbol{B}_{\sigma(2)}^k(\boldsymbol{x}_1^k - \boldsymbol{x}_0^k))$$

$$= (\boldsymbol{I} - \eta\boldsymbol{B}_{\sigma(2)}^k)[(\boldsymbol{I} - \eta\boldsymbol{A}_{\sigma(1)}^k)\boldsymbol{x}_0^k - \eta \nabla f_{\sigma(1)}(\boldsymbol{0})] - \eta \nabla f_{\sigma(2)}(\boldsymbol{0}) - \eta\boldsymbol{A}_{\sigma(2)}^k \boldsymbol{x}_0^k + \eta\boldsymbol{B}_{\sigma(2)}^k \boldsymbol{x}_0^k$$

$$= [\boldsymbol{I} - \eta \boldsymbol{A}_{\sigma(2)}^k - \eta(\boldsymbol{I} - \eta \boldsymbol{B}_{\sigma(2)}^k)\boldsymbol{A}_{\sigma(1)}^k]\boldsymbol{x}_0^k - \eta \nabla f_{\sigma(2)}(\boldsymbol{0}) - \eta(\boldsymbol{I} - \eta \boldsymbol{B}_{\sigma(2)}^k)\nabla f_{\sigma(1)}(\boldsymbol{0}).$$

Repeating this, one can write the last iterate $\boldsymbol{x}_n^k$ (or equivalently, $\boldsymbol{x}_0^{k+1}$) of the epoch as the following:

$$\boldsymbol{x}_0^{k+1} = \underbrace{\left[\boldsymbol{I} - \eta \sum_{j=1}^{n}\left(\prod_{t=n}^{j+1}(\boldsymbol{I} - \eta \boldsymbol{B}_{\sigma(t)}^k)\right)\boldsymbol{A}_{\sigma(j)}^k\right]}_{=:\widetilde{\boldsymbol{S}}_k}\boldsymbol{x}_0^k - \eta \underbrace{\left[\sum_{j=1}^{n}\left(\prod_{t=n}^{j+1}(\boldsymbol{I} - \eta \boldsymbol{B}_{\sigma(t)}^k)\right)\nabla f_{\sigma(j)}(\boldsymbol{0})\right]}_{=:\widetilde{\boldsymbol{t}}_k}$$

$$= \widetilde{\boldsymbol{S}}_k \boldsymbol{x}_0^k - \eta \widetilde{\boldsymbol{t}}_k.$$

Now, repeating this $K$ times, we get the equation for the iterate after $K$ epochs, which we take as the output of the algorithm:

$$\boldsymbol{x}_0^{K+1} = \left(\prod_{k=K}^{1}\widetilde{\boldsymbol{S}}_k\right)\boldsymbol{x}_0^1 - \eta \sum_{k=1}^{K}\left(\prod_{t=K}^{k+1}\widetilde{\boldsymbol{S}}_t\right)\widetilde{\boldsymbol{t}}_k = \widetilde{\boldsymbol{S}}_{K:1}\boldsymbol{x}_0^1 - \eta \sum_{k=1}^{K}\widetilde{\boldsymbol{S}}_{K:k+1}\widetilde{\boldsymbol{t}}_k.$$

We aim to get an upper bound on $\mathbb{E}[\|\boldsymbol{x}_0^{K+1}\|^2]$, where the expectation is over the randomness of permutation $\sigma$. To this end, using $\|\boldsymbol{a} + \boldsymbol{b}\|^2 \leq 2\|\boldsymbol{a}\|^2 + 2\|\boldsymbol{b}\|^2$,

$$\|\boldsymbol{x}_0^{K+1}\|^2 \leq 2\left\|\widetilde{\boldsymbol{S}}_{K:1}\boldsymbol{x}_0^1\right\|^2 + 2\eta^2\left\|\sum_{k=1}^{K}\widetilde{\boldsymbol{S}}_{K:k+1}\widetilde{\boldsymbol{t}}_k\right\|^2. \tag{F.1}$$

The remaining proof bounds each of the terms, using the following two lemmas. The proofs of Lemmas F.2 and F.3 are deferred to Sections F.2, and F.3, respectively.

**Lemma F.2.** *For any* $0 \leq \eta \leq \frac{1}{5nL\kappa}$, *any permutation* $\sigma$, *and* $k \in [K]$, *we have*

$$\left\|\widetilde{\boldsymbol{S}}_k\right\| \leq 1 - \frac{\eta n \mu}{2}.$$

**Lemma F.3.** *For any* $0 \leq \eta \leq \frac{1}{5nL\kappa}$,

$$\mathbb{E}\left[\left\|\sum_{k=1}^{K}\widetilde{\boldsymbol{S}}_{K:k+1}\widetilde{\boldsymbol{t}}_k\right\|^2\right] \leq \frac{66nL^2G^2\log n}{\mu^2}.$$

Since Lemma F.2 holds for any permutation $\sigma$ and $k \in [K]$ (for $\eta \leq \frac{1}{5nL\kappa}$), we have

$$\left\|\widetilde{\boldsymbol{S}}_{K:1}\boldsymbol{x}_0^1\right\|^2 \leq \left(\prod_{k=1}^{K}\left\|\widetilde{\boldsymbol{S}}_k\right\|^2\right)\|\boldsymbol{x}_0^1\|^2 \leq \left(1 - \frac{\eta n \mu}{2}\right)^{2K}\|\boldsymbol{x}_0^1\|^2.$$

The second term is bounded by Lemma F.3, which uses Lemma F.2 in its proof.

Substituting these bounds to (F.1), we have

$$\mathbb{E}[\|\boldsymbol{x}_0^{K+1}\|^2] \leq 2\left(1 - \frac{\eta n \mu}{2}\right)^{2K}\|\boldsymbol{x}_0^1\|^2 + \frac{132\eta^2 nL^2G^2\log n}{\mu^2}.$$

Now substitute the step size $\eta = \frac{2\log(n^{1/2}K)}{\mu nK}$. Then, we get

$$\mathbb{E}[\|\boldsymbol{x}_0^{K+1}\|^2] \leq \frac{2\|\boldsymbol{x}_0^1\|^2}{nK^2} + \mathcal{O}\left(\frac{L^2G^2\log^3(nK)}{\mu^4 nK^2}\right),$$

and in terms of the function value,

$$\mathbb{E}[F(\boldsymbol{x}_0^{K+1}) - F^*] \leq \frac{2L\|\boldsymbol{x}_0^1\|^2}{nK^2} + \mathcal{O}\left(\frac{L^3G^2\log^3(nK)}{\mu^4 nK^2}\right).$$

Recall that the bound holds for $\eta \leq \frac{1}{5nL\kappa}$, so $K$ must be large enough so that

$$\frac{2\log(n^{1/2}K)}{\mu nK} \leq \frac{1}{5nL\kappa}.$$

This gives us the epoch condition $K \geq 10\kappa^2\log(n^{1/2}K)$.

## F.2  Proof of Lemma F.2

**Decomposition into (modified) elementary polynomials.**  We expand $\widetilde{S}_k$ in the following way:

$$\widetilde{S}_k = I - \eta \sum_{j=1}^{n} \left( \prod_{t=n}^{j+1} (I - \eta B^k_{\sigma(t)}) \right) A^k_{\sigma(j)} = \sum_{m=0}^{n} (-\eta)^m \underbrace{\sum_{1 \le t_1 < \cdots < t_m \le n} B^k_{\sigma(t_m)} \cdots B^k_{\sigma(t_2)} A^k_{\sigma(t_1)}}_{=: \widetilde{e}_m},$$

where $\widetilde{e}_m$ be viewed as a modified version of noncommutative elementary polynomial (B.1). Since $k$ and $\sigma$ are fixed in this section, we use $A$ to denote the mean $\frac{1}{n} \sum_{i=1}^{n} A^k_{\sigma(i)}$. Recall that by definition of $A^k_{\sigma(i)}$'s and strong convexity of $F := \frac{1}{n} \sum_i f_i$, we have $A \succeq \mu I$. In what follows, we will decompose $\widetilde{S}_k$ into the sum of $1 - \eta n A$ and remainder terms. By bounding the spectral norm of $1 - \eta n A$ and the remainder terms, we will get the desired bound on the spectral norm of $\widetilde{S}_k$.

**Spectral norm bound.**  It is easy to check that $\widetilde{e}_0 = I$ and $\widetilde{e}_1 = \sum_{i=1}^{n} A^k_{\sigma(i)} = nA$, so

$$\widetilde{S}_k = I - \eta n A + \sum_{m=2}^{n} (-\eta)^m \widetilde{e}_m,$$

and we get the spectral norm bound

$$\left\| \widetilde{S}_k \right\| \le \| I - \eta n A \| + \sum_{m=2}^{n} \eta^m \| \widetilde{e}_m \| . \tag{F.2}$$

It is now left to bound each of the norms.

**Bounding each term of the spectral norm bound.**  First, note that for any eigenvalue $s$ of the positive definite matrix $A$, the corresponding eigenvalue of $I - \eta n A$ is $1 - \eta n s$. Recall $\eta \le \frac{1}{5nL\kappa} \le \frac{1}{5nL}$, so $\eta n s \le 1/5$ for any eigenvalue $s$ of $A$. Since the function $t \mapsto 1 - t$ is positive and decreasing on $[0, 0.2]$, the maximum singular value (i.e., spectral norm) of $I - \eta n A$ comes from the minimum eigenvalue of $A$. Hence,

$$\| I - \eta n A \| \le 1 - \eta n \mu.$$

Next, consider $\| \widetilde{e}_m \|$. It contains $\binom{n}{m}$ terms, and each of the terms have spectral norm bounded above by $L^m$. This gives

$$\| \widetilde{e}_m \| \le \binom{n}{m} L^m \le (nL)^m.$$

**Concluding the proof.**  Substituting the bounds to (F.2) yields

$$\left\| \widetilde{S}_k \right\| \le 1 - \eta n \mu + \sum_{m=2}^{n} (\eta n L)^m \le 1 - \eta n \mu + \frac{(\eta n L)^2}{1 - \eta n L} \le 1 - \eta n \mu + \frac{5}{4} (\eta n L)^2,$$

where the last inequality used $\eta n L \le 1/5$. The remaining step is to show that the right hand side of the inequality is bounded above by $1 - \frac{\eta n \mu}{2}$ for $0 \le \eta \le \frac{1}{5nL\kappa}$.

Define $z = \eta n L$. Using this, we have

$$1 - \eta n \mu + \frac{5}{4} (\eta n L)^2 \le 1 - \frac{\eta n \mu}{2} \text{ for } 0 \le \eta \le \frac{1}{5nL\kappa}$$

$$\Leftrightarrow g(z) := \frac{z}{2\kappa} - \frac{5z^2}{4} \ge 0 \text{ for } 0 \le z \le \frac{1}{5\kappa},$$

so it suffices to show the latter. One can check that $g(0) = 0$, $g'(0) > 0$ and $g'(z)$ is monotonically decreasing in $z \ge 0$, so $g(z) \ge 0$ holds for $z \in [0, c]$ for some $c > 0$. This also means that if we have $g(c) \ge 0$ for some $c > 0$, $g(z) \ge 0$ for all $z \in [0, c]$.

Consider $z = \frac{1}{5\kappa}$. Substituting to $g$ gives

$$g\left( \frac{1}{5\kappa} \right) = \frac{1}{10\kappa^2} - \frac{1}{20\kappa^2} = \frac{1}{20\kappa^2} > 0.$$

This means that $g(z) \ge 0$ for $0 \le z \le \frac{1}{5\kappa}$, hence proving the lemma.

## F.3 Proof of Lemma F.3

First, note that if $0 \leq \eta \leq \frac{1}{5nL\kappa}$, Lemma F.2 tells us that the following holds for any $k \in [K]$ and any underlying permutation $\sigma$:

$$\left\| \widetilde{\boldsymbol{S}}_k \right\| \leq 1 - \frac{\eta n \mu}{2}.$$

Therefore, for any permutation $\sigma$, we have

$$\left\| \sum_{k=1}^{K} \widetilde{\boldsymbol{S}}_{K:k+1} \widetilde{\boldsymbol{t}}_k \right\|^2 \leq \left( \sum_{k=1}^{K} \left\| \widetilde{\boldsymbol{S}}_{K:k+1} \widetilde{\boldsymbol{t}}_k \right\| \right)^2 \leq \left( \sum_{k=1}^{K} \left(1 - \frac{\eta n \mu}{2}\right)^{K-k} \left\| \widetilde{\boldsymbol{t}}_k \right\| \right)^2. \quad \text{(F.3)}$$

Now, it is left to bound the right hand side of the inequality (F.3), which involves $\|\widetilde{\boldsymbol{t}}_k\|$. The proof technique used to bound $\|\widetilde{\boldsymbol{t}}_k\|$ is similar to Lemma B.2; we use the Hoeffding-Serfling inequality [13] and union bound.

Due to summation by parts, the following identity holds, even when multiplication of $a_j$ and $b_j$ is noncommutative:

$$\sum_{j=1}^{n} a_j b_j = a_n \sum_{j=1}^{n} b_j - \sum_{i=1}^{n-1} (a_{i+1} - a_i) \sum_{j=1}^{i} b_j.$$

We can apply the identity to $\widetilde{\boldsymbol{t}}_k$, by substituting $a_j = \prod_{t=n}^{j+1}(\boldsymbol{I} - \eta \boldsymbol{B}_{\sigma(t)}^k)$ and $b_j = \nabla f_{\sigma(j)}(\boldsymbol{0})$:

$$\left\| \widetilde{\boldsymbol{t}}_k \right\| = \left\| \sum_{j=1}^{n} \left( \prod_{t=n}^{j+1} (\boldsymbol{I} - \eta \boldsymbol{B}_{\sigma(t)}^k) \right) \nabla f_{\sigma(j)}(\boldsymbol{0}) \right\|$$

$$= \left\| \sum_{j=1}^{n} \nabla f_{\sigma(j)}(\boldsymbol{0}) - \sum_{i=1}^{n-1} \left( \prod_{t=n}^{i+2} (\boldsymbol{I} - \eta \boldsymbol{B}_{\sigma(t)}^k) - \prod_{t=n}^{i+1} (\boldsymbol{I} - \eta \boldsymbol{B}_{\sigma(t)}^k) \right) \sum_{j=1}^{i} \nabla f_{\sigma(j)}(\boldsymbol{0}) \right\|$$

$$= \left\| \eta \sum_{i=1}^{n-1} \left( \prod_{t=n}^{i+2} (\boldsymbol{I} - \eta \boldsymbol{B}_{\sigma(t)}^k) \right) \boldsymbol{B}_{\sigma(i+1)}^k \sum_{j=1}^{i} \nabla f_{\sigma(j)}(\boldsymbol{0}) \right\|$$

$$\leq \eta \sum_{i=1}^{n-1} \left\| \left( \prod_{t=n}^{i+2} (\boldsymbol{I} - \eta \boldsymbol{B}_{\sigma(t)}^k) \right) \boldsymbol{B}_{\sigma(i+1)}^k \sum_{j=1}^{i} \nabla f_{\sigma(j)}(\boldsymbol{0}) \right\| \leq \eta L (1 + \eta L)^n \sum_{i=1}^{n-1} \left\| \sum_{j=1}^{i} \nabla f_{\sigma(j)}(\boldsymbol{0}) \right\|, \quad \text{(F.4)}$$

where the last step used $\|\boldsymbol{B}_{\sigma(t)}^k\| \leq L$. Recall that $\eta \leq \frac{1}{5nL\kappa} \leq \frac{1}{5nL}$, which implies that $(1 + \eta L)^n \leq e^{1/5}$. Also, note that the right hand side of the inequality now does *not* depend on $k$. Thus, any bound on the norm of partial sums $\|\sum_{j=1}^{i} \nabla f_{\sigma(j)}(\boldsymbol{0})\|$ applies to *all* $\widetilde{\boldsymbol{t}}_k$. Next, we use the Hoeffding-Serfling inequality (Lemma A.1) for bounded random vectors. We restate the lemma here, for readers' convenience.

**Lemma A.1** ([13, Theorem 2]). *Suppose $n \geq 2$. Let $\boldsymbol{v}_1, \boldsymbol{v}_2, \ldots, \boldsymbol{v}_n \in \mathbb{R}^d$ satisfy $\|\boldsymbol{v}_j\| \leq G$ for all $j$. Let $\bar{\boldsymbol{v}} = \frac{1}{n} \sum_{j=1}^{n} \boldsymbol{v}_j$. Let $\sigma \in \mathcal{S}_n$ be a uniform random permutation of $n$ elements. Then, for $i \leq n$, with probability at least $1 - \delta$, we have*

$$\left\| \frac{1}{i} \sum_{j=1}^{i} \boldsymbol{v}_{\sigma(j)} - \bar{\boldsymbol{v}} \right\| \leq G \sqrt{\frac{8(1 - \frac{i-1}{n}) \log \frac{2}{\delta}}{i}}.$$

Recall that $\bar{\boldsymbol{v}} = \frac{1}{n} \sum_{j=1}^{n} \nabla f_j(\boldsymbol{0}) = \boldsymbol{0}$ in our setting, so with probability at least $1 - \delta$, we have

$$\left\| \sum_{j=1}^{i} \nabla f_{\sigma(j)}(\boldsymbol{0}) \right\| \leq G \sqrt{8i \log \frac{2}{\delta}}.$$

Using the union bound for all $i = 1, \ldots, n-1$, we have with probability at least $1 - \delta$,

$$\sum_{i=1}^{n-1} \left\| \sum_{j=1}^{i} \nabla f_{\sigma(j)}(\mathbf{0}) \right\| \leq G\sqrt{8\log\frac{2n}{\delta}} \sum_{i=1}^{n-1} \sqrt{i} \leq G\sqrt{8\log\frac{2n}{\delta}} \int_{1}^{n} \sqrt{y} dy \leq \frac{2G}{3}\sqrt{8\log\frac{2n}{\delta}} n^{3/2}.$$

(F.5)

Substituting this to (F.4) leads to the following bound that holds for all $k \in [K]$, without having to invoke any union bounds over different $k$'s:

$$\left\| \widetilde{\boldsymbol{t}}_k \right\| \leq \frac{4\sqrt{2}e^{1/5}}{3} \eta n^{3/2} LG \sqrt{\log\frac{2n}{\delta}}.$$

Using this bound, we can bound the right hand side of (F.3) as follows:

$$\left( \sum_{k=1}^{K} \left(1 - \frac{\eta n \mu}{2}\right)^{K-k} \left\| \widetilde{\boldsymbol{t}}_k \right\| \right)^2 \leq \left( \frac{4\sqrt{2}e^{1/5}}{3} \eta n^{3/2} LG \sqrt{\log\frac{2n}{\delta}} \sum_{k=1}^{K} \left(1 - \frac{\eta n \mu}{2}\right)^{K-k} \right)^2$$

$$\leq \left( \frac{8\sqrt{2}e^{1/5}n^{1/2}LG}{3\mu} \sqrt{\log\frac{2n}{\delta}} \right)^2$$

$$= \frac{128e^{2/5}nL^2G^2}{9\mu^2} \log\frac{2n}{\delta}.$$

which holds with probability at least $1 - \delta$.

Now, set $\delta = 1/n$, and let $E$ be the probabilistic event that (F.5) holds. Let $E^c$ be the complement of $E$. Given the event $E^c$, directly bounding (F.4) leads to

$$\left\| \widetilde{\boldsymbol{t}}_k \right\| \leq e^{1/5}\eta L \sum_{i=1}^{n-1} \left\| \sum_{j=1}^{i} \nabla f_{\sigma(j)}(\mathbf{0}) \right\| \leq \frac{e^{1/5}\eta n^2 LG}{2},$$

which yields the following bound on (F.3), conditional on $E^c$:

$$\left( \sum_{k=1}^{K} \left(1 - \frac{\eta n \mu}{2}\right)^{K-k} \left\| \widetilde{\boldsymbol{t}}_k \right\| \right)^2 \leq \left( \frac{e^{1/5}\eta n^2 LG}{2} \sum_{k=1}^{K} \left(1 - \frac{\eta n \mu}{2}\right)^{K-k} \right)^2 \leq \left( \frac{e^{1/5}nLG}{\mu} \right)^2 = \frac{e^{2/5}n^2L^2G^2}{\mu^2}.$$

Finally, putting everything together and using $\log\frac{2n}{\delta} = \log(2n^2) \leq 3\log n$,

$$\mathbb{E}\left[ \left\| \sum_{k=1}^{K} \widetilde{\boldsymbol{S}}_{K:k+1}\widetilde{\boldsymbol{t}}_k \right\|^2 \right] = \mathbb{E}\left[ \left\| \sum_{k=1}^{K} \widetilde{\boldsymbol{S}}_{K:k+1}\widetilde{\boldsymbol{t}}_k \right\|^2 \mid E \right] \mathbb{P}[E] + \mathbb{E}\left[ \left\| \sum_{k=1}^{K} \widetilde{\boldsymbol{S}}_{K:k+1}\widetilde{\boldsymbol{t}}_k \right\|^2 \mid E^c \right] \mathbb{P}[E^c]$$

$$\leq \frac{128e^{2/5}nL^2G^2\log n}{3\mu^2} + \frac{e^{2/5}n^2L^2G^2}{\mu^2}\frac{1}{n}$$

$$\leq \frac{66nL^2G^2\log n}{\mu^2},$$

which finishes the proof.

## Footnotes

[1]Note that since we have already taken the union bound over all $i = 1, \ldots, n-1$ and $k = 1, \ldots, K$ in Section A.3, additional union bounds are not needed.