[Reviews · NeurIPS 2020]

Review 1

Summary and Contributions: The paper studies SGD with shuffling method for solving finite sum optimization problems in a variety of settings. They make a number of contributions, including optimal rates without component convexity, as well as small/no epoch requirements and poly-log factor free bounds. Moreover, their results imply a number of interesting insights, for example there is no advantage of additional shuffling when optimizing smooth strongly convex functions. Furthermore, towards proving these results, they make a number of key technical contributions. --------------- Post author feedback comments --------------- I have read the author feedback, and keep my score unchanged.

Strengths: 1. The paper is extremely well-written, with all the subtleties and gaps in (existing) knowledge carefully explained. Moreover, the paper does a very good job of concisely surveying the results and techniques of previous work. All in all, the main paper as well as the supplementary is very polished - the technical challenges, key ideas, proofs (and proof sketches), as well as the open questions this work points to are discussed very well. 2. The paper makes a number of fundamental contributions in understanding SGD with shuffling in various settings, some of which I listed in the summary above. Moreover, given the number of different settings that the authors consider, the writing could have easily become confusing, but the authors do a very good job of the organization and presentation in the paper. 3. The paper also contributes new technical tools, for example: a variant of Chung's lemma (as well the correct proof of the original), matrix AM-GM-like inequality, which could be useful in general.

Weaknesses: -

Correctness: I skimmed through the poofs (mainly of Theorem 1 and 2), and they appear correct (to me). Moreover, the general proof ideas are sound and explained well.

Clarity: Very well-written.

Relation to Prior Work: Yes, discussed well.

Reproducibility: Yes

Additional Feedback:


Review 2

Summary and Contributions: Post Author Response update: Thanks for the detailed response. I'm happy with your answers, raising evaluation to 7. ntations of SGD as practiced in ML: sampling a permutation of the data and then sequentially going through the data using this permutation to define stochastic gradient updates till the permutation (or epoch) is exhausted, at which point either a new permutation is sampled or the original permutation is re-used from the beginning. The authors claim that their analysis of the methods provide a more nuanced rate expressions for finite-time performance of the shuffled-SGD methods under slightly broader conditions.

Strengths: The paper seeks to make their case by claiming to ground their rate analysis on weaker restrictions and more reasonable assumptions. The paper is very well written and a quick perusal of the proof (in the supplement) of the first theorem shows no obvious flaws. However, the significance of the results beyond existing literature, as described by the authors in Sec 1.3, seem minor: for instance, not assuming that constituent functions f_i of the sum F = \sum_i f_i are not convex , but assuming the properties only for the sum F. In another instance, the authors compare results obtained for constant-step algorithms (existing literature ) with decreasing step algorithms (Thm 3&4) to show improvements in rates, but this is a bit oranges and apples.

Weaknesses: - Distinction between F and constituent f_i being convex or strongly convex: why does this matter in practice? In the general learning case, f_i(x) is usually of the form f(x, \xi_i) where \xi_i is the i-th scenario or data. So, many common properties like smoothness, convexity etc. are naturally assumed to "flow" from f to the sample/population average F. What practical scenario do you envision where f_i could be distinct functional forms that are not convex but F is? where would an example like this make sense: f_1 (x)=-x^2 (concave) , f_2 (x)= 2x^2 leading to F(x) = x^2 (convex) ? - Assumption 1: this is quite unsatisfactory because you are imposing a restriction on the *optimization* formulation in order to gain an *algorithmic* benefit. Is there any guarantee that doing something like projecting all iterates on to your \cal{S}_{x_0} set (defined on pg 3 of the supplement) to enforce this assumption will not lead to sub optimal results? - Indeed, the use of A1 (or similar "closeness of iterates" arguments) in establishing some form of (3.1) raises the troubling implication that may be we do not need to bother with shuffling of any sort, just take deterministic full-batch sized gradient descent steps. How does your analysis of the progress made in the shuffled SGD steps compare to progress that would have been made w.r.t epoch count K using deterministic full gradient descent with a proper step-length determination like Armijo rule?

Correctness: The paper is very well written and a quick perusal of the proof (in the supplement) of the first theorem shows no obvious flaws. The claims and methods feel sound.

Clarity: The paper is very well written and clear and detailed in its description of prior-art and the current contributions. Proofs are well explained with appropriate sketches.

Relation to Prior Work: yes, see above answer on 'well written'.

Reproducibility: Yes

Additional Feedback:


Review 3

Summary and Contributions: 1. The paper gives convergence rates for the widely used algorithms Random Shuffling and Single Shuffle. Although these are used a lot in practice, these have remained theoretically difficult to analyze. 2. This is the first paper that proves tight upper bounds for the Single Shuffle algorithm, for strongly convex functions, under mild assumptions. 3. The paper improves upon the results of previous works on Random Shuffling in many aspects, like reducing large epoch requirements and individual function convexity. 4. The paper also proves that for varying step size regime, the poly log factors in the upper bounds for Random Shuffling can be removed.

Strengths: 1. This is the first paper to provide tight upper bound for Single Shuffle algorithm for strongly convex functions, under the mild assumptions of epoch requirements and no assumptions on the individual function convexity. 2. For Random Shuffle algorithm, the paper improves upon previous works in multiple ways like removing the large epoch requirements and the individual convexity requirement. 3. The analysis seems to be simpler that the Wasserstein distance and coupling based analysis of prior work. 4. By analyzing the varying step sizes regime, the paper removes the poly-log factors in error rates of prior work. Note however, that the corresponding upper bounds may still not be tight because error lower bounds for varying step sizes have not been proposed yet. 5. The removal of individual convexity is really important in the following sense: There can be settings where the overall function $F$ behaves like a strongly convex function near its minima, whereas one (or more) of its component functions is non-convex in the same neighborhood. This can arise in the case of deep neural network training. The results of this paper will hold even in these settings.

Weaknesses: My only concern is regarding the existence of the constant $G$. The constant can grow with $n$. For example if $b_{2i-1}=-i$ and $b_{2i}=i$, then for any even $n$, the minimizer of $F$ is at 0 and the constant $G$ grows linearly in $n$. Note that the other problem constants like $\mu$ and $\kappa$ still remain unchanged. Thus, even though $G$ exists, the bound of Theorem 2 for example, has an extra multiplicative factor of $\Omega(n^2)$, making it suboptimal now. To handle this issue, the statement of Theorem 2 should specify explicitly the dependence on $G$ along with the definition of this constant. ----- Post author feedback comments ----- The authors have agreed to address my concern if the paper gets accepted.

Correctness: The proofs seem to be correct.

Clarity: The paper is really well written.

Relation to Prior Work: The paper discusses prior work in great detail. The literature survey is really comprehensive and I enjoyed reading it.

Reproducibility: Yes

Additional Feedback: Typos: line 284: finer -> fine line 294: limitations in -> limitations of


Review 4

Summary and Contributions: The authors give 4 theorems providing error guarantees on SGD with shuffling for finite sum optimization, under varying assumptions on the regularity, convexity, "quadraticity" of the considered functions, and on the number of epochs. These results outperform prior art, and in several cases match previously established lower bounds.

Strengths: The results appear to be quite significant, and highly non trivial. The writing is clear, which is noticeable given the technical nature of the results.

Weaknesses: No serious weakness spotted.

Correctness: I did a couple of checks of the proofs, and did not spot any mistakes.

Clarity: Yes.

Relation to Prior Work: Yes.

Reproducibility: Yes

Additional Feedback: In the proof of Theorem 1, supplementary material page 3, there is a constant G which appears in the bound in line 64, which is not explicited in the statement in the main text. It would be better if the authors either explicited it, or bounded it in terms of the other problem parametes (kappa, L mu etc).

[Author Response · NeurIPS 2020]

**Response to Reviewer 1.** Thank you for your compliments on our contributions as well as our presentation!

**Response to Reviewer 2.** Thank you for your thoughtful comments and questions!

*Q. Significance of relaxing component convexity?* For practical significance, it is crucial for understanding the
(local) behavior of shuffling algorithms for nonconvex problems, such as neural network training. As Reviewer 3 also
mentioned, in such settings, the function $F$ would behave like convex near a neighborhood around a local minimum,
while each component function $f_i$ could be highly nonconvex in the neighborhood.

For theoretical significance, this relaxation signifies an innovation in our proof techniques. In fact, the component
convexity is heavily exploited in the previous works [10, 13]. At a high level, the previous works use this assumption to
ensure that the algorithm makes a sufficient progress during *each* iteration. In contrast, our proof technique proves fast
convergence rates without having to show such a per-iterate progress. We also highlight that it is thanks to our proof
technique that we obtain the first optimal rate for nonconvex PL function class in Theorem 1.

*Q. Constant step algorithms vs. decreasing step size algorithms are not comparable?* We view the decreasing step
size case as choosing a different set of hyperparameters for the *same* algorithm. In light of this, our main goal in
analyzing decreasing step sizes was to see whether the common limitation in the prior works (large epoch requirement)
is inherent to the algorithm. Our main results indeed show that the common limitation is a derivative of the constant step
size choice. On a technical note, capturing the desired convergence rate with decreasing step sizes requires a non-trivial
analysis (Section 6.3 or Appendix E), and we believe that our work develops a toolkit for analyzing decreasing step
sizes for epoch-based algorithms like shuffling SGD.

*Q. Guarantees without Assumption 1?* We agree that Assumption 1 is a bit unsatisfactory, but please note that
this assumption is also present in many prior results (see Line 144). While our Theorems 1 and 2 *do not require*
Assumption 1, Theorems 3 and 4 *do* rely on Assumption 1 because they are built on existing results [10, 13] that
make use of Assumption 1. We believe that removing this assumption, as done in [12] or a concurrent work "Random
Reshuffling: Simple Analysis with Vast Improvements," is an important future research direction. We will add a remark
on this in our revision.

*Q. Comparison to GD?* Our current proof techniques do not show a regime where rates for shuffling SGD can be
better than that of GD. Indeed, this limitation is shared among the existing works, as their analyses rely on comparing
the epoch progress of shuffling SGD to that of GD. In other words, this question is currently beyond the scope of
existing proof techniques, and in particular, finding a regime where shuffling SGD outperforms both SGD and GD
would be an interesting future direction to pursue. We will add this discussion in our revision.

**Response to Reviewer 3.** Thank you for detailing our contributions into a list. In particular, we agree that the removal
of individual convexity has important consequences in practical applications. Also, as per your suggestion, we will
explicitly state the definition of the constant $G$ in Theorem 2.

**Response to Reviewer 4.** Thank you for appreciating the strengths of our paper! We did not spell out $G$ in Theorem 1
because existence of $G$ is implied by the other assumptions. We will explicitly state the constant $G$ in the theorem
statement as per your suggestion.

On a separate note, we would like to fix one technical mistake in the proof of Theorem 1. After the submission
deadline, it came to our attention that the Hoeffding-Serfling (HS) inequality used in the proof of Theorem 1 can be
applied only to RANDOMSHUFFLE. In SINGLESHUFFLE, the first iterate $x_0^k$ of the $k$-th epoch is not independent
of the permutation $\sigma$ as soon as $k > 1$, hence we cannot apply the HS inequality. In light of this, we note that
Theorem 1 only holds for RANDOMSHUFFLE. As to our claims on SINGLESHUFFLE, we will add an additional theorem
in the revision. The new theorem shows that if $F$ is $\mu$-strongly convex and $f_i$'s are $L$-smooth, SINGLESHUFFLE
with number of epochs $K \geq 10\kappa^2 \log(n^{1/2}K)$, step size $\eta_i^k = \eta := 2\log(n^{1/2}K)/\mu nK$, and initialization $x_0$ satisfies
$\mathbb{E}[F(x_0^{K+1})] - F^* \leq 2L\|x_0 - x^*\|^2/nK^2 + c \cdot \log^3(nK)/nK^2$, for some $c = O(\kappa^4)$.

Although this change slightly weakens our initial claim on SINGLESHUFFLE, we believe this new theorem is still
interesting progress, because (i) it is a tight bound (up to log factors) for SINGLESHUFFLE on strongly convex functions;
(ii) it does not require convexity of individual components or bounded iterates assumption (Assumption 1); (iii) it
shows that the optimal rates for minimizing strongly convex functions are the same for RANDOMSHUFFLE and SIN-
GLESHUFFLE; and (iv) the proof can be easily extended to any algorithms between the spectrum of RANDOMSHUFFLE
and SINGLESHUFFLE. The proof of the new theorem involves an end-to-end analysis similar to the one-dimensional
quadratic result [16] and a modified version of the approximate matrix AM-GM inequality in eq (6.4). For this theorem,
the HS inequality is applied only to the partial sums of the individual gradients at the global minimum $x^*$, which is
always independent of the permutation $\sigma$.

[Meta-Review · NeurIPS 2020]

All reviewers agree on the value and depth of this paper. The presented results are the best in the particular topic (in terms of tightness, and assumptions), and are very likely to impact the related without replacement SGD literature.